# TAB-DRW: A DFT-BASED ROBUST WATERMARK FOR GENERATIVE TABULAR DATA

## ABSTRACT

The rise of generative AI has enabled the production of high-fidelity synthetic tabular data across fields such as healthcare, finance, and public policy, raising growing concerns about data provenance and misuse. Watermarking offers a promising solution to address these concerns by ensuring the traceability of synthetic data, but existing methods face many limitations: they are computationally expensive due to reliance on large diffusion models, struggle with mixed discrete-continuous data, or lack robustness to post-modifications. To address them, we propose TAB-DRW, an efficient and robust post-editing watermarking scheme for generative tabular data. TAB-DRW embeds watermark signals in the frequency domain: it normalizes heterogeneous features via the Yeo–Johnson transformation and standardization, applies the discrete Fourier transform (DFT), and adjusts the imaginary parts of adaptively selected entries according to precomputed pseudorandom bits. To further enhance robustness and efficiency, we introduce a novel rank-based pseudorandom bit generation method that enables row-wise retrieval without incurring storage overhead. Experiments on five benchmark tabular datasets show that TAB-DRW achieves strong detectability and robustness against common post-processing attacks, while preserving high data fidelity and fully supporting mixed-type features.

## 1 INTRODUCTION

Tabular data is a predominant format for structured information in many fields such as healthcare, finance, and public policy (Borisov et al., 2022). It facilitates tasks such as decision-making, risk assessment, and resource allocation. However, access to high-quality tabular data is often restricted by privacy concerns, regulatory constraints on data sharing, and the cost of human annotation. Recent advances in generative AI have revolutionized synthetic data generation (Xu et al., 2019; Zhao et al., 2021; Zhang et al., 2021; Liu et al., 2023; Kotelnikov et al., 2023; Zhang et al., 2024a; Gulati & Roysdon, 2024; Wang & Nguyen, 2025), which creates high-fidelity tabular datasets that closely match real-world data. Synthetic tabular data now offers a compelling alternative for data sharing and model training in various domains (Bauer et al., 2024).

Despite its benefits, synthetic tabular data also introduces new risks. Misuse can lead to civil disputes, regulatory violations, or societal harm (Guo & Chen, 2024). For instance, generating synthetic datasets from copyrighted materials without authorization may infringe intellectual property rights (Vyas et al., 2023); in finance, synthetic transaction records can facilitate fraud (Assefa et al., 2020); in healthcare, biased or inaccurate synthetic patient data can mislead clinical decisions and cause harmful consequences (Qian et al., 2024). As synthetic data becomes increasingly realistic and widespread, ensuring accountability and provenance has become critical (Liu et al., 2024).

To address concerns surrounding the misuse of synthetic tabular data, watermarking has emerged as a promising solution. The core idea is to embed invisible statistical signals into synthetic data before release, allowing reliable detection by a verifier with access to secretly shared information. An effective watermarking scheme should satisfy four key properties: 1) **fidelity**, preserving the quality and utility of the data; 2) **detectability**, allowing reliable identification through a private detection process; 3) **applicability**, enabling efficient watermark embedding even after data generation and for mixed discrete-continuous tabular data; and 4) **robustness**, ensuring resilience against post-processing attacks such as deletions or value modifications (Podilchuk & Delp, 2001; Kuditipudi et al., 2024). Although significant progress has been made in watermarking text data (Kirchenbauer et al., 2023;

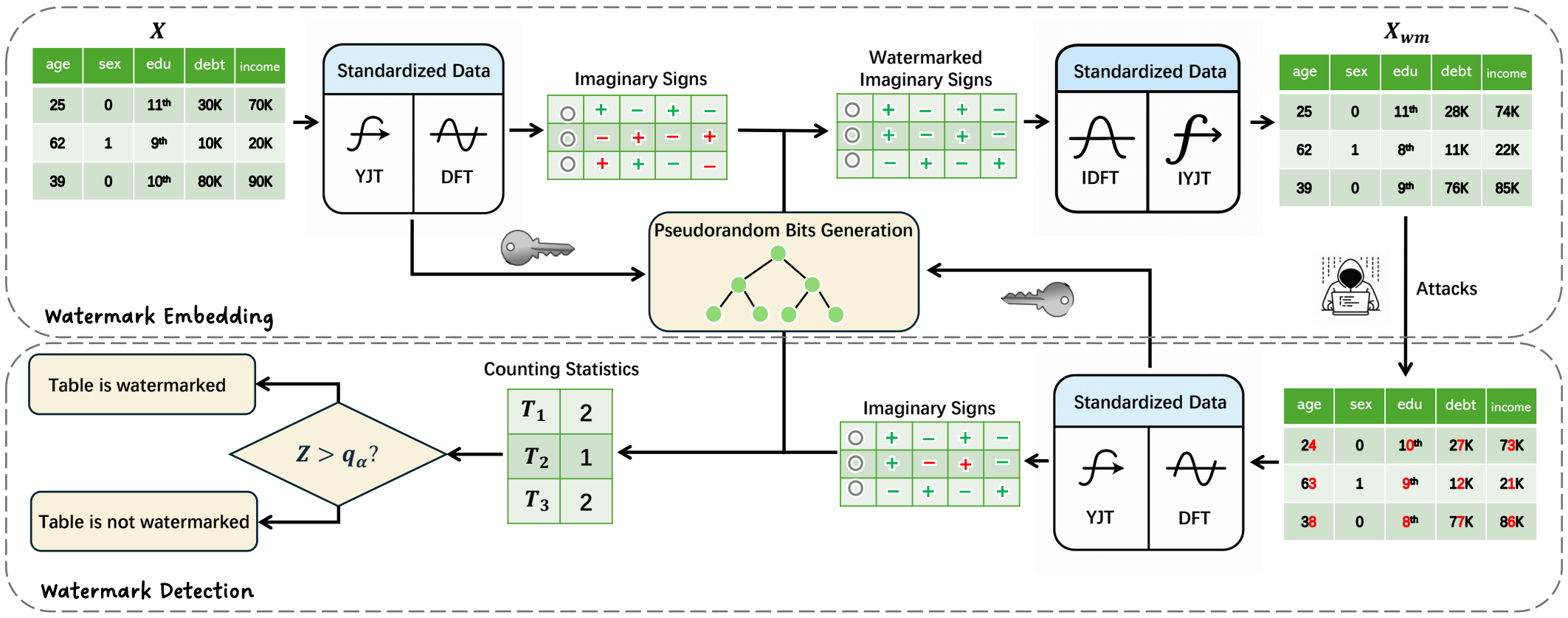

Figure 1: Our proposed watermarking scheme, Tab-Drw, embeds watermarks into tabular data by modifying the imaginary components of the frequency-domain representation to align with pseudo-random bits. Detection evaluates the degree of alignment: strong alignment indicates watermarked data, while weak alignment suggests non-watermarked data.

Table 1: Comparison of our method with existing works. ○ indicates “not satisfied”, ◐ indicates “partially satisfied”, and ● indicates ”satisfied”.

| Methods | Category | Data type | Fidelity | Detectability | Applicability | Robustness |
|---|---|---|---|---|---|---|
| Zhu et al. (2025) | Sampling-phase | Continuous & Discrete | ● | ● | ◐ | ◐ |
| Fang et al. (2025) | Sampling-phase | Continuous & Discrete | ● | ● | ◐ | ◐ |
| He et al. (2024) | Post-editing | Only continuous | ● | ● | ○ | ◐ |
| Zheng et al. (2024) | Post-editing | Continuous & Discrete | ● | ● | ○ | ◐ |
| **Tab-Drw (Ours)** | **Post-editing** | **Continuous & Discrete** | ● | ● | ● | ● |

Kuditipudi et al., 2024; Zhao et al., 2024) and images (Wen et al., 2023; Yang et al., 2024; Zhang et al., 2024b), existing watermarking schemes for synthetic tabular data fail to simultaneously achieve these four properties.

**Existing works.** Current approaches to watermarking synthetic tabular data mainly fall into two categories: sampling-phase watermarking (Zhu et al., 2025; Fang et al., 2025) and post-editing watermarking (He et al., 2024; Zheng et al., 2024). Sampling-phase methods typically change the sampling process of large diffusion models. Specifically, Zhu et al. (2025) embeds watermark signals into structured latent noise and detects the structure by measuring correlations with noise reconstructed via the inverse process. While achieving high fidelity and robustness, it relies on reversible sampling strategies, such as DDIM (Song et al., 2021), which is prone to reconstruction errors and computationally expensive. Fang et al. (2025) generates multiple samples at the same time and outputs the one with the highest pseudorandom score, which incurs higher computational cost though preserving generation quality. In contrast, post-editing methods are lightweight: they often modify generated or existing datasets with pseudorandom operations, but don’t change the sampling process or invoke large neural networks. For example, He et al. (2024) proposes a “green list” method that bins each tabular value into key-selected intervals and detects whether values fall into the designated sets. Although effective in preserving fidelity and detectability, it struggles with mixed-type (continuous and discrete) data and lacks robustness against noise attacks. Zheng et al. (2024) embeds watermark signals by perturbing key cells with values randomly selected from the so-called “green domain”. However, it requires storing the original dataset for perturbation recovery, leading to substantial space overhead in generative settings. Overall, existing methods offer valuable insights but fall short of providing a lightweight, robust, and broadly applicable solution for synthetic tabular data. See Appendix A for more details on related work.

**Our contribution.** In this work, we propose a new watermarking method that simultaneously satisfies the four desired properties above. Our contributions are summarized below.

1. **A new watermarking scheme.** We propose Tab-Drw, a post-editing watermarking method that embeds robust watermark signals in the frequency domain (see Figure 1 for the workflow and Section 2 for the formal introduction). Specifically, it modifies the discrete Fourier transform (DFT) representation of tabular data to align with a precomputed pseudorandom bit sequence.

Detection then evaluates the degree of alignment: strong alignment indicates watermarked data, while weak alignment suggests non-watermarked data. TAB-DRW is computationally efficient, requires no model access, and is applicable to both existing and generative tabular data while preserving data fidelity with minimal distortion. Optional rounding and outlier clipping are used to support mixed-type tabular data.

2. **A new pseudorandom bit generation method.** We introduce a rank-based pseudorandom bit generation scheme to further enhance robustness against post-processing attacks (see Figure 2; detailed in Section 2). This scheme ensures that small modifications to the tabular data do not change the underlying pseudorandom bits. It is also efficient; during detection, pseudorandom bits are retrieved by querying an implicit storage structure based on robust statistics computed from key-selected columns.

3. **Theoretical analysis and empirical validation.** We provide theoretical insights into the bias and robustness of TAB-DRW. In particular, we show that the embedded watermark signals remain detectable under moderate post-modification. Empirically, we evaluate TAB-DRW using TabSyn (Zhang et al., 2024a) as the synthetic data generator. Experiments across five benchmark tabular datasets demonstrate that TAB-DRW achieves superior detectability and robustness, while preserving high data fidelity compared to existing methods.

## 2 METHOD

### 2.1 WATERMARK EMBEDDING

**High-level description.** As shown in Figure 1, the embedding process can be divided into three steps. It begins by preprocessing the given tabular data through two transformations: first, a column-wise Yeo-Johnson transformation (YJT) (Yeo & Johnson, 2000) with standardization to reduce heterogeneity and unify the scale; second, a row-wise discrete Fourier transform (DFT) (Oppenheim, 1999) to map the data into the frequency domain. Next, the algorithm modifies the imaginary components of the frequency-domain representation to align with a precomputed pseudorandom bit sequence. Finally, it applies the inverse transformations and un-shuffling to reconstruct the modified data in the original domain, which is then released for public use and future detection. We provide a detailed explanation of these steps below. [1][2]

**Definition 1** (YJT)**.** *For an input $x \in \mathbb{R}$, the YJT $\Psi(\lambda, x)$ is defined as $\Psi(\lambda, x) = \frac{(x+1)^\lambda - 1}{\lambda}$ if $x \geq 0, \lambda \neq 0$; $\ln(x+1)$ if $x \geq 0, \lambda = 0$; $-\frac{(-x+1)^{2-\lambda}-1}{2-\lambda}$ if $x < 0, \lambda \neq 2$; $-\ln(-x+1)$ if $x < 0, \lambda = 2$. $\lambda$ is a parameter automatically selected to reduce heterogeneity (SciPy Developers, 2025).*

**Definition 2** (DFT and IDFT)**.** *Given a row of tabular data $\boldsymbol{x} = (x_0, \ldots, x_{p-1}) \in \mathbb{R}^p$, its DFT is defined as $\boldsymbol{y} = \texttt{DFT}(\boldsymbol{x}) := (y_0, \ldots, y_{p-1}) \in \mathbb{C}^p$ where $y_t = \frac{1}{\sqrt{p}} \sum_{n=0}^{p-1} x_n e^{-\mathrm{i}\frac{2\pi}{p}tn}$ for each $t = 0, \ldots, p-1$, The inverse DFT (IDFT) is given by $\boldsymbol{x} = \texttt{IDFT}(\boldsymbol{y}) \in \mathbb{R}^p$ where $x_n = \frac{1}{\sqrt{p}} \sum_{k=0}^{p-1} y_k e^{\mathrm{i}\frac{2\pi}{p}kn}$ for each $n = 0, \ldots, p-1$. Since $\texttt{IDFT} \circ \texttt{DFT} = \texttt{Id}$, their composition implies an exact recovery of the original input.*

**Step 1: Column-wise and row-wise transformations.** In general, features in a tabular dataset exhibit heterogeneous scales and types (continuous or discrete). This heterogeneity would prevent a uniform watermarking process among features, as features with larger magnitudes could dominate others. To address this, we first apply a column-wise YJT defined in Def. 1, and then standardize each transformed column. A crucial property of YJT is that it is monotonic and invertible, allowing for exact recovery of the original data through its inverse. After the YJT followed by standardization, each row becomes a real-valued sequence with unified scale. We then apply a row-wise DFT to obtain the frequency-domain representation $\boldsymbol{y} = \texttt{DFT}(\boldsymbol{x}) \in \mathbb{C}^p$, as defined in Def. 2.

**Step 2: Modification on the imaginary parts of the DFT.** Let $\boldsymbol{x} := (x_0, x_1, \ldots, x_{p-1})$ denote a row of tabular data $\mathbf{X} \in \mathbb{R}^{N \times p}$ after applying the YJT and standardization. Let $\boldsymbol{y} := \texttt{DFT}(\boldsymbol{x}) = (y_0, y_1, \ldots, y_{p-1})$ be its frequency-domain representation obtained by the DFT. Since the DFT satisfies conjugate symmetry, i.e., $y_t = \overline{y_{p-t}}$ for any $t$, it suffices to modify only the first half of the

[1] Throughout this paper, we denote the imaginary unit by $\mathrm{i}$, to avoid confusion with the index $i$.
[2] We also introduce a privacy-enhanced variant to support multi-key scenarios. See Appendix B & G.4 for details.

**Algorithm 1** Watermark embedding of TAB-DRW

1: **Input**: Tabular data $\mathbf{X} \in \mathbb{R}^{N \times p}$, parameters $\gamma \in [0,1]$ and $\delta \in [-1,1]$.
2: **Initial**: Transform $\mathbf{X}$ using YJT and standardization (still denoted as $\mathbf{X}$ for simplicity).
3: **for each** row $\boldsymbol{x}$ in $\mathbf{X}$ **do**
4: Compute $\boldsymbol{y} \leftarrow \texttt{DFT}(\boldsymbol{x})$ and generate pseudorandom bits $\{\zeta_t\}_{t=1}^m$ via Algorithm 3.
5: Modify $\boldsymbol{y}$ according to soft variant (2) to obtain $\boldsymbol{y}^{\mathrm{wm}}$.
6: Recover $\boldsymbol{x}^{\mathrm{wm}} \leftarrow \texttt{IDFT}(\boldsymbol{y}^{\mathrm{wm}})$.
7: **end for**
8: Collect each $\boldsymbol{x}^{\mathrm{wm}}$ to form a matrix $\mathbf{X}^{\mathrm{wm}}$.
9: Apply inverse standardization and inverse YJT to $\mathbf{X}^{\mathrm{wm}}$, round and clip if needed, and release.

entries (i.e. $m = \lfloor \frac{p-1}{2} \rfloor$ entries), as the remaining values can be determined by conjugate symmetry. We call the first $m$ entries effective entries. For each entry $y_t$, we denote its real and imaginary parts by $\Re(y_t)$ and $\Im(y_t)$ respectively (as $y_t \in \mathbb{C}$). For each effective entry, we generate a 0-1 pseudorandom bit $\zeta_t \sim \text{Bernoulli}(0.5)$ and modify $\Im(y_t)$ to align with the corresponding $\zeta_t$. We consider two modification strategies, described below.

**Initial idea: Hard sign flip.** The most natural strategy is to force the sign of $\Im(y_t)$ to match $\zeta_t$. Specifically, for each $t = 1, \dots, m$, we define:

$$y_t^{\mathrm{wm}} = \Re(y_t) + (2\zeta_t - 1)\mathtt{i} \cdot |\Im(y_t)|, \quad \text{and} \quad y_{p-t}^{\mathrm{wm}} = \overline{y_t^{\mathrm{wm}}}. \tag{1}$$

Under this rule, if $\Im(y_t)$ already matches the sign of $\zeta_t$, no change is made and $y_t^{\mathrm{wm}} = y_t$. Otherwise, the sign of $\Im(y_t)$ is flipped so that $\Im(y_t^{\mathrm{wm}}) = -\Im(y_t)$.

**Refinement: Soft variant.** The hard sign flipping may potentially introduce large distortions, degrading data fidelity. As a refinement, we introduce a softer modification controlled by two soft hyperparameters $(\gamma, \delta)$. Specifically, we modify $y_t$ only if $|\Im(y_t)|$ is among the $\gamma$-smallest values in $\{|\Im(y_t)|\}_{t=1}^m$ and the sign of $\Im(y_t)$ differs from $2\zeta_t - 1$. Furthermore, we shrink the imaginary part by a factor $\delta \in [-1,1]$ to further limit the distortion:

$$y_t^{\mathrm{wm}} = \begin{cases} \Re(y_t) - \mathtt{i}\delta \cdot \Im(y_t), & \text{if } \Im(y_t)\cdot(2\zeta_t - 1) < 0 \text{ and } |\Im(y_t)| \le \text{Quantile}_\gamma(\{|\Im(y_t)|\}_{t=1}^m), \\ y_t, & \text{otherwise,} \end{cases} \tag{2}$$

and $y_{p-t}^{\mathrm{wm}} = \overline{y_t^{\mathrm{wm}}}$. When $\gamma = \delta = 1$, the soft variant (2) reduces to the hard sign flip (1); when $\gamma = 0$ and $\delta = -1$, it reduces to no watermarking. In practice, varying $(\gamma, \delta)$ enables flexible control over the trade-off between watermark strength and data fidelity. For example, one can tune $(\gamma, \delta)$ to maximize detectability under a given distortion budget by performing a lightweight grid search over synthetic samples. See Appendix G.2 for detailed computation overhead evaluation.

**Step 3: Inverse steps to return to the original data domain.** In the final step, we apply the inverse DFT to each modified $\boldsymbol{y}^{\mathrm{wm}}$, collect the resulting vectors to form a matrix $\mathbf{X}^{\mathrm{wm}}$, and then apply the inverse standardization followed by the inverse YJT to map it back into the original domain. For discrete features, we round values to the nearest valid entry. For example, a value of 0.4 for the "sex" entry would be rounded to 0 (female), while 0.6 would be rounded to 1 (male). For bounded features, we clip values to stay within the valid range. The full procedure is summarized in Algorithm 1. Appendix G.1 presents an ablation study on the effects of rounding and clipping on watermark detectability. Appendix H shows how TAB-DRW handles low-cardinality categorical variables (e.g., gender) in a conservative, adaptive way that preserves semantic validity.

**Remark 1** (Related work)**.** *Modifying the DFT to embed watermarks has also been explored for diffusion models, such as Tree-Ring Watermarking (Wen et al., 2023). However, this method typically applies deterministic, structured modifications (e.g., zeroing subregions), which are unsuitable for tabular data where each feature has distinct semantic meanings. In contrast,* TAB-DRW *performs fine-grained, row-wise perturbations guided by pseudorandom bits, preserving feature fidelity while enabling robust detection through a rank-based pseudorandom bit generation scheme.*

**Remark 2** (Column selection for watermarking)**.** *Since* TAB-DRW *is a lightweight post-editing watermark, model providers or dataset owners can flexibly choose any subset of columns to watermark. For example, the watermark can be applied only to columns containing sensitive or high-value*

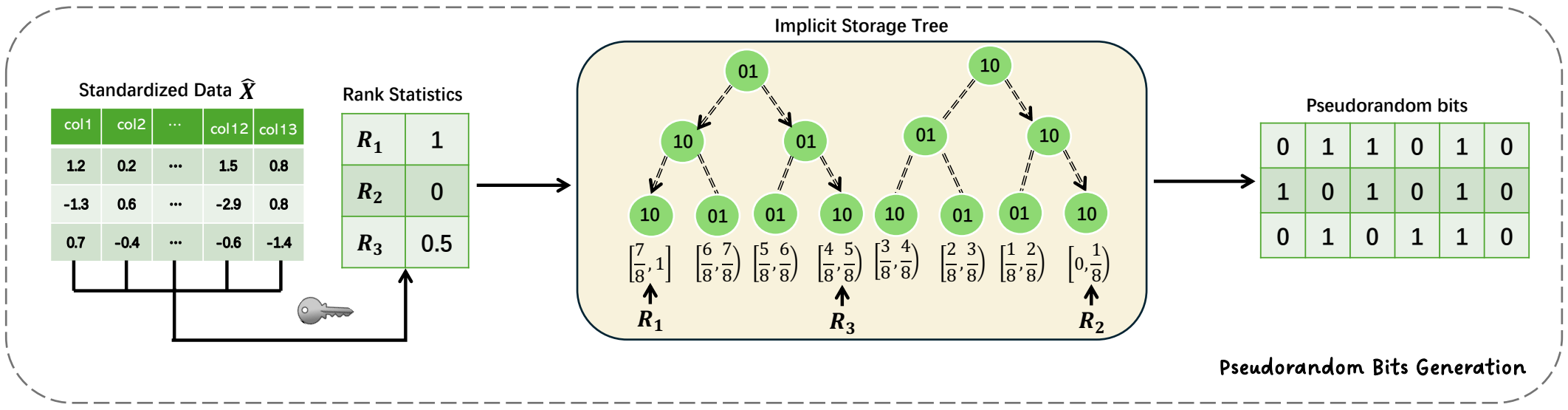

Figure 2: In the proposed pseudorandom bit generation scheme, bit sequence for each row is generated by mapping a row-wise rank statistic to a leaf node in a binary tree.

*information that attackers are unlikely to modify. In Section 4, we do not use any specialized column-selection strategy and exclude only columns with extreme distributions, which contribute little to the watermark signal and can cause scaling issues in a small number of rows even after YJT.*

## 2.2 Watermark Detection

During detection, we can recover the pseudorandom bits using secret keys. Recall that Tab-DRW embeds watermark signals by flipping the imaginary signs in the frequency domain via the alignment with these pseudorandom bits. As a result, watermarked rows are expected to exhibit stronger alignment with the recovered pseudorandom bits. A natural detection strategy is to count the number of aligned entries in the suspect data under investigation: if the alignment is significantly higher than the expected without watermarking, we declare the data watermarked; otherwise, we do not. Statistically speaking, we solve the following hypothesis testing problem (He et al., 2024):

$$H_0 : \text{ The table is not watermarked} \quad \text{vs.} \quad H_1 : \text{ The table is watermarked.}$$

Given a suspect tabular data $\mathbf{X} \in \mathbb{R}^{N\times p}$, we first apply **Step 1** of the watermark embedding procedure to obtain its frequency-domain representation $\mathbf{Y} = \{y_{i,j}\}_{i,j} \in \mathbb{C}^{N\times p}$ , and denote the corresponding pseudorandom bits by $\{\zeta_{i,j}\}_{i,j}$. For each row $i$, we define the alignment count $T_i = \sum_{j=1}^{m} \mathbb{I}\left[\Im(y_{i,j}) \cdot (2\zeta_{i,j} - 1) > 0\right]$. We compute a one-sided Z-score to measure deviation from the expected alignment under $H_0$:

$$Z = \frac{\frac{1}{N}\sum_{i=1}^{N} T_i - \mu_{\text{nwm}}}{\frac{\sigma_{\text{nwm}}}{\sqrt{N}}}, \tag{3}$$

where $\mu_{\text{nwm}}$ and $\sigma_{\text{nwm}}$ denote the mean and standard deviation of $T_i$ under $H_0$. Given a critical value $q_\alpha$ for a significance level $\alpha$, we reject $H_0$ and declare the table watermarked if $Z > q_\alpha$. In practice, we approximate $\mu_{\text{nwm}}, \sigma_{\text{nwm}}$ and $q_\alpha$ using Monte Carlo simulation.

## 2.3 Pseudorandom Bits Generation

The remaining issue is how to construct the pseudorandom bits used for embedding and detection. The design must satisfy two key requirements: 1) **robustness**, ensuring that recovered pseudorandom bits remain stable under post-processing attacks; and 2) **memory efficiency**, avoiding the impractical cost of explicitly storing pseudorandom bits for each generated table.

**Informal description.** To achieve the two goals, we propose a new pseudorandom bit generation scheme with two key components: 1) an implicit storage structure based on a binary tree, and 2) a retrieval mechanism using rank statistics. Specifically, for each row of the standardized tabular data, we use a secret key to select a subset $\mathcal{I}$ of columns, then compute the sum of the entries in $\mathcal{I}$ as a score for that row. This score determines the row's rank among all rows. We then normalize this rank to lie in $[0, 1]$. For example, if a row with $m$ effective entries has rank $r = 1$ among $n = 3$ rows (i.e., the second-largest), its normalized rank is $\frac{r}{n-1} = 0.5$. We partition $[0, 1]$ into $2^{\lceil \frac{m}{2} \rceil}$ equal-sized bins and construct a binary tree of depth $\lceil \frac{m}{2} \rceil$, where each node is deterministically assigned a pseudorandom bit pair and each leaf corresponds to one bin. Each row's rank determines its bin, and the path from the root to the leaf encodes the pseudorandom bit sequence for that row. See Figure 2 for an illustration of the case $m = 6$ and Algorithm 3 in for a formal description. Our pseudorandom bit

generation scheme can be viewed as a variant of a Gray-code encoder; additional details on this connection are provided in Appendix C.

**Robustness of the pseudorandom bits.** Our pseudorandom bit generation is robust against perturbations owing to two mechanisms. First, the subset $\mathcal{I}$ of columns used for score computation is determined by the secret key. An adversary without it cannot targetedly modify the specific entries that contribute to pseudorandom bit generation. Second, the sum-based rank statistic is highly stable, so small perturbations to a subset of columns (even those within $\mathcal{I}$) often do not change the bin to which the row belongs. As a result, the recovered pseudorandom bits typically remain correct. Even when the statistic shifts noticeably, it usually moves only to an adjacent bin. Our node–bit mapping policy ensures that adjacent bins differ by only a single bit pair, which limits the effect of such shifts on the recovered bit sequence. The tree-based structure enables deterministic computation of these mappings without requiring explicit storage.

## 3 Analysis on Distortion and Robustness

This section presents a theoretical analysis of the distortion and robustness properties of our watermarking scheme, offering insight into its foundational guarantees. The analysis is carried out on normalized tabular data after YJT and standardization, where each entry has zero mean and unit variance. This normalization reduces heterogeneity and facilitates theoretical analysis. Although the analysis assumes an idealized setting in the transformed domain without refitting the YJT parameters, all experiments use the full watermark embedding and detection pipeline as shown in Figure 1, consistent with realistic deployment. In Appendix D, we provide an empirical case study and additional evaluations to show that the effect of YJT refitting on both the data distribution and the watermark signal induced by sign-bit alignment is minimal, supporting the soundness of the idealized setting. We also provide further analysis and complete proofs in Appendix E.

### 3.1 Watermark Distortion

Let $\mathbf{X} = \{x_{i,j}\}_{i,j} \in \mathbb{R}^{N\times p}$ be the tabular data where each column is centered and standardized, i.e., $\sum_{i=1}^{N} x_{i,j} = 0$ for all $j$ and $\mathbf{\Sigma} = \frac{1}{N}\mathbf{X}^\top\mathbf{X}$ with $\operatorname{diag}(\mathbf{\Sigma}) = \mathbb{I}_{p\times p}$. Recall that each row vector $\boldsymbol{x}_i := (x_{i,0}, \dots, x_{i,p-1}) \in \mathbb{R}^{1\times p}$ is first mapped to the frequency domain via the DFT, then modified in some of imaginary components, and finally transformed back to the original domain. The following proposition characterizes the resulting entry-wise distortion—that is, the difference between the unwatermarked and watermarked table entries.

**Proposition 1** (Entry-wise differences). *Let $S \subseteq \{1, \dots, m\}$ with $m = \lfloor \frac{p-1}{2} \rfloor$ denote the set of frequency coordinates whose imaginary signs are modified by our watermarking method. Let $\Delta x_{i,j} = x_{i,j}^{\mathrm{wm}} - x_{i,j}$ denote the entry-wise difference. Then*

$$\Delta x_{i,j} = -\alpha\, \boldsymbol{\beta}_j^\top \boldsymbol{x}_i, \quad \alpha = \frac{2(1+\delta)}{p},$$

*where $\boldsymbol{\beta}_j = \big(\beta_S(0,j), \dots, \beta_S(p-1,j)\big)^\top$, and $\beta_S(n,j) = \sum_{k\in S} \sin\left(\frac{2\pi kn}{p}\right) \sin\left(\frac{2\pi kj}{p}\right)$.*

**Theorem 1** (Column-wise differences). *Our watermark affects column-wise quantities as follows:*

1. ***Mean.*** *For each column $j$, the column mean is preserved: $\frac{1}{N}\sum_{i=1}^{N} \Delta x_{i,j} = 0$.*
2. ***Pearson correlation coefficients (PCC).*** *Let $r_{j\ell}$ and $r_{j\ell}^{\mathrm{wm}}$ be the PCCs between columns $j$ and $\ell$ before and after watermarking, respectively. Define $\Delta r_{j\ell} := r_{j\ell}^{\mathrm{wm}} - r_{j\ell}$. Then,*
$$\Delta r_{j\ell} = -\alpha\left([\mathbf{\Sigma}\boldsymbol{\beta}_\ell]_j + [\mathbf{\Sigma}\boldsymbol{\beta}_j]_\ell\right) + \alpha^2 \boldsymbol{\beta}_j^\top \mathbf{\Sigma} \boldsymbol{\beta}_\ell.$$
3. ***Empirical distribution.*** *For each column $j$, let $\rho_j = \frac{1}{N}\sum_{i=1}^{N} \delta_{x_{i,j}}$ denote the empirical distribution of the unwatermarked entries, and let $\rho_j^{\mathrm{wm}} = \frac{1}{N}\sum_{i=1}^{N} \delta_{x_{i,j}^{\mathrm{wm}}}$ denote the corresponding distribution after watermarking. Let $\mathcal{W}_2(\cdot,\cdot)$ denote the Wasserstein-2 distance. Then,*
$$\mathcal{W}_2(\rho_j, \rho_j^{\mathrm{wm}}) \le \alpha\sqrt{\boldsymbol{\beta}_j^\top \mathbf{\Sigma} \boldsymbol{\beta}_j}.$$

The above theorem characterizes column-wise differences and highlights how the parameters $(\gamma, \delta)$ influence them. When $\delta = -1$, no imaginary components are changed, so $\alpha = 0$ and all three

column-wise quantities remain unchanged. Similarly, when $\gamma = 0$, the selected frequency set $S$ is empty, implying $\boldsymbol{\beta}_j = \mathbf{0}$ for all $j$, and again, no distortion occurs. In contrast, when $(\gamma, \delta) \neq (0, -1)$, the bounds quantify the extent of distortion, revealing how the watermark parameters affect the data.

### 3.2 Watermark Robustness

In this section, we analyze the robustness of our watermarking scheme under Gaussian noise. We focus on the Z-score defined in (3). For unwatermarked data with independence assumption, the Z-score converges in distribution to standard normal distribution $\mathcal{N}(0,1)$ (see Lemma 3 in the appendix). In contrast, for the watermarked table, the Z-score is significantly elevated. The following theorem shows that even after corruption with Gaussian noise, the expected Z-score remains high, indicating that our method embeds a strong and resilient watermark signal. In Appendix E.4, we extend the robustness analysis to a broader class of distributions by relaxing the Gaussian assumption in Theorem 2 to a sub-Gaussian setting. This setting accommodates non-Gaussian features with light-tailed distributions, including bounded or discrete categorical features.

**Theorem 2** (Robustness)**.** *We assume that unwatermarked tabular data* $\mathbf{X} \in \mathbb{R}^{N\times p}$ *has rows* $\boldsymbol{x}_i \overset{\text{i.i.d.}}{\sim} \mathcal{N}(0, \Sigma)$*, where* $\Sigma \in \mathbb{R}^{p\times p}$ *is positive-definite. Denote the smallest and largest eigenvalue of* $\Sigma$ *by* $\lambda_{\min}$ *and* $\lambda_{\max}$*, respectively. Define* $Z(\gamma,\delta,\sigma)$ *as the standard Z-score (as in (3)) computed on the Gaussian noise-corrupted table* $\mathbf{X}_{\text{wm}} + \boldsymbol{\varepsilon}$*, where* $\mathbf{X}_{\text{wm}} \in \mathbb{R}^{N\times p}$ *denote the table watermarked under soft hyperparameters* $(\gamma,\delta)$ *and* $\varepsilon_{i,j} \overset{\text{i.i.d.}}{\sim} \mathcal{N}(0,\sigma^2)$*. Then, for any* $\gamma \in [0,1]$ *and* $\delta, \sigma > 0$*,*

$$\mathbb{E}\left[Z(\gamma,\delta,\sigma)\right] \geq \sqrt{mN}\gamma \left[1 - \mathcal{I}(\sigma) - \mathcal{I}\left(\frac{\sigma}{\delta}\right)\right], \tag{4}$$

*where* $m = \lfloor \frac{p-1}{2} \rfloor$ *and the function* $\mathcal{I} : (0,\infty) \to \mathbb{R}$ *is defined as*

$$\mathcal{I}(s) := \frac{s}{\sqrt{s^2+\lambda_{\min}}}\left[\Phi\left(\sqrt{1+\frac{\lambda_{\min}}{s^2}}\right) - \frac{1}{2}\right] + \frac{s}{\sqrt{s^2+\lambda_{\max}}}\left[1 - \Phi\left(\sqrt{1+\frac{\lambda_{\max}}{s^2}}\right)\right] + \frac{1}{\sqrt{8\pi e}}\left[E_1\left(\frac{\lambda_{\min}}{2s^2}\right) - E_1\left(\frac{\lambda_{\max}}{2s^2}\right)\right].$$

*with* $\Phi(\cdot)$ *denoting the standard normal CDF and* $E_1(u) = \int_u^\infty \frac{e^{-t}}{t}\,dt$ *the exponential integral.*

**Remark 3.** *We provide a numerical example to illustrate the guarantee in Theorem 2. When* $(N,p) = (1000, 11)$ *and* $\Sigma = \mathbb{I}_{p\times p}$*, the theoretical lower bound in the right-hand side of (4) is:*

$$\mathbb{E}\left[Z(0.5, 0.5, \sigma)\right] \geq \begin{cases} 30.13, & \text{if } \sigma = 0.1, \\ 14.95, & \text{if } \sigma = 0.5, \\ 7.04, & \text{if } \sigma = 1.0. \end{cases}$$

*Note that for a standard normal Z-score, the 0.99 quantile is 2.32. The above values are substantially larger, implying that the watermark signal remains significant even after noise corruption.*

## 4 Experiments

In this section, we evaluate the performance of our proposed watermarking scheme on five benchmark tabular datasets along three dimensions: 1) data fidelity, 2) watermark detectability, and 3) robustness against post-processing attacks. We also examine how the soft hyperparameters influence watermark strength, highlighting the inherent trade-off between data fidelity and detectability. We refer readers to Appendix G for additional empirical results, including comprehensive ablation studies in Appendix G.1, runtime analysis for watermark embedding and detection in Appendix G.2, extended robustness evaluations under stronger and adaptive attacks in Appendix G.3, and a practical evaluation of the privacy-enhanced Tab-DRW in real-world deployment settings in Appendix G.4.

### 4.1 Experimental Setup

**Datasets.** Experiments are conducted on five real-world tabular datasets with both continuous and discrete feature types: Adult (Becker & Kohavi, 1996), Magic (Bock, 2004), Shoppers (Sakar & Kastro, 2018), Default (Yeh, 2009), and Drybean (Koklu & Özkan, 2020). See details in Appendix F.1.

**Baselines.** We consider four baselines: two post-editing watermarking methods, GLW (He et al., 2024) and TabularMark (Zheng et al., 2024), and two sampling-phase methods, TabWak* with valid bit mechanism (Zhu et al., 2025) and MUSE (Fang et al., 2025). We reproduce TabWak* using the official open-source code and implement other baselines according to the authors' specifications. To ensure fair comparison, We follow Zhu et al. (2025) to synthesize tabular data using TabSyn (Zhang et al., 2024a) with DDIM sampling. Further implementation details are provided in Appendix F.3 & F.4.

Table 2: Data fidelity and watermark detectability. No watermarking is denoted as "W/O". Our proposed TAB-DRW is evaluated with the hyperparameter $(\gamma, \delta) = (0.5, 0.5)$. Best performances are shown in **bold**, and second-best are underlined.

| Datasets | Method | Fidelity Metric | | | | Z-score | | FPR / TPR | |
|---|---|---|---|---|---|---|---|---|---|
| | | **Density ↑** | **Corr ↑** | **C2ST ↑** | **MLE ↑** | **1k rows ↑** | **5k rows ↑** | **1k rows ↑** | **5k rows ↑** |
| Adult | W/O | 0.922±0.001 | 0.872±0.001 | 0.611±0.004 | 0.824±0.005 | – | – | – | – |
| | GLW | 0.912±0.002 | 0.871±0.002 | 0.604±0.015 | 0.821±0.017 | 7.293±0.96 | 16.54±1.05 | 0.00/0.91 | 0.00/1.00 |
| | MUSE | 0.921±0.001 | **0.877±0.003** | 0.599±0.007 | 0.823±0.005 | 6.712±0.89 | 14.81±1.23 | 0.00/0.78 | 0.00/1.00 |
| | TabWak* | 0.912±0.002 | 0.863±0.004 | 0.604±0.009 | 0.793±0.009 | 6.796±1.03 | 15.67±0.97 | 0.00/0.78 | 0.00/1.00 |
| | TabularMark | **0.922±0.001** | 0.872±0.001 | 0.598±0.006 | **0.823±0.003** | 9.674±3.00 | 22.53±3.05 | 0.00/0.89 | 0.00/1.00 |
| | **TAB-DRW** | 0.915±0.005 | 0.864±0.004 | **0.604±0.008** | 0.816±0.009 | **12.81±1.17** | **29.55±1.12** | 0.00/1.00 | 0.00/1.00 |
| Magic | W/O | 0.917±0.001 | 0.945±0.003 | 0.672±0.004 | 0.823±0.007 | – | – | – | – |
| | GLW | 0.915±0.001 | **0.944±0.002** | 0.669±0.013 | 0.816±0.006 | **77.05±0.55** | **172.2±0.51** | 0.00/1.00 | 0.00/1.00 |
| | MUSE | 0.912±0.002 | 0.943±0.006 | 0.672±0.008 | 0.824±0.017 | 15.84±0.86 | 35.31±0.70 | 0.00/1.00 | 0.00/1.00 |
| | TabWak* | 0.912±0.007 | 0.921±0.003 | 0.671±0.006 | **0.827±0.029** | 8.608±0.98 | 19.83±1.01 | 0.00/1.00 | 0.00/1.00 |
| | TabularMark | **0.917±0.001** | 0.943±0.003 | 0.674±0.005 | 0.822±0.009 | 9.666±2.82 | 22.02±3.62 | 0.00/0.90 | 0.00/1.00 |
| | **TAB-DRW** | 0.917±0.005 | 0.937±0.003 | **0.676±0.009** | 0.818±0.014 | 27.34±0.93 | 61.42±1.02 | 0.00/1.00 | 0.00/1.00 |
| Shoppers | W/O | 0.919±0.002 | 0.910±0.001 | 0.704±0.005 | 0.902±0.012 | – | – | – | – |
| | GLW | 0.903±0.001 | **0.908±0.001** | 0.706±0.018 | 0.893±0.009 | 17.84±1.12 | 39.08±1.05 | 0.00/1.00 | 0.00/1.00 |
| | MUSE | 0.911±0.001 | 0.908±0.002 | 0.710±0.009 | 0.895±0.012 | 12.80±0.88 | 28.83±0.87 | 0.00/1.00 | 0.00/1.00 |
| | TabWak* | **0.916±0.009** | 0.906±0.001 | 0.674±0.021 | **0.905±0.047** | 4.071±1.06 | 10.38±1.02 | 0.00/0.04 | 0.00/1.00 |
| | TabularMark | 0.914±0.003 | 0.908±0.001 | 0.704±0.005 | 0.897±0.018 | 10.28±3.24 | 22.94±3.36 | 0.00/0.91 | 0.00/1.00 |
| | **TAB-DRW** | 0.909±0.006 | 0.902±0.003 | **0.712±0.013** | 0.891±0.014 | **18.18±1.28** | **40.74±1.26** | 0.00/1.00 | 0.00/1.00 |
| Default | W/O | 0.930±0.001 | 0.907±0.001 | 0.717±0.003 | 0.797±0.009 | – | – | – | – |
| | GLW | 0.926±0.002 | 0.906±0.003 | 0.710±0.027 | 0.787±0.011 | 12.10±1.09 | 27.08±0.99 | 0.00/1.00 | 0.00/1.00 |
| | MUSE | 0.928±0.001 | 0.907±0.002 | 0.714±0.008 | 0.790±0.007 | 15.50±0.97 | 34.42±0.95 | 0.00/1.00 | 0.00/1.00 |
| | TabWak* | **0.934±0.011** | **0.912±0.014** | **0.723±0.048** | 0.775±0.009 | 10.49±1.03 | 23.60±1.00 | 0.00/1.00 | 0.00/1.00 |
| | TabularMark | 0.927±0.005 | 0.902±0.006 | 0.718±0.007 | **0.796±0.009** | 9.526±2.91 | 23.94±3.18 | 0.00/0.89 | 0.00/1.00 |
| | **TAB-DRW** | 0.929±0.010 | 0.907±0.011 | 0.717±0.018 | 0.791±0.013 | **15.98±0.92** | **35.84±0.91** | 0.00/1.00 | 0.00/1.00 |
| Drybean | W/O | 0.932±0.001 | 0.935±0.001 | 0.640±0.003 | 0.878±0.009 | – | – | – | – |
| | GLW | 0.929±0.002 | 0.933±0.004 | 0.637±0.017 | 0.872±0.013 | **55.14±0.66** | **123.3±0.68** | 0.00/1.00 | 0.00/1.00 |
| | MUSE | 0.930±0.003 | 0.934±0.005 | 0.649±0.011 | 0.878±0.014 | 14.14±1.03 | 31.43±0.91 | 0.00/1.00 | 0.00/1.00 |
| | TabWak* | 0.924±0.014 | 0.925±0.008 | **0.659±0.032** | 0.875±0.015 | 7.999±0.92 | 17.80±0.97 | 0.00/0.99 | 0.00/1.00 |
| | TabularMark | **0.932±0.002** | **0.935±0.001** | 0.641±0.005 | 0.878±0.011 | 7.760±3.14 | 17.28±2.73 | 0.00/0.71 | 0.00/1.00 |
| | **TAB-DRW** | 0.931±0.013 | 0.928±0.007 | 0.655±0.029 | **0.880±0.019** | 38.03±1.03 | 85.05±0.67 | 0.00/1.00 | 0.00/1.00 |

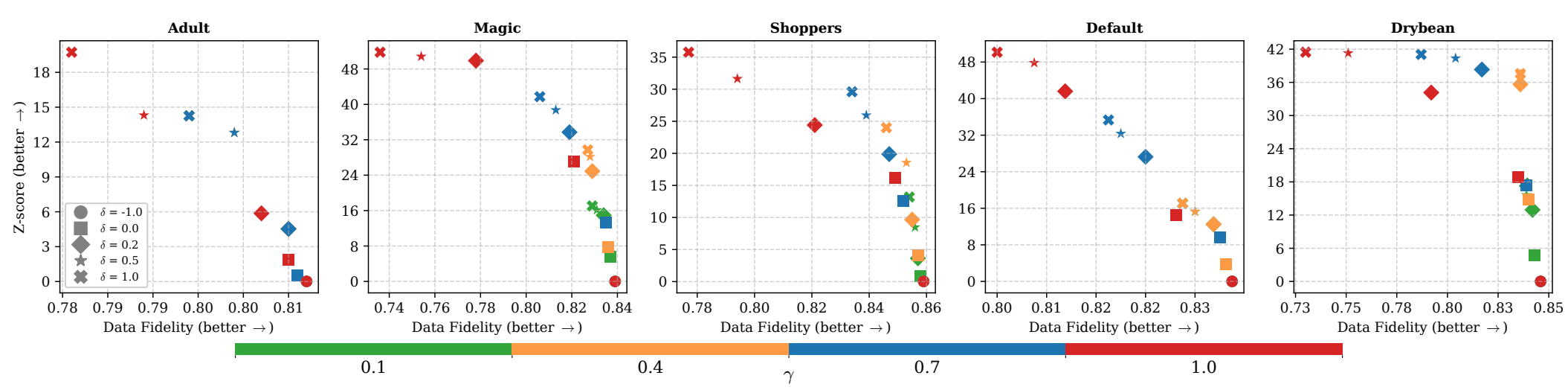

Figure 3: Trade-off between average Z-score on 1K-rows tables and data fidelity under varying $(\gamma, \delta)$.

**Metrics.** We evaluate data fidelity using four metrics introduced in Zhang et al. (2024a): Marginal distribution (**Density**), inter-column correlation (**Corr**), classifier-two-sample-test score (**C2ST**), and machine learning efficiency (**MLE**). For watermark detection, we report two statistical metrics: the one-sided **Z-score** defined in (3), which quantifies the distributional shift of a pivotal statistic induced by watermarking, and **FPR / TPR**, the false and true positive rates under the critical value $q_\alpha = 6$, chosen to better distinguish detection performance. See Appendix F.2 for further details.

## 4.2 Data Fidelity vs. Watermark Detectability

To evaluate data fidelity, we generate synthetic tabular datasets with the same number of rows as the original for each of the five datasets. For watermark detectability, we compute both Z-scores and FPR/TPR using two batch sizes: 1k and 5k rows. Table 2 reports the mean and standard deviation of each fidelity metric over 10 independent trials, and each detectability metric over 100 trials.

On one hand, our watermarking scheme introduces minimal distortion. Across all datasets, its fidelity scores closely match those of the unwatermarked data (degrading by no more than 0.01) and are comparable to existing baselines. While TabWak* and TabularMark often achieve the highest fidelity, our method with $(\gamma, \delta) = (0.5, 0.5)$ performs similarly well. Overall, all baseline methods yield comparable fidelity scores, indicating that our approach preserves data quality on par with prior work. On the other hand, in terms of watermark detectability, our method achieves the highest Z-scores on the Adult, Shoppers, and Default datasets. GLW performs best on Magic and Drybean, likely

Table 3: Watermark robustness against attacks. Average Z-score on 5k rows under ten post-processing attacks. Each value is obtained by repeating the attacks 100 times (10 times for "TabWak*") and averaging the results. Our proposed TAB-DRW is evaluated with the hyperparameter $(\gamma, \delta) = (0.5, 0.5)$. Best performances are shown in **bold**, and second-best are underlined.

| Datasets | Method | Attacks | | | | | | | | | |
|---|---|---|---|---|---|---|---|---|---|---|---|
| | | **Row Del.** | **Col Del.** | **Cell Del.** | **G-Noise** | **C-Noise** | **A-Noise** | **Truncation** | **Quantization** | **Resample** | **Shuffle** |
| Adult | GLW | 15.69 | 14.55 | 14.88 | 0.00 | 16.54 | 7.26 | 16.54 | 8.63 | 16.89 | 16.54 |
| | MUSE | 14.00 | 6.26 | 9.16 | 12.83 | 10.91 | 4.53 | 14.81 | 10.96 | 20.15 | 14.81 |
| | TabWak* | 14.98 | 11.32 | 11.08 | 0.91 | 15.67 | 14.50 | 5.09 | 11.27 | 16.37 | 15.67 |
| | TabularMark | 21.65 | 15.56 | 16.42 | 2.83 | 6.90 | 1.29 | 2.72 | 0.00 | 4.62 | 17.44 |
| | **TAB-DRW** | **27.98** | **17.78** | **20.46** | **20.36** | **24.59** | **23.72** | **29.55** | **20.95** | **28.15** | **29.55** |
| Magic | GLW | **163.33** | **140.15** | **154.89** | 0.03 | **172.20** | 1.09 | 14.71 | **47.28** | **170.81** | **172.20** |
| | MUSE | 33.55 | 7.16 | 17.91 | 15.99 | 34.33 | 9.09 | 20.06 | 14.54 | 33.59 | 35.31 |
| | TabWak* | 18.92 | 13.18 | 16.33 | 17.99 | 19.76 | 13.86 | 16.87 | 16.50 | 17.62 | 19.76 |
| | TabularMark | 20.65 | 12.05 | 13.80 | 0.00 | 20.05 | 0.35 | 13.66 | 0.65 | 21.04 | 15.26 |
| | **TAB-DRW** | 58.28 | 17.45 | 35.78 | **46.18** | 54.48 | **40.72** | **52.62** | 45.14 | 37.61 | 61.42 |
| Shoppers | GLW | 36.82 | **36.02** | **36.15** | 0.00 | **39.08** | 1.11 | 24.61 | 7.20 | **31.92** | 39.08 |
| | MUSE | 27.34 | 11.37 | 15.34 | 23.56 | 21.58 | 16.74 | 23.14 | 19.84 | 28.63 | 28.83 |
| | TabWak* | 9.55 | 3.33 | 4.68 | 0.02 | 10.47 | 5.08 | 9.05 | 7.82 | 10.71 | 10.47 |
| | TabularMark | 15.48 | 11.17 | 14.66 | 1.46 | 18.43 | 0.13 | 11.40 | 3.02 | 18.23 | 18.62 |
| | **TAB-DRW** | **38.43** | 19.35 | 22.99 | **39.66** | 36.26 | **20.66** | **30.28** | **32.46** | 29.28 | **40.74** |
| Default | GLW | 25.67 | 24.59 | **23.88** | 0.00 | 27.08 | 9.08 | 27.08 | 11.94 | 15.82 | 27.08 |
| | MUSE | 32.75 | 11.38 | 13.11 | 19.81 | 24.25 | 4.98 | 34.42 | 11.17 | **36.60** | 34.42 |
| | TabWak* | 22.91 | 18.36 | 18.96 | 22.95 | 23.70 | **21.79** | 22.80 | 19.83 | 31.49 | 23.70 |
| | TabularMark | 21.66 | 15.79 | 12.46 | 0.00 | 21.53 | 0.23 | 12.36 | 0.86 | 20.94 | 23.33 |
| | **TAB-DRW** | **33.92** | **25.03** | 22.56 | **30.03** | **32.22** | 21.55 | **35.84** | **21.93** | 32.36 | **35.84** |
| Drybean | GLW | **116.96** | **112.05** | **110.89** | 0.00 | **123.28** | 13.29 | 28.02 | 29.37 | **123.68** | **123.28** |
| | MUSE | 29.78 | 7.76 | 12.32 | 6.22 | 29.56 | 6.28 | 10.43 | 1.89 | 31.19 | 31.43 |
| | TabWak* | 17.16 | 0.00 | 2.86 | 14.11 | 17.53 | 13.59 | 16.80 | 6.56 | 15.38 | 17.53 |
| | TabularMark | 13.79 | 7.18 | 9.55 | 0.00 | 13.56 | 0.00 | 7.57 | 0.66 | 12.88 | 10.46 |
| | **TAB-DRW** | 80.62 | 42.82 | 50.99 | **31.12** | 80.43 | **58.50** | **42.14** | **61.23** | 68.69 | 85.05 |

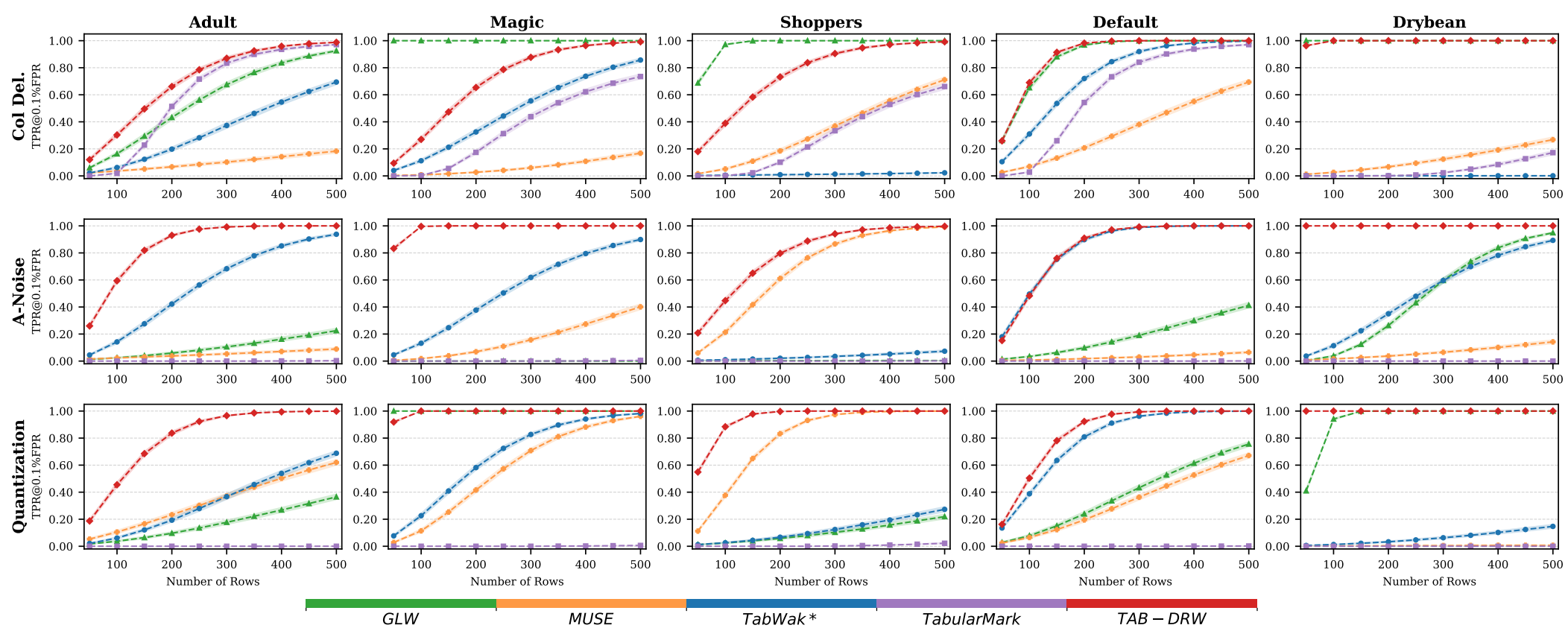

Figure 4: TPR@0.1%FPR versus row count under three representative attacks. Dashed lines show the bootstrap mean estimate (500 resamples), and shaded regions indicate the 90% confidence interval.

due to the predominance of continuous attributes. Most methods—including ours—successfully control false positive rates and show nontrivial true positive rates under the critical threshold $q_\alpha = 6$, demonstrating reliable detectability. The results in Figure 3 also reveal a trade-off between data fidelity and watermark strength for our method. Increasing both $\gamma$ and $\delta$ enhances the Z-score, but also increases distortion. This trade-off is inherent to post-editing watermarking: stronger signals inevitably introduce more distortion. Due to space limit, similar additional results on TabSyn with score-based diffusion process and two other tabular generators, together with an ablation study on YJT, are presented in Appendix G.1.

### 4.3 ROBUSTNESS AGAINST POST-PROCESSING ATTACKS

We next evaluate the robustness of watermarking methods against ten post-processing attacks, which can be grouped into four categories: 1) deletion attacks, which remove or replace information at different granularities (**Row Del.**, **Col Del.**, **Cell Del.**); 2) noise attacks, which perturb numerical or categorical values with Gaussian, categorical, or adaptive noise (**G-Noise**, **C-Noise**, **A-Noise**); 3) discretization attacks, which reduce numeric precision through truncation or quantization (**Truncation**, **Quantization**); and 4) structural attacks, which alter table structure by resampling label distribu-

Table 4: Robustness of TAB-DRW against adaptive row deletion and rewatermarking attacks of varying strength. Z-scores are computed on tables with 5k rows and averaged over 100 independent trials. "Rewatermarking@$n$" denotes rewatermarking the table using $n$ randomly sampled keys.

| Datasets | No-attack | Adv. Row Del. | | | Rewatermarking | | |
|---|---|---|---|---|---|---|---|
| | | @0.1 | @0.2 | @0.5 | @1 | @3 | @10 |
| Adult | $29.12 \pm 1.12$ | $28.55 \pm 1.44$ | $26.35 \pm 2.61$ | $18.79 \pm 4.47$ | $23.66 \pm 1.17$ | $16.26 \pm 1.09$ | $17.26 \pm 1.34$ |
| Magic | $61.42 \pm 1.02$ | $56.71 \pm 2.98$ | $49.36 \pm 5.77$ | $28.41 \pm 7.28$ | $53.23 \pm 0.91$ | $34.32 \pm 0.93$ | $29.17 \pm 1.00$ |
| Shoppers | $40.74 \pm 1.26$ | $36.47 \pm 2.62$ | $30.40 \pm 3.71$ | $17.12 \pm 4.18$ | $31.97 \pm 1.15$ | $20.14 \pm 1.09$ | $16.67 \pm 1.09$ |
| Default | $35.84 \pm 0.91$ | $31.91 \pm 1.90$ | $27.13 \pm 3.23$ | $15.36 \pm 4.54$ | $32.85 \pm 1.00$ | $19.40 \pm 1.07$ | $26.28 \pm 1.18$ |
| Drybean | $85.05 \pm 0.67$ | $79.27 \pm 2.67$ | $71.67 \pm 5.04$ | $52.39 \pm 9.99$ | $44.79 \pm 0.81$ | $29.47 \pm 0.83$ | $33.77 \pm 0.95$ |

tions or randomly shuffling rows (**Resample**, **Shuffle**). Due to space limit, detailed definition and implementations of the attacks are provided in Appendix F.5, and additional results of stronger attack intensities are reported in Appendix G.3.

Table 3 reports the average one-sided $Z$-score over 5k rows. Our watermarking method consistently demonstrates superior robustness across all attack types and datasets, ranking either first or second. In contrast, GLW and TabularMark remain resilient to deletion and structural attacks but often fail under noise and discretization. TabWak* and MUSE exhibit some robustness to noise and discretization on certain datasets, but they are vulnerable to deletion attacks due to their reliance on complete column information. Figure 4 shows TPR@0.1%FPR versus the number of rows under three representative and strong attacks. Among ten out of fifteen cases, our method reaches 1.0 TPR@0.1%FPR using only 300 rows, with the remaining five cases requiring fewer than 500 rows. In contrast, baseline methods often suffer reduced true positive rates or completely lose detectability under these conditions.

### 4.4 Robustness against Adaptive Attacks

In this section, we implement two adaptive attacks targeting TAB-DRW. In both cases, we assume adversaries fully understand our pipeline, including the privacy-enhanced version (see Appendix B) used for real-world deployment, but do not know the secret key. The first attack, Adaptive Row Deletion, corrupts the row ranking to impair rank-based bit retrieval. The attacker samples a random key, computes normalized row ranks (following lines 3–6 of Algorithm 3), and deletes a contiguous block of rows in rank space. For example, with strength 0.1, the adversary removes rows whose normalized ranks lie in a random interval of length 0.1 in $[0, 1]$, which disrupts ranking far more than random deletion. The second attack, Rewatermarking, aims to erase sign-bit alignment in the frequency domain. It leverages two properties of our privacy-enhanced TAB-DRW: strong fidelity preservation and the fact that a watermark embedded with one key is not detectable by another. An informed attacker can repeatedly rewatermark the table with different keys to perturb the original alignment and make detection under the original key fail.

The results in Table 4 show that TAB-DRW remains highly detectable even under substantial adaptive row-deletion attacks. Although detectability decreases slightly relative to random row deletion, the use of a secret key and stable tree-based bit storage makes TAB-DRW resilient to attacks specifically designed to disrupt the row-ranking process. We also observe that TAB-DRW remains highly detectable even after ten rounds of rewatermarking, by which point tabular fidelity has already degraded noticeably. These findings demonstrate that, without knowledge of the key used in Algorithm 2 and 3, an attacker—despite understanding the TAB-DRW pipeline—cannot substantially disrupt sign-bit alignment while preserving data fidelity.

## 5 Conclusion

In this paper, we present TAB-DRW, a lightweight and robust post-editing watermarking scheme for tabular data. TAB-DRW normalizes heterogeneous features via the Yeo–Johnson transformation and standardization, and embeds watermarks by adjusting the imaginary parts of adaptively selected frequency-domain entries. It achieves strong detectability and high fidelity across mixed-type datasets—without relying on large diffusion models or explicitly storing unwatermarked data. Our proposed rank-based pseudorandom bit generation method enables efficient row-wise retrieval via robust rank statistics, further enhancing resilience to post-processing attacks. We provide theoretical analysis of watermark distortion and robustness against noise perturbations; and validate our approach on five benchmark datasets, demonstrating broad applicability, high fidelity, strong detectability, and great robustness. We believe that TAB-DRW offers a solid foundation for advancing secure data sharing and the development of privacy-preserving generative AI.

## Ethics Statement

The research presented in this work fully conforms to the ICLR Code of Ethics. This paper does not involve crowdsourcing or research with human subjects. The content poses no safety risks or negative societal impacts, including potential malicious or unintended uses (e.g., disinformation, fake profiles, surveillance), fairness concerns (e.g., technologies that could disproportionately affect specific groups), privacy risks, or security vulnerabilities. On the contrary, we highlight positive societal impacts, such as facilitating the responsible use of synthetic tabular data, in Section 1, Section 5, and Appendix I.

## Reproducibility Statement

We ensure full reproducibility by disclosing all implementation details of our work. The proposed method is described in Section 2, with watermark embedding detailed in Algorithm 1 and pseudorandom bit generation in Algorithm 3. For watermark detection, the hypothesis test and test statistic are also specified in Section 2. Theoretical assumptions, along with further justification and proofs, appear in Section 3, Appendix D, and Appendix E. Dataset details are provided in Appendix F.1, while evaluation metrics are given in Appendix F.2. Generator configurations, baselines, and attack implementations are documented in Appendices F.3–F.5. Supplementary materials include source code and configuration files for reproducing all main results, together with running instructions.

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

# CONTENTS

# A Related Works

**Watermarking LLM-generated text.** Watermarking in large language models (LLMs) can be broadly categorized into unbiased and biased approaches, both aiming to embed detectable signals without substantially degrading text quality. Unbiased watermarks preserve the original next-token distribution exactly. Aaronson (2023) draws independent pseudorandom variables and samples next token using a deterministic decoder, preserving the multinomial distribution via the Gumbel-max trick. Similarly, Kuditipudi et al. (2024) generates the next token based on inverse transform sampling of the multinomial distribution. Optimal detection rules for these two unbiased LLM watermarks are derived under the statistical framework (Li et al., 2025). In contrast, biased watermarks perturb the token distribution to embed watermark signals. The KGW watermarking scheme (Kirchenbauer et al., 2023) randomly partitions the token vocabulary into green and red lists and then increases the sampling probability of green tokens to create detectable deviations in green-token frequency. Subsequent works have focused on improving robustness (Zhao et al., 2024) and optimizing the trade-off between detectability and text quality (Wouters, 2024; Huang & Wan, 2025). Additionally, Xie et al. (2025) and Wu et al. (2024) proposed unbiased variants of the KGW watermark by introducing decoding algorithm based on maximal coupling and reweighting strategy, respectively. However, due to their reliance on the order of tokens, these text watermarking methods can not be directly applied to structural tabular data, where attacks like row reordering are so common.

**Watermarking generated images.** Image watermarking methods embed invisible signals either during training or sampling phase. Lukas & Kerschbaum (2023) proposes a method to watermark pre-trained GANs without access to training data. Wen et al. (2023) exploits DDIM invertibility to embed structural patterns into the frequency domain of the initial noise vector during sampling, achieving an effective and invisible image watermarking. Yang et al. (2024) implements a performance-preserving watermarking for diffusion models by incorporating diffused bit information into Gaussian noise in the latent space. Zhu et al. (2025) has explored generalizing image watermarking techniques to structural tabular data. However, these methods still either fail to support row-wise detection or suffer from poor data fidelity and limited robustness.

**Watermarking synthetic tabular data.** Tabular watermarking methods primarily fall into two categories: sampling-phase watermarking and post-editing watermarking. The former embeds watermark into the latent space during the denoising generation phase or modifies the generative workflow. For instance, Zhu et al. (2025) implements row-wise watermark embedding using self-cloning and seeded shuffling techniques, ensuring close approximation to the standard Gaussian distribution. However, due to its reliance on large diffusion models using DDIM sampling strategy, this method is unsuitable for scenarios with limited GPU resources. Fang et al. (2025) proposes MUSE, a model-agnostic method that selects watermarked rows via a pseudorandom scoring mechanism across multiple candidates, preserving fidelity but increasing generation cost. The latter offers lightweight watermarking by modifying synthesized tabular data after generation. He et al. (2024) bins continuous feature values into predefined 'green' intervals and Zheng et al. (2024) extends post-editing watermarking to tabular datasets with mixed-type features by selectively perturbing cells within a designated value range. While model-agnostic and computationally efficient, these approaches are either restricted by the feature type or vulnerable to noise and deletion attacks. The limitations of existing tabular watermarking methods highlight opportunities for improvement in four key dimensions: fidelity, detectability, applicability, and robustness. In this work, our proposed TAB-DRW addresses these limitations, achieving superior performance across all four dimensions.

# B Privacy-Enhanced Tab-Drw

Inspired by the KGW watermark (Kirchenbauer et al., 2023), we extend TAB-DRW with a privacy-enhanced variant designed to increase the difficulty of watermark removal. The modification is straightforward: prior to **Step 1** of watermark embedding, the columns of the tabular data are shuffled according to a pseudorandom permutation determined by the watermark key $\kappa$ (which can be shared with the key in Algorithm 3), and after **Step 3**, the columns are reshuffled back to their original order. During detection, a verifier holding the correct key can reproduce the same column permutation to obtain the watermarked frequency-domain representation. See Figure 5 and Algorithm 2 for the complete procedure of the privacy-enhanced TAB-DRW.

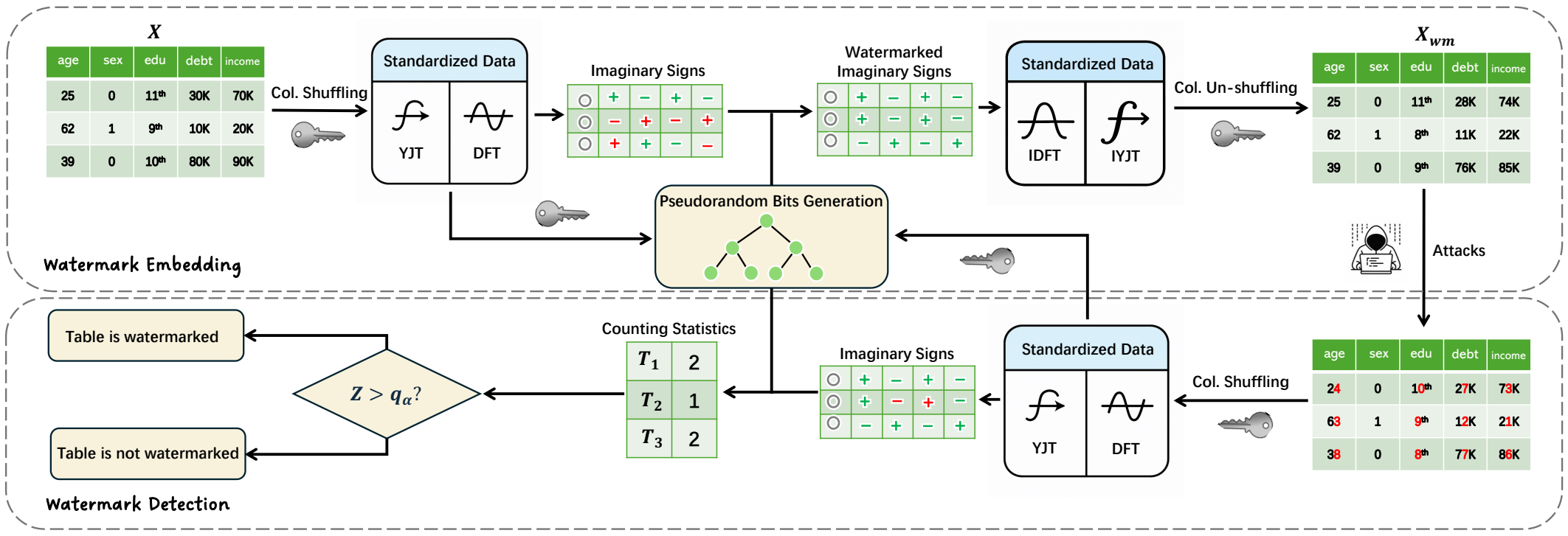

Figure 5: Work flow of privacy-enhanced TAB-DRW.

---

**Algorithm 2** Watermarking embedding of privacy-enhanced TAB-DRW

---

**Input**: Tabular data $\mathbf{X} \in \mathbb{R}^{N \times p}$, parameters $\gamma \in [0, 1]$ and $\delta \in [-1, 1]$, watermark key $\kappa$.
Shuffle $\mathbf{X}$ using key-derived permutation $P_\kappa$ to obtain $\mathbf{X}P_\kappa$.
Transform $\mathbf{X}P_\kappa$ using YJT and standardization (still denoted as $\mathbf{X}P_\kappa$ for simplicity).
**for** each row $\boldsymbol{x}$ in $\mathbf{X}P_\kappa$ **do**
  Compute $\boldsymbol{y} \leftarrow \texttt{DFT}(\boldsymbol{x})$ and generate pseudorandom bits $\{\zeta_t\}_{t=1}^m$ via Algorithm 3.
  Modify $\boldsymbol{y}$ according to soft variant (2) to obtain $\boldsymbol{y}^{\text{wm}}$.
  Recover $\boldsymbol{x}^{\text{wm}} \leftarrow \texttt{IDFT}(\boldsymbol{y}^{\text{wm}})$.
**end for**
Collect each $\boldsymbol{x}^{\text{wm}}$ to form a matrix $\mathbf{X}^{\text{wm}}$.
Apply inverse standardization and inverse YJT to $\mathbf{X}^{\text{wm}}$, then unshuffle it to obtain $\mathbf{X}^{\text{wm}}P_\kappa^{-1}$.
Perform rounding and clipping if needed, and release.

---

Privacy-enhanced TAB-DRW satisfies two crucial requirements:

1. **The key-dependent variability in the frequency-domain representation does not substantially affect watermark distortion or detectability**. In other words, given randomly selected watermark keys, the privacy-enhanced TAB-DRW should exhibit nearly consistent performance in terms of data fidelity and watermark detectability. From a theoretical perspective, since $P_\kappa$ is an orthogonal permutation matrix and the DFT/IDFT are unitary, inserting $P_\kappa$ before and $P_\kappa^{-1}$ after the frequency-domain transformation amounts to a norm-preserving change of entries in the original domain, meaning the total $\ell_2$ distortion remains unchanged. Moreover, because detection evaluates sign alignment in the same keyed frequency coordinates, the resulting $Z$-score is invariant up to index relabeling. Empirical evidence supporting this claim is provided in Table 22 of Appendix G.4.
2. **The approach supports multi-key scenarios, as a watermark embedded with one key cannot be detected using another, thereby effectively avoiding false positives.** Furthermore, the collision-free key space must be sufficiently large to support large-scale deployment. The motivation behind this design lies in the sensitivity of the row-wise DFT to column order: frequency-domain representations derived from different pseudorandom permutations are nearly independent and exhibit nontrivial discrepancies. As a result, the watermark key is effectively encoded into the frequency-domain watermark signal as a unique, secret pattern. Furthermore, this sensitivity induces a combinatorially large key space of size $\mathcal{O}(p!)$ for a tabular dataset with $p$ columns. Comprehensive empirical evaluations are presented in Table 23 of Appendix G.4.

## C MISSING DETAILS FROM SECTION 2

Algorithm 3 details the steps to generate (during watermark embedding) or retrieve (during watermark detection) pseudorandom bit sequence for a target row $\boldsymbol{x}^* \in \mathbb{R}^{1 \times p}$ within standardized tabular data $\mathbf{X} \in \mathbb{R}^{N \times p}$. We first sample a subset of column indices $\mathcal{I}$ using the secret watermark key (Line 3).

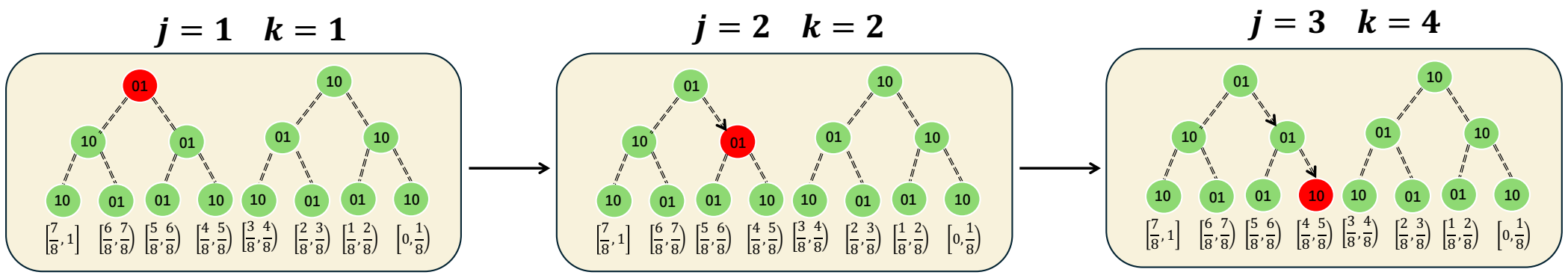

Figure 6: Illustration of Lines 7–9 in Algorithm 3 for the case $x^*_{\text{rank}} = 0.5$ and $m = 6$. The red circle highlights the $k$-th node at level $j$.

For each row, we compute the sum of entries in $\mathcal{I}$, based on which we obtain $x^*_{\text{rank}}$, the rank of the target row among all rows in $\mathbf{X}$. Then we normalize $x^*_{\text{rank}}$ to $[0, 1]$ (Lines 4–6).

Next, We traverse an implicitly constructed binary tree of depth $\lceil \frac{m}{2} \rceil$, where each node deterministically binds with a bit pair, from the root down to a leaf. At each level $j$, we use $x^*_{\text{rank}}$ to locate the underlying node and append its bit pair to the list $\mathbf{S}$ (Lines 7–9, see Figure 6 for an illustration). Lines 8–9 define both the node-bit binding policy and the coupling rule between each of the $2^{\lceil \frac{m}{2} \rceil}$ equal-sized bins and the corresponding leaf nodes. Therefore, the above procedure is equivalent to traversing the path from the root to the leaf corresponding to the bin containing $x^*_{\text{rank}}$, as described in Section 2.3. Finally, we truncate $\mathbf{S}$ to its first $m$ entries to obtain the pseudorandom bit sequence for the target row. In practice, even if $x^*_{\text{rank}}$ shifts to a neighboring leaf node due to post-processing attacks, the retrieved bit sequence differs from the original by only a bit pair, thanks to the tailored node-bits binding policy.

**Algorithm 3** Row-wise Pseudorandom Bits Generation

1: **Input**: Standardized tabular data $\mathbf{X} \in \mathbb{R}^{N \times p}$, target row $\boldsymbol{x}^* \in \mathbb{R}^{1 \times p}$, and watermark key $\kappa$.
2: **Initial**: An empty pseudorandom bit list $\mathbf{S}$, $m = \lfloor (p-1)/2 \rfloor$.
3: Sample a subset of column indices $\mathcal{I} \subset \{0, \ldots, p-1\}$ using $\kappa$.
4: Compute the sum of selected entries for each row of $\mathbf{X}$.
5: Compute the rank of the target row among all rows in $\mathbf{X}$ to obtain $x^*_{\text{rank}}$.
6: Normalize $x^*_{\text{rank}}$ to lie in $[0, 1]$: $x^*_{\text{rank}} \leftarrow x^*_{\text{rank}}/(N-1)$.
7: **for** $j \leftarrow 1$ **to** $\lceil m/2 \rceil$ **do** ▷ Traverse the path from the root to the leaf.
8: Locate the underlying node in the path: $k \leftarrow \lfloor 2^j \cdot x^*_{\text{rank}} \rfloor$.
9: Append $[1, 0]$ to $\mathbf{S}$ if $k\%4 = 0$ or 3; else $[0, 1]$.
10: **end for**
11: Truncate $\mathbf{S}$ to its first $m$ entries, and release.

**Connection to Gray codes.** From the perspective of Gray codes, our pseudorandom bit generation scheme can be viewed as a special case of a 2-Gray code. We present the formal construction process below.

Let $n \in \mathbb{N}$. Denote by $\{0,1\}^n$ the set of all binary strings of length $n$, and by $d_H(\cdot, \cdot)$ the Hamming distance on $\{0,1\}^n$. Let $G_n = (g_0^n, g_1^n, \ldots, g_{2^n-1}^n)$ with $g_i^n \in \{0,1\}^n$ be a standard $n$-bit 1-Gray code, specified up to cyclic permutation. By definition, $d_H(g_i^n, g_{i+1}^n) = 1$ for all $i \in \{0, \ldots, 2^n - 1\}$. We define the pair-encoding map $\varphi : \{0,1\} \to \{0,1\}^2, \varphi(0) = 10, \varphi(1) = 01$ and extend $\varphi$ to a map $\phi : \{0,1\}^n \to \{0,1\}^{2n}$ by applying it coordinatewise: for $g^n = b_1 b_2 \cdots b_n \in \{0,1\}^n$, $b_j \in \{0,1\}$, let $\phi(g^n) = \varphi(b_1)\,\varphi(b_2) \cdots \varphi(b_n)$. For $n \in \mathbb{N}$, we then can define the set $H_{2n} = (h_0^{2n}, h_1^{2n}, \ldots, h_{2^n-1}^{2n})$ and $H_{2n-1} = (\pi(h_0^{2n}), \pi(h_1^{2n}), \ldots, \pi(h_{2^n-1}^{2n}))$, where $h_i^{2n} := \phi(g_i^n) \in \{0,1\}^{2n}$ and $\pi : \{0,1\}^{2n} \to \{0,1\}^{2n-1}$ be the projection that deletes the last coordinate: $\pi(x_1, \ldots, x_{2n}) = (x_1, \ldots, x_{2n-1})$.

By construction, the family $\{H_n\}$ forms a collection of 2-Gray codes, and any two consecutive codewords $h_i^n$ and $h_{i+1}^n$ satisfy $d_H(h_i^n, h_{i+1}^n) \leq 2$. In our paper, a tree of depth $N$ stores the sequence $\{H_n\}_{n=1}^{2N}$, and each rank bin corresponds to a distinct $h_i^{2N}$ or $\pi(h_i^{2N})$. Compared with

using a standard 1-Gray code, this construction has the advantage of slowing the growth rate of rank bins as the number of table columns increases, since $|H_n| = 2^{\lceil n/2 \rceil}$ whereas $|G_n| = 2^n$. This reduction leads to greater robustness of bit retrieval against rank shifts. In Appendix G.1, we provide additional evaluations on using a 1-Gray code for bit sequence generation .

## D MISSING DETAILS FROM SECTION 3

**Soundness of robustness analysis in the transformed domain.** Our theoretical analysis is conducted entirely in the transformed domain, where both the original data $\mathbf{X}$ and the watermarked data $\mathbf{X}_{\text{wm}}$ have already been processed using the Yeo-Johnson transformation (YJT) and standardization. This setup deliberately omits the inverse YJT during watermark embedding and avoids re-estimating transformation parameters during watermark detection, thus ignoring the distribution shifts of watermarked frequency-domain representation caused by the parameters refitting.

Introducing an adaptive, nonlinear YJT transformation, where the parameter $\lambda$ is estimated via maximum likelihood, would prevent us from obtaining a closed-form lower bound on the Z-score and would not offer additional insight into the core mechanism driving robustness. under the Gaussian assumption used in the analysis, the strong fidelity preservation of TAB-DRW keeps the watermarked data very close to the original Gaussian distribution. Consequently, the effect of YJT refitting on both the data distribution and the watermark signal induced by sign-bit alignment is minimal. This makes the idealized model appropriate for the theoretical robustness study, and it does not alter the conclusions we derive.

To validate our point above, we provide a case study to illustrate the minimal difference between the original YJT parameters and those refitted after watermark embedding. Specifically, we generate multivariate Gaussian tables with row counts $N \in \{100, 1000, 10000\}$ and column counts $p \in \{10, 50, 100\}$. Two covariance structures are considered: the identity matrix and an AR(1) matrix $\Sigma_{ij} = \rho^{|i-j|}$ with $\rho = 0.4$. Each table is watermarked using TAB-DRW with $(\gamma, \delta) = (0.5, 0.5)$. The YJT parameters $\lambda$, mean $\mu$, and standard deviation $\sigma$ are recorded before and after watermarking and averaged across all columns. The results show that TAB-DRW introduces negligible perturbations to these parameters across different dimensions and covariance structures.

Table 5: YJT parameters summary (before vs. after watermarking) across varying dimensions and covariance structures.

| $\boldsymbol{\Sigma}$ | N | p | $\lambda$ Before | $\lambda$ After | $\mu$ Before | $\mu$ After | $\sigma$ Before | $\sigma$ After |
|---|---|---|---|---|---|---|---|---|
| Identity | 100 | 10 | 1.0164 | 1.0117 | -0.0230 | -0.0229 | 0.9792 | 0.9592 |
| Identity | 100 | 50 | 1.0065 | 1.0030 | -0.0017 | -0.0027 | 0.9957 | 0.9835 |
| Identity | 100 | 100 | 0.9743 | 0.9807 | -0.0190 | -0.0166 | 0.9917 | 0.9817 |
| Identity | 1000 | 10 | 1.0113 | 1.0113 | 0.0418 | 0.0420 | 1.0002 | 0.9883 |
| Identity | 1000 | 50 | 0.9971 | 0.9940 | -0.0070 | -0.0081 | 1.0020 | 0.9939 |
| Identity | 1000 | 100 | 0.9885 | 0.9861 | -0.0056 | -0.0064 | 1.0014 | 0.9943 |
| Identity | 10000 | 10 | 0.9970 | 0.9999 | 0.0017 | 0.0026 | 0.9988 | 0.9889 |
| Identity | 10000 | 50 | 1.0029 | 1.0027 | 0.0000 | -0.0001 | 1.0000 | 0.9925 |
| Identity | 10000 | 100 | 1.0003 | 1.0011 | 0.0017 | 0.0020 | 0.9997 | 0.9930 |
| AR(1) | 100 | 10 | 0.9849 | 0.9735 | 0.0075 | 0.0028 | 1.0367 | 1.0231 |
| AR(1) | 100 | 50 | 1.0158 | 1.0128 | 0.0072 | 0.0064 | 0.9766 | 0.9680 |
| AR(1) | 100 | 100 | 0.9917 | 0.9921 | -0.0222 | -0.0223 | 0.9811 | 0.9724 |
| AR(1) | 1000 | 10 | 0.9866 | 0.9892 | -0.0024 | -0.0015 | 0.9963 | 0.9873 |
| AR(1) | 1000 | 50 | 0.9926 | 0.9911 | -0.0046 | -0.0051 | 0.9988 | 0.9926 |
| AR(1) | 1000 | 100 | 0.9972 | 0.9967 | 0.0010 | 0.0009 | 0.9954 | 0.9898 |
| AR(1) | 10000 | 10 | 1.0037 | 1.0051 | 0.0008 | 0.0013 | 0.9958 | 0.9873 |
| AR(1) | 10000 | 50 | 1.0023 | 1.0032 | -0.0051 | -0.0048 | 0.9998 | 0.9938 |
| AR(1) | 10000 | 100 | 0.9998 | 0.9998 | -0.0001 | -0.0000 | 0.9999 | 0.9946 |

We also provide additional empirical evaluations by comparing detection performance under 1) **idealized setting:** No parameter refitting, as assumed in Section 3, and 2) **practical setting:** With parameter refitting, as used in experiments. As shown in Table 6, the impact of distribution shifts on

detection performance is negligible relative to post-processing attacks, and the embedded watermark remains highly detectable even under such shifts. These results demonstrate both the robustness of our approach and the validity of conducting robustness analysis in the transformed domain.

Table 6: Detection performance under different parameter-refitting settings. Each entry reports the average Z-score over 1K rows, evaluated using TAB-DRW with $(\gamma, \delta) = (0.5, 0.5)$ under 100 trials.

| Dataset | Idealized setting | Practical setting |
|---|---|---|
| Adult | 15.13±1.04 | 12.81±1.17 |
| Magic | 30.47±0.85 | 27.34±0.93 |
| Shoppers | 21.14±0.84 | 18.18±1.28 |
| Default | 19.02±0.85 | 15.98±0.92 |
| Drybean | 41.36±0.98 | 38.03±1.03 |

**Soundness of robustness analysis under Gaussian assumption.** Real-world tabular data are highly heterogeneous and rarely follow a strict multivariate Gaussian distribution. However, after applying the Yeo–Johnson transformation (YJT), the data typically become much closer to Gaussian. As discussed earlier, YJT standardizes heterogeneous feature scales and reduces skewness, enabling more tractable analysis in the transformed space.

Although the Gaussian assumption does not fully capture the complexity of real-world data, deriving closed-form robustness guarantees under arbitrary, non-Gaussian distributions is generally intractable. Our aim is not to provide universal theoretical guarantees, but to clarify the underlying robustness mechanism of our method. In particular, we show how sign alignment in the frequency domain, together with the hyperparameters $(\gamma, \delta)$, preserves the watermark signal under perturbations. Therefore, we adopt the multivariate Gaussian model as a simplified yet widely used analytical tool to make this intuition concrete. As the saying goes, "All models are wrong, but some are useful." Our analysis is intended to shed light on why our method is robust—not to claim robustness under all possible data distributions.

To extend our robustness analysis to a broader class of distributions, we relax the Gaussian assumption to a sub-Gaussian setting and derive a corresponding lower bound on the Z-score under noise corruption. This setting accommodates features with light-tailed distributions, including bounded or discrete categorical features. See formal theorems and proof details in Appendix E.4.

# E THEORETICAL ANALYSIS AND PROOFS

## E.1 PROOF OF PROPOSITION 1

*Proof of Proposition 1.* Let $\Delta y_{i,j} = y_{i,j}^{\mathrm{wm}} - y_{i,j}$ denote the entry-wise difference in the frequency domain, then by Def. 2 and the Algorithm 1 with soft parameters $(\gamma, \delta)$, we have

$$\Delta y_{i,k} = \begin{cases} -\mathtt{i}(1+\delta) \cdot \Im(y_{i,k}), & k \in S, \\ \mathtt{i}(1+\delta) \cdot \Im(y_{i,p-k}), & p-k \in S, \\ 0, & \text{otherwise.} \end{cases}$$

By the inverse DFT as defined in Def. 2, the entry-wise difference $\Delta x_{i,j}$ is given by

$$\begin{aligned} \Delta x_{i,j} &= \frac{1}{\sqrt{p}} \left[ \sum_{k \in S} \Delta y_{i,k} e^{\mathtt{i} \frac{2\pi}{p} kj} + \sum_{p-k \in S} \Delta y_{i,k} e^{\mathtt{i} \frac{2\pi}{p} kj} \right] \\ &= \frac{2(1+\delta)}{\sqrt{p}} \left[ \sum_{k \in S} \Im(y_{i,k}) \sin \frac{2\pi}{p} kj \right]. \end{aligned}$$

Plugging in $\Im(y_{i,k}) = -\frac{1}{\sqrt{p}} \sum_{n=0}^{p-1} x_{in} \sin \frac{2\pi kn}{p}$ leads to

$$\Delta x_{i,j} = -\frac{2(1+\delta)}{p} \sum_{n=0}^{p-1} \left[ \sum_{k \in S} \sin \frac{2\pi kn}{p} \sin \frac{2\pi kj}{p} \right] x_{in},$$

which is precisely $\Delta x_{i,j} = -\alpha\, \boldsymbol{\beta}_j^\top \boldsymbol{x}_i$ with $\alpha = \frac{2(1+\delta)}{p}$ and the stated $\boldsymbol{\beta}_j$ and $\boldsymbol{x}_i$. □

### E.2 Proof of Theorem 1

We prove items one by one.

1. **Mean.** For each column $j = 0, \ldots, p-1$, we have
$$\frac{1}{N}\sum_{i=1}^{N} \Delta x_{i,j} = -\alpha\, \boldsymbol{\beta}_j^\top \left(\frac{1}{N}\sum_{i=1}^{N} \boldsymbol{x}_i\right) = 0,$$
since each column is centered.
2. **Pearson correlation coefficients (PCC).** Given that each column is centered and standardized, the difference of PCC between each column pair $(j, \ell)$ is given by
$$\Delta r_{j\ell} = \frac{1}{N}\sum_{i=1}^{N}(x_{i,j}\Delta x_{i,\ell} + x_{i,\ell}\Delta x_{i,j} + \Delta x_{i,j}\Delta x_{i,\ell}).$$
Plugging $\Delta x_{i,j} = -\alpha\, \boldsymbol{\beta}_j^\top \boldsymbol{x}_i$ leads to
$$\Delta r_{j\ell} = -\alpha\,(\,[\boldsymbol{\Sigma}\boldsymbol{\beta}_\ell]_j +\, [\boldsymbol{\Sigma}\boldsymbol{\beta}_j]_\ell) + \alpha^2\, \boldsymbol{\beta}_j^\top \boldsymbol{\Sigma}\boldsymbol{\beta}_\ell,$$
where $\boldsymbol{\Sigma} = \frac{1}{N}\mathbf{X}^\top\mathbf{X}$ with $\mathrm{diag}(\boldsymbol{\Sigma}) = \mathbb{I}$.
3. **Empirical distribution.** Consider the coupling matching $x_{i,j}$ to $x_{i,j} + \Delta x_{i,j}$ for each $i$, we bound the transport cost as below:
$$\mathcal{W}_2^2(\rho_j, \nu_j) \le \frac{1}{N}\sum_{i=1}^{N}(\Delta x_{i,j})^2 = \alpha^2 \boldsymbol{\beta}_j^\top \boldsymbol{\Sigma}\boldsymbol{\beta}_j,$$
which leads to the claimed inequality.

### E.3 Proof of Theorem 2

**Lemma 1** (Gaussian tail bound)**.** *Let $\Phi(u)$ denote the standard normal CDF and $Q(u) := 1 - \Phi(u)$. For any $u > 0$,*
$$Q(u) \le \frac{1}{2}e^{-u^2/2}.$$

*Proof of Lemma 1.* We discuss the bound under two cases.
**Case 1**: When $u > \sqrt{\frac{2}{\pi}}$, through integration by parts, we have
$$Q(u) = \frac{1}{\sqrt{2\pi}}\int_u^\infty e^{-t^2/2}dt \le \frac{1}{\sqrt{2\pi}}\left[\frac{e^{-u^2/2}}{u} - \int_u^\infty \frac{e^{-t^2/2}}{t^2}dt\right].$$
Dropping the negative integral preserves the inequality, yielding
$$Q(u) \le \frac{1}{u\sqrt{2\pi}}e^{-u^2/2} \le \frac{1}{2}e^{u^2/2}.$$

**Case 2**: When $0 < u \le \sqrt{\frac{2}{\pi}}$, we have
$$\frac{d}{du}\left(\frac{1}{2}e^{-u^2/2}\right) = -\frac{u}{2}e^{-u^2/2} \ge -\frac{1}{\sqrt{2\pi}}e^{-u^2/2} = \frac{d}{du}Q(u),$$
where the inequality follows from $u \le \sqrt{\frac{2}{\pi}}$. Integrating from 0 to $u$ gives
$$\int_0^u d\left(\frac{1}{2}e^{-t^2/2}\right) \ge \int_0^u dQ(t) \Rightarrow Q(u) \le \frac{1}{2}e^{-u^2/2}.$$

Combining the two cases yields the stated bound. □

**Lemma 2** (Noise in the frequency domain)**.** *Let* $\boldsymbol{\varepsilon} = (\varepsilon_0, \ldots, \varepsilon_{p-1})^\top \sim \mathcal{N}(\mathbf{0}, \sigma^2 \mathbf{I}_p)$ *be a real-valued Gaussian noise vector of length* $p$*. Apply the DFT in Def. 2 to obtain* $\hat{\boldsymbol{\varepsilon}} = (\hat{\varepsilon}_0, \ldots, \hat{\varepsilon}_{p-1})^\top$*. Denote by*

$$z_t = \Im(\hat{\varepsilon}_t), \quad t = 1, \ldots, m,$$

*the imaginary part of the noise component at the* $t$*-th effective entry. Then*

$$z_t \sim \mathcal{N}\left(0, \frac{\sigma^2}{2}\right).$$

*Proof of Lemma 2.* Denote $\theta_{t,n} := \frac{2\pi t n}{p}$, we have

$$z_t = -\frac{1}{\sqrt{p}} \sum_{n=0}^{p-1} \varepsilon_n \sin\left(\theta_{t,n}\right).$$

Note that $z_t$ is a linear combination of independent Gaussian variables, hence still be a Gaussian with zero mean. Since $\mathrm{Var}(\varepsilon_n) = \sigma^2$ and the $\varepsilon_n$'s are independent,

$$\mathrm{Var}[z_t] = \frac{1}{p}\sigma^2 \sum_{n=0}^{p-1} \sin^2\left(\theta_{t,n}\right).$$

Using the trigonometric identity $\sin^2 u = \frac{1}{2}\left(1 - \cos 2u\right)$,

$$\sum_{n=0}^{p-1} \sin^2(\theta_{t,n}) = \frac{p}{2} - \frac{1}{2}\sum_{n=0}^{p-1} \cos(2\theta_{t,n}).$$

The second sum is a geometric series whose value is 0 whenever $t \notin \{0, \frac{p}{2}\}$:

$$\sum_{n=0}^{p-1} e^{i\frac{4\pi t n}{p}} = \frac{1 - e^{i4\pi t}}{1 - e^{i\frac{4\pi t}{p}}} = 0.$$

Hence $\sum_{n=0}^{p-1} \sin^2(\theta_{t,n}) = \frac{p}{2}$ for each $t = 1, \ldots, m$. Substituting back,

$$\mathrm{Var}[z_t] = \frac{\sigma^2}{p} \cdot \frac{p}{2} = \frac{\sigma^2}{2}.$$

□

**Lemma 3** (Standard Z-score)**.** *If pseudorandom bits* $\zeta_{i,j} \overset{\text{i.i.d.}}{\sim}$ *Bernoulli*$(0.5)$ *and are independent of effective entries* $\{y_{i,j}\}$*, then* $\{T_i\}_{i=1}^N$*, as defined in Section 2, i.i.d. follows* $B(m, \frac{1}{2})$ *under* $H_0$*, thus has expected value* $\frac{m}{2}$ *and variance* $\frac{m}{4}$*. By Central Limit Theorem, the Z-score under* $H_0$ *follows*

$$Z = \frac{\sum_{i=1}^N T_i - \frac{mN}{2}}{\sqrt{\frac{mN}{4}}} \xrightarrow{d} \mathcal{N}(0, 1) \ \ \textit{as} \ \ N \to \infty.$$

*Proof of Lemma 3.* For each index pair $(i, j)$ of effective entries, define the events

$$E_{i,j} := \{\mathrm{sign}(\Im(y_{i,j})) = 2\,\zeta_{i,j} - 1\}, \qquad A_{i,j} := \{\mathrm{sign}(\Im(y_{i,j})) = 1\}.$$

We will show that the indicator variables $\{\mathbb{I}(E_{i,j})\}_{i,j}$ are independent and identically distributed as Bernoulli$(0.5)$.

First, set

$$p_{i,j} := \mathbb{P}\left(\mathrm{sign}(\Im(y_{i,j})) = 1\right).$$

By conditioning on $\zeta_{i,j} \in \{0, 1\}$ and using the fact that $\mathbb{P}(\zeta_{i,j} = 1) = \mathbb{P}(\zeta_{i,j} = 0) = \frac{1}{2}$, we obtain

$$\mathbb{P}(E_{i,j}) = p_{i,j}\,\mathbb{P}(\zeta_{i,j} = 1) + (1 - p_{i,j})\,\mathbb{P}(\zeta_{i,j} = 0) = \frac{1}{2}\,p_{i,j} + \frac{1}{2}\,(1 - p_{i,j}) = \frac{1}{2}.$$

Hence each $\mathbb{I}(E_{i,j}) \sim \text{Bernoulli}(0.5)$.

To verify independence, for any finite index set $\mathcal{I} \subseteq \{(i,j) \colon 1 \le i \le N, 1 \le j \le m\}$. we consider a family of events

$$\mathcal{B} = \{\, B_{\mathcal{I}_1,\mathcal{I}_2} \colon \mathcal{I}_1 \cup \mathcal{I}_2 = \mathcal{I}, \mathcal{I}_1 \cap \mathcal{I}_2 = \emptyset \}\,, \quad B_{\mathcal{I}_1,\mathcal{I}_2} = \left( \bigcap_{(i,j)\in\mathcal{I}_1} A_{i,j} \right) \cap \left( \bigcap_{(i,j)\in\mathcal{I}_2} A_{i,j}^c \right).$$

Clearly $\mathcal{B}$ is a partition of the sample space $\Omega$, hence we have

$$\begin{aligned} \mathbb{P}\left( \bigcap_{(i,j)\in\mathcal{I}} E_{i,j} \right) &= \sum_{B_{\mathcal{I}_1,\mathcal{I}_2}\in\mathcal{B}} \mathbb{P}(B_{\mathcal{I}_1,\mathcal{I}_2}) \prod_{(i,j)\in\mathcal{I}_1} \mathbb{P}(\zeta_{i,j}=1) \prod_{(i,j)\in\mathcal{I}_2} \mathbb{P}(\zeta_{i,j}=0) \\ &= \sum_{B_{\mathcal{I}_1,\mathcal{I}_2}\in\mathcal{B}} \mathbb{P}(B_{\mathcal{I}_1,\mathcal{I}_2}) \frac{1}{2^{|\mathcal{I}|}} = \frac{1}{2^{|\mathcal{I}|}} = \prod_{(i,j)\in\mathcal{I}} \mathbb{P}(E_{i,j}), \end{aligned}$$

This implies that the collection of events $\{E_{i,j}\}_{i,j}$ is mutually independent. Together with the fact that $\mathbb{P}(E_{i,j}) = \frac{1}{2}$, this completes the proof.

□

*Proof of Theorem 2.* We continue with the notations established in Lemmas 2 and 3. Let

$$\mathbf{X} = \{x_{i,j}\} \in \mathbb{R}^{N\times p}, \quad \boldsymbol{x}_i := (x_{i,0},\ldots,x_{i,p-1}) \overset{\text{i.i.d.}}{\sim} \mathcal{N}(0,\Sigma),$$

and denote the frequency-domain representation by $\mathbf{Y} \in \mathbb{C}^{N\times p}$. Then for each row, the effective entries satisfy:

$$\Im(y_t) = -\frac{1}{\sqrt{p}} \sum_{n=0}^{p-1} x_n \sin(\theta_{t,n}),$$

where $\theta_{t,n} = \frac{2\pi tn}{p}$. Let $\boldsymbol{s}_t$ denotes $(\sin(\theta_{t,0}),\ldots,\sin(\theta_{t,p-1})) \in \mathbb{R}^{1\times p}$. Since each $\boldsymbol{x}_i$ is Gaussian, $\Im(y_t)$ is Gaussian with zero mean and

$$\text{Var}\left[\Im(y_t)\right] = \frac{1}{p} \boldsymbol{s}_t^\top \Sigma \boldsymbol{s} \in \left[ \frac{\lambda_{\min}}{p} \left\|\boldsymbol{s}_t\right\|_2^2, \frac{\lambda_{\max}}{p} \left\|\boldsymbol{s}_t\right\|_2^2 \right] = \left[ \frac{\lambda_{\min}}{2}, \frac{\lambda_{\max}}{2} \right],$$

where $\|\boldsymbol{s}_t\|_2^2 = \frac{p}{2}$ follows from Lemma 2 and $\lambda_{\min}$ and $\lambda_{\max}$ denote the smallest and largest eigenvalue of the covariance matrix $\Sigma$, respectively.

Given a pseudorandom bit $\zeta_t \in \{0,1\}$, the process of watermark embedding in Algorithm 1 replaces $\Im(y_t)$ by

$$\Im\left(y_t^{\text{wm}}\right) = \begin{cases} -\delta \cdot \Im(y_t), & \text{if } \Im(y_t)\cdot(2\zeta_t - 1) < 0 \text{ and } |\Im(y_t)| \le \text{Quantile}_\gamma(\{|\Im(y_t)|\}_{t=1}^m), \\ \Im\left(y_t\right), & \text{otherwise,} \end{cases}$$

Let

$$\alpha_t := \left|\Im(y_t^{\text{wm}})\right|, \quad \frac{\lambda}{2} := \text{Var}\big[\Im(y_t)\big] \in \Big[\frac{\lambda_{\min}}{2}, \frac{\lambda_{\max}}{2}\Big],$$

and define

$$F(x) := \frac{2}{\sqrt{\pi\lambda}}\, e^{-\frac{x^2}{\lambda}}\, \mathbb{I}(x \ge 0), \quad F_\delta(x) := \frac{2}{\delta\sqrt{\pi\lambda}} \exp\!\Big(-\frac{x^2}{\delta^2\lambda}\Big)\, \mathbb{I}(x \ge 0),$$

which are the PDFs of $\alpha_t$ and $\delta\,\alpha_t$, respectively. Under large $p$, there are three scenarios for each $t$:

- **Case 1**: With probability $\frac{1}{2}$, $\alpha_t \sim F$ and $\Im(y_t^{\text{wm}}) \cdot (2\zeta_t - 1) > 0$.
- **Case 2**: With probability $\frac{\gamma}{2}$, $\alpha_t \sim F_\delta$ and $\Im(y_t^{\text{wm}}) \cdot (2\zeta_t - 1) > 0$.
- **Case 3**: With probability $\frac{1-\gamma}{2}$, $\alpha_t \sim F$ and $\Im(y_t^{\text{wm}}) \cdot (2\zeta_t - 1) < 0$.

**Sign-flip probability under additive noise.** Let $z_t \sim \mathcal{N}(0, \frac{\sigma^2}{2})$ be the imaginary-part noise as derived in Lemma 2. Conditioned on $\alpha_t = x$, the probability that noise flips the sign of $\Im(y_t^{\mathrm{wm}})$ in **Case 1** and **Case 3** is

$$\begin{aligned}
\mathbb{P}_{\text{flip}}(\sigma) &= \mathbb{P}\left(z_t > \alpha_t | \alpha_t \sim F\right) \\
&= \int_0^\infty \mathbb{P}\left(z_t > \alpha_t |\, \alpha_t = x\right) F(x) dx \\
&= \int_0^\infty \frac{2}{\sqrt{\pi\lambda}} e^{-\frac{x^2}{\lambda}} Q\left(\frac{\sqrt{2}\,x}{\sigma}\right) dx \\
&\leq \frac{1}{\sqrt{\pi\,\lambda_{\min}}} \int_0^{\sqrt{\frac{\lambda_{\min}}{2}}} e^{-x^2\left(\frac{1}{\lambda_{\min}}+\frac{1}{\sigma^2}\right)}\, dx \\
&\quad + \frac{1}{\sqrt{\pi\,\lambda_{\max}}} \int_{\sqrt{\frac{\lambda_{\max}}{2}}}^\infty e^{-x^2\left(\frac{1}{\lambda_{\max}}+\frac{1}{\sigma^2}\right)}\, dx \\
&\quad + \frac{e^{-\frac{1}{2}}}{\sqrt{2\pi}} \int_{\sqrt{\frac{\lambda_{\min}}{2}}}^{\sqrt{\frac{\lambda_{\max}}{2}}} \frac{e^{-\frac{x^2}{\sigma^2}}}{x} dx \\
&= \frac{\sigma}{\sqrt{\sigma^2+\lambda_{\min}}}\left[\Phi\left(\sqrt{1+\frac{\lambda_{\min}}{\sigma^2}}\right) - \frac{1}{2}\right] \\
&\quad + \frac{\sigma}{\sqrt{\sigma^2+\lambda_{\max}}}\left[1-\Phi\left(\sqrt{1+\frac{\lambda_{\max}}{\sigma^2}}\right)\right] \\
&\quad + \frac{1}{\sqrt{8\pi e}}\left[E_1\left(\frac{\lambda_{\min}}{2\sigma^2}\right) - E_1\left(\frac{\lambda_{\max}}{2\sigma^2}\right)\right],
\end{aligned}$$

where the inequality follows from Lemma 1 and the local monotonicity of $F(x)$. Similarly, if the amplitude of $\alpha_t$ is scaled by $\delta$, we obtains $\mathbb{P}^{(\delta)}_{\text{flip}}(\sigma) = \mathbb{P}\left(z_t > \alpha_t | \alpha_t \sim F_\delta\right) \leq \mathcal{I}(\frac{\sigma}{\delta})$, where

$$\mathcal{I}(s) := \frac{s}{\sqrt{s^2+\lambda_{\min}}}\Big[\Phi\Big(\sqrt{1+\frac{\lambda_{\min}}{s^2}}\Big) - \frac{1}{2}\Big] + \frac{s}{\sqrt{s^2+\lambda_{\max}}}\Big[1-\Phi\Big(\sqrt{1+\frac{\lambda_{\max}}{s^2}}\Big)\Big] + \frac{1}{\sqrt{8\pi e}}\Big[E_1\Big(\frac{\lambda_{\min}}{2s^2}\Big) - E_1\Big(\frac{\lambda_{\max}}{2s^2}\Big)\Big].$$

**Alignment probability after attack.** Let $p_{i,j}$ be the probability that the $j$-th effective entry in row $i$ maintains alignment with its corresponding pseudorandom bit under attack. Combining the three cases above, we have

$$\begin{aligned}
p_{i,j} &= \frac{1}{2}\left(1-\mathbb{P}_{\text{flip}}(\sigma)\right) + \frac{\gamma}{2}\left(1-\mathbb{P}^{(\delta)}_{\text{flip}}(\sigma)\right) + \frac{1-\gamma}{2}\,\mathbb{P}_{\text{flip}}(\sigma) \\
&= \frac{1+\gamma}{2} - \frac{\gamma}{2}\left[\mathbb{P}_{\text{flip}}(\sigma) + \mathbb{P}^{(\delta)}_{\text{flip}}(\sigma)\right] \\
&\geq \frac{1+\gamma}{2} - \frac{\gamma}{2}\left[\mathcal{I}(\sigma) + \mathcal{I}\Big(\frac{\sigma}{\delta}\Big)\right]
\end{aligned} \tag{5}$$

**Lower bound on the expected Z-score.** Under this setting, we recall Lemma 3 for the standard Z-score $Z = \dfrac{\sum_{i,j}\mathbb{I}\{E_{i,j}\} - \frac{mN}{2}}{\sqrt{\frac{mN}{4}}}$, then we obtain

$$\mathbb{E}\left[Z(\gamma,\delta,\sigma)\right] = \frac{mN\,p_{i,j} - \frac{mN}{2}}{\sqrt{\frac{mN}{4}}} \geq \sqrt{mN}\gamma\left[1-\mathcal{I}(\sigma)-\mathcal{I}(\frac{\sigma}{\delta})\right],$$

which completes the proof. □

### E.4 Further Analysis on Robustness

Based on the assumption and notations in Theorem 2, we also derived a conservative, non-asymptotic lower bound on the number of rows $N$ required to achieve a statistical test with power $1-\beta$ at

significance level $\alpha$. Theorem 3 gives a formal description. Since real-world tabular data are highly heterogeneous and rarely follow a strict multivariate Gaussian distribution even after YJT, we relax the Gaussian assumption in Theorem 2 to a sub-Gaussian setting and provide a corresponding lower bound on the Z-score under noise corruption. This setting accommodates features with light-tailed distributions, including bounded or discrete categorical features. A formal statement is given in Theorem 4.

**Theorem 3.** *We assume that unwatermarked tabular data* $\mathbf{X}$ *has rows* $\boldsymbol{x}_i \overset{\text{i.i.d.}}{\sim} \mathcal{N}(0, \Sigma)$, *where* $\Sigma \in \mathbb{R}^{p\times p}$ *is positive-definite. Denote the smallest and largest eigenvalue of* $\Sigma$ *by* $\lambda_{\min}$ *and* $\lambda_{\max}$, *respectively. Define* $N_{\alpha,\beta}(\gamma, \delta, \sigma)$ *as the number of rows required for the Gaussian noise-corrupted table* $\mathbf{X}_{\text{wm}} + \boldsymbol{\varepsilon}$ *to achieve a statistical test with power* $1-\beta$ *at significance level* $\alpha$, *where* $\mathbf{X}_{\text{wm}}$ *denote the table watermarked under soft hyperparameters* $(\gamma, \delta)$ *and* $\varepsilon_{i,j} \overset{\text{i.i.d.}}{\sim} \mathcal{N}(0, \sigma^2)$. *Then, for any* $\gamma, \alpha, \beta \in [0, 1]$ *and* $\delta, \sigma > 0$,

$$N_{\alpha,\beta}(\gamma, \delta, \sigma) \geq \frac{\left[q_\alpha + \sqrt{2m \ln\left(\frac{1}{\beta}\right)}\right]^2}{m\gamma^2 \left[1 - \mathcal{I}(\sigma) - \mathcal{I}(\frac{\sigma}{\delta})\right]^2}, \tag{6}$$

*where* $m = \lfloor \frac{p-1}{2} \rfloor$ *and* $q_\alpha$ *is the critical value for a one-sided test at level* $\alpha$. *The function* $\mathcal{I} : (0, \infty) \to \mathbb{R}$ *is defined as*

$$\mathcal{I}(s) := \frac{s}{\sqrt{s^2+\lambda_{\min}}}\left[\Phi\Big(\sqrt{1+\frac{\lambda_{\min}}{s^2}}\Big) - \frac{1}{2}\right] + \frac{s}{\sqrt{s^2+\lambda_{\max}}}\left[1 - \Phi\Big(\sqrt{1+\frac{\lambda_{\max}}{s^2}}\Big)\right] + \frac{1}{\sqrt{8\pi e}}\left[E_1\Big(\frac{\lambda_{\min}}{2s^2}\Big) - E_1\Big(\frac{\lambda_{\max}}{2s^2}\Big)\right].$$

*with* $\Phi(\cdot)$ *denoting the standard normal CDF and* $E_1(u) = \int_u^\infty \frac{e^{-t}}{t}\, dt$ *the exponential integral.*

*Proof of Theorem 3.* Denote the random variables after noise perturbation as below: $I_{i,j} := \mathbb{I}\{\Im(y_{i,j})(2\zeta_{i,j} - 1) > 0\}$, $T_i := \sum_{j=1}^m I_{i,j}$, and $S_N := \sum_{i=1}^N T_i$. We omit their explicit dependence on hyperparameters $(\gamma, \delta, \sigma)$ here for simplicity. From Lemma 3, the Z-score follows

$$Z = \frac{S_N - \frac{mN}{2}}{\sqrt{\frac{mN}{4}}}.$$

By Eq.(5), we have

$$\mathbb{E}_{H_1}[S_N] = \sum_{i=1}^N \sum_{j=1}^m p_{i,j} \geq mN\left(\frac{1+\gamma}{2} - \frac{\gamma}{2}\left[\mathcal{I}(\sigma) + \mathcal{I}(\frac{\sigma}{\delta})\right]\right).$$

Then for a one-sided level-$\alpha$ test with threshold $q_\alpha$, we have

$$\{Z \leq q_\alpha\} \subseteq \{S_N - \mathbb{E}_{H_1}[S_N] \leq -t_N\}, \quad t_N := \left(\sqrt{mN}\gamma\left[1 - \mathcal{I}(\sigma) - \mathcal{I}(\frac{\sigma}{\delta})\right] - q_\alpha\right)\frac{\sqrt{mN}}{2},$$

Since $T_i$ are independent and bounded in $[0, m]$ (i.i.d. rows and row-wise watermarking), We apply Hoeffding's inequality:

$$\mathbb{P}_{H_1}\{Z \leq q_\alpha\} \leq \exp\left(-\frac{2t_N^2}{Nm^2}\right) = \exp\left(-\frac{\left(\sqrt{mN}\gamma\left[1 - \mathcal{I}(\sigma) - \mathcal{I}(\frac{\sigma}{\delta})\right] - q_\alpha\right)^2}{2m}\right).$$

Imposing $\mathbb{P}_{H_1}\{Z \leq q_\alpha\} \leq \beta$ gives a conservative lower bound:

$$\sqrt{mN}\gamma\left[1 - \mathcal{I}(\sigma) - \mathcal{I}\left(\frac{\sigma}{\delta}\right)\right] \geq q_\alpha + \sqrt{2m\ln(1/\beta)},$$

which implies the nonasymptotic sample-size lower bound to achieve a test of power $1-\beta$ at level $\alpha$:

$$N_{\alpha,\beta}(\gamma, \delta, \sigma) \geq \frac{\left[q_\alpha + \sqrt{2m \ln\left(\frac{1}{\beta}\right)}\right]^2}{m\gamma^2 \left[1 - \mathcal{I}(\sigma) - \mathcal{I}(\frac{\sigma}{\delta})\right]^2}.$$

□

The Remark 4 below provides a numerical illustration of Theorem 3.

**Remark 4.** *We provide a numerical example to illustrate the guarantee in Theorem 3. When $p = 11 (m = 5)$ and $\Sigma = \mathbb{I}_{p\times p}$, the theoretical lower bound (the right-hand side of (6)) for the number of rows $N$ required to achieve a test of power $1 - \beta = 0.99$ at level $\alpha = 0.001$, i.e. 0.99 TPR@0.1%FPR, is given by:*

$$N_{0.001,0.01}(0.5, 0.5, \sigma) \geq \begin{cases} 108, & \text{if } \sigma = 0.1, \\ 153, & \text{if } \sigma = 0.2, \\ 437, & \text{if } \sigma = 0.5. \end{cases}$$

**Theorem 4** (Robustness under sub-Gaussian samples)**.** *Assume that unwatermarked tabular data $\mathbf{X} \in \mathbb{R}^{N\times p}$ has i.i.d. rows $\boldsymbol{x_i} \in \mathbb{R}^p$ with zero mean and covariance $\Sigma \in \mathbb{R}^{p\times p}$, and are $\Sigma$–sub-Gaussian, meaning there exists $\kappa \geq 1$ such that for every $u \in \mathbb{R}^p$, $\|\langle u, \boldsymbol{x_i}\rangle\|_{\psi_2} \leq \kappa\sqrt{u^\top \Sigma u}$ (Vershynin, 2018). Denote the smallest eigenvalue of $\Sigma$ by $\lambda_{\min}$. Define $Z(\gamma, \delta, \sigma)$ as the standard Z-score (as in (3)) computed on the Gaussian noise-corrupted table $\mathbf{X}_{\text{wm}} + \boldsymbol{\varepsilon}$, where $\mathbf{X}_{\text{wm}} \in \mathbb{R}^{N\times p}$ denote the table watermarked under soft hyperparameters $(\gamma, \delta)$ and $\varepsilon_{i,j} \overset{\text{i.i.d.}}{\sim} \mathcal{N}(0, \sigma^2)$. Fix any $\theta \in (0, 1)$ and let $C_4 > 0$ denote a constant such that for every real sub-Gaussian $U$ with variance $v$ we have $\mathbb{E}U^4 \leq C_4\, \kappa^4\, v^2$. Define $\rho(\kappa, \theta) := \frac{(1-\theta)^2}{2\,C_4\,\kappa^4}$. Then, for any $\gamma \in [0, 1]$ and $\delta, \sigma > 0$,*

$$\mathbb{E}\big[Z(\gamma, \delta, \sigma)\big] \;\geq\; \sqrt{mN}\,\gamma \sup_{\theta\in(0,1)} \left\{\rho(\kappa, \theta)\left[2 - \exp\Big(-\frac{\theta\,\lambda_{\min}}{2\,\sigma^2}\Big) - \exp\Big(-\frac{\theta\,\lambda_{\min}\,\delta^2}{2\,\sigma^2}\Big)\right]\right\}.$$

*Proof of Theorem 4.* Let $\boldsymbol{x} = (x_0, \dots, x_{p-1})$ be a standardized row and $\boldsymbol{y} = \texttt{DFT}(\boldsymbol{x})$. For the $t$-th effective frequency, we have

$$\Im(y_t) \;=\; -\frac{1}{\sqrt{p}}\sum_{n=0}^{p-1} x_n \sin\Big(\frac{2\pi t n}{p}\Big) \;=\; -\frac{1}{\sqrt{p}}\,\boldsymbol{s_t}^\top \boldsymbol{x}, \qquad \|\boldsymbol{s_t}\|_2^2 = \sum_{n=0}^{p-1}\sin^2\Big(\frac{2\pi t n}{p}\Big) = \frac{p}{2}.$$

By the $\Sigma$–sub-Gaussian assumption and linearity, $\Im(y_t)$ is sub-Gaussian with

$$v_t := \mathrm{Var}[\Im(y_t)] = \frac{1}{p}\,\boldsymbol{s_t}^\top \Sigma \boldsymbol{s_t} \;\in\; \Big[\frac{\lambda_{\min}}{2}, \frac{\lambda_{\max}}{2}\Big], \qquad \|\Im(y_t)\|_{\psi_2} = \Big\|-\frac{1}{\sqrt{p}}\,\boldsymbol{s_t}^\top \boldsymbol{x}\Big\|_{\psi_2} \;\leq\; \kappa\sqrt{v_t}.$$

Let $\alpha_t := |\Im(y_t^{\text{wm}})|$ and $z_t := \Im(\widehat{\varepsilon}_t) \sim N(0, \sigma^2/2)$ be the imaginary-part noise (Lemma 2). Follwing the analysis in Theorem 2, for each effective entry the alignment probability with its bit satisfies

$$p_{i,j} \;=\; \frac{1+\gamma}{2} \;-\; \frac{\gamma}{2}\Big(\mathbb{P}_{\text{flip}}(\sigma) + \mathbb{P}^{(\delta)}_{\text{flip}}(\sigma)\Big), \tag{7}$$

where $\mathbb{P}_{\text{flip}}(\sigma) = \mathbb{E}\big[\mathbb{I}\{z_t > \alpha_t\}\big] = \mathbb{E}\big[Q(\sqrt{2}\,\alpha_t/\sigma)\big]$, and $\mathbb{P}^{(\delta)}_{\text{flip}}$ is the same quantity with the amplitude scaled by $|\delta|$. Here $Q(u) = 1 - \Phi(u)$.

By Lemma 1, $Q(u) \leq \frac{1}{2}e^{-u^2/2}$, hence $\mathbb{P}_{\text{flip}}(\sigma) \leq \frac{1}{2}\,\mathbb{E}\exp(-\alpha_t^2/\sigma^2)$. Write $U := \Im(y_t)$ and $Y := U^2$. For any $\theta \in (0, 1)$, Paley–Zygmund inequality gives

$$\mathbb{P}\big(Y \geq \theta\,\mathbb{E}Y\big) \;\geq\; \frac{(1-\theta)^2\,(\mathbb{E}Y)^2}{\mathbb{E}Y^2} \;=\; \frac{(1-\theta)^2\,v_t^2}{\mathbb{E}U^4} \;\geq\; \frac{(1-\theta)^2}{C_4\,\kappa^4} \;=:\; \eta,$$

where we used the sub-Gaussian fourth-moment bound $\mathbb{E}U^4 \leq C_4\,\kappa^4 v_t^2$. Thus, by conditioning on the event $\{Y \geq \theta v_t\}$,

$$\mathbb{E}\exp(-\alpha_t^2/\sigma^2) \;\leq\; (1-\eta)\cdot 1 \;+\; \eta\cdot\exp\Big(-\frac{\theta v_t}{\sigma^2}\Big) \;\leq\; 1 - \eta\Big(1 - e^{-\theta\lambda_{\min}/(2\sigma^2)}\Big).$$

Consequently,

$$\mathbb{P}_{\text{flip}}(\sigma) \;\leq\; \frac{1}{2}\Big[1 - \eta\Big(1 - e^{-\theta\lambda_{\min}/(2\sigma^2)}\Big)\Big], \qquad \mathbb{P}^{(\delta)}_{\text{flip}}(\sigma) \;\leq\; \frac{1}{2}\Big[1 - \eta\Big(1 - e^{-\theta\lambda_{\min}\delta^2/(2\sigma^2)}\Big)\Big].$$

Plugging these bounds into Eq.(7), then we obtain

$$p_{i,j} \ \geq \ \frac{1}{2} \ + \ \frac{\gamma\eta}{4}\left[2 - e^{-\theta\lambda_{\min}/(2\sigma^2)} - e^{-\theta\lambda_{\min}\delta^2/(2\sigma^2)}\right].$$

Using $\mathbb{E}Z = 2\sqrt{mN}\,(p_{i,j} - \frac{1}{2})$, we arrive at

$$\mathbb{E}[Z(\gamma,\delta,\sigma)] \ \geq \ \sqrt{mN}\,\gamma\,\frac{\eta}{2}\left[2 - e^{-\theta\lambda_{\min}/(2\sigma^2)} - e^{-\theta\lambda_{\min}\delta^2/(2\sigma^2)}\right].$$

Recalling that $\eta = (1-\theta)^2/(C_4\kappa^4)$ gives the result for $\rho(\kappa,\theta) = (1-\theta)^2/(2C_4\kappa^4)$. Since $\theta \in (0,1)$ is a fixed hyperparameter, it can be tuned to obtain the tightest possible lower bound

$$\mathbb{E}\big[Z(\gamma,\delta,\sigma)\big] \ \geq \ \sqrt{mN}\,\gamma \sup_{\theta\in(0,1)} \left\{\rho(\kappa,\theta)\left[2 - \exp\Big(-\frac{\theta\,\lambda_{\min}}{2\,\sigma^2}\Big) - \exp\Big(-\frac{\theta\,\lambda_{\min}\,\delta^2}{2\,\sigma^2}\Big)\right]\right\}.$$

□

**Remark 5** (What the sub-Gaussian assumption covers)**.** *The $\Sigma$-sub-Gaussian assumption strictly generalizes the Gaussian assumption used in Theorem 2 and many non-Gaussian settings that are common in tabular data: 1) bounded/quantized/discrete features (e.g., "gender" or "education"features); and 2) finite mixtures of light-tailed distributions with uniformly bounded covariances (a finite mixture of sub-Gaussians is sub-Gaussian with the worst-component parameter). In conclusion, non-Gaussian distribution with light tails are covered.*

# F Experimental Details

## F.1 Datasets

The datasets used for evaluation are described in Table 7, where # Rows, # Categorical, # Numerical, # Continuous indicate the number of rows, the number of categorical columns, the number of numerical columns, and the number of numerical columns with continuous density function, respectively. # Train and # Test indicate the number of samples in the training and testing set for downstream machine learning tasks, respectively. The **Adult** (Becker & Kohavi, 1996) dataset was extracted from the 1994 Census database, containing 9 categorical and 6 numerical columns. The **Magic** (Bock, 2004) dataset simulates registration of high energy gamma particles in a ground-based atmospheric Cherenkov gamma telescope and consists of one categorical column and 10 numerical columns. The **Shoppers** (Sakar & Kastro, 2018) dataset quantifies online shoppers' purchasing intentions with 10 categorical columns and 8 numerical columns. The **Default** (Yeh, 2009) dataset presents the default payments of credit card clients, including 10 categorical columns and 14 numerical columns. The **Drybean** (Koklu & Özkan, 2020) dataset provides image information of seven different registered dry beans, comprising one categorical column and 16 numerical columns.

Here, we explicitly distinguish between continuous from integer-valued (discrete) numerical features, rather than conflating "numerical" with "continuous". As shown in Table 7, the **Adult** and **Default** datasets contain only discrete feature (0 continous features), while **Magic** and **Drybean** are dominated by continuous features. Therefore, we believe that the selected benchmark datasets provide a sufficiently balanced evaluation across both discrete and continuous data types, supporting the claim that TAB-DRW is applicable to heterogeneous tabular data.

Table 7: Descriptions of datasets used in evaluation.

| Name | Domain | # Rows | # Categorical | # Numerical | # Continuous | # Train | # Test | Task |
|---|---|---|---|---|---|---|---|---|
| Adult | Society | 48,842 | 9 | 6 | 0 | 32,561 | 16,281 | Classification |
| Magic | Physics | 19,019 | 1 | 10 | 10 | 17,117 | 1,902 | Classification |
| Shoppers | Business | 12,330 | 8 | 10 | 3 | 11,097 | 1,233 | Classification |
| Default | Finance | 30,000 | 10 | 14 | 0 | 27,000 | 3,000 | Classification |
| Drybean | Biology | 13,611 | 1 | 16 | 14 | 12,249 | 1,362 | Classification |

### F.2 METRICS

We detail our data fidelity metrics below:

1. **Density** measures the distributional similarity between synthetic and real data. For each numerical column, we compute the Kolmogorov–Smirnov statistic (KST); for each categorical column, we compute the total variation distance (TVD). The per-column scores are then averaged to yield the overall Density score. Higher values indicate closer alignment of marginal distributions.

2. **Corr** evaluates preservation of inter-column relationships. We compute the Pearson correlation coefficient for every pair of columns and report their mean as the Corr score. Larger values indicate more faithful reproduction of real feature dependencies.

3. **C2ST** quantifies statistical indistinguishability between synthetic and real data. A logistic regression model is trained and evaluated on the training and validation sets, both of which contain a mix of real and synthetic data. We then report the complement of the ROC AUC averaged over all validation splits. Higher values indicate that the model cannot distinguish synthetic from real data.

4. **MLE**: assesses downstream machine learning utility on supervised tasks. We train an XGBoost model (Chen & Guestrin, 2016) on synthetic data, then evaluate it on the real testing set, using AUC for the classification task and RMSE for the regression task. Higher MLE scores reflect better machine learning utility of the synthetic data.

Regarding the metric for watermark detectability, we introduce the one-sided **Z-score** defined in (3) and **FPR / TPR**. A larger Z-score indicates stronger alignment with the pseudorandom bits, thus demonstrating better watermark detectability. we synthesize 100 unwatermarked tables with 1K rows (total 100K rows) and perform Monte Carlo simulation to obtain statistics under $H_0$. Specifically, for the estimation of empirical critical value (eg. $q_{0.001}$), we conduct the following procedure:

1. Generate 100 unwatermarked tables with 1K rows (100K rows in total) using TabSyn.

2. Bootstrap sampling rows from 100K rows to construct 100K synthetic tables for watermark detection.

3. Set the 100-th order-statistic $Z_{(100)}$ as the threshold.

Then we have $F_{H_0}(Z_{(100)}) \sim \text{Beta}(100, 99901)$. By Clopper-Pearson interval, the estimation procedure above is sufficient to calibrate the critical value $q_{0.001}$, since the realized FPR has a 95% confidence interval of roughly $0.001 \pm 2 \times 10^{-4}$.

Table 8 presents the empirical mean and standard deviation of the alignment count $T_i$ defined in (3) under $H_0$ and critical values $q_\alpha$ for $\alpha \in \{0.01, 0.005, 0.001\}$.

FPR / TPR denotes the true and false positive rates under the critical value $q_\alpha = 6$. The FPR refers to the probability of incorrectly identifying an unwatermarked table as containing watermark signal, while the TPR refers to the probability of correctly detecting the watermark signal in a watermarked table. Therefore, an FPR / TPR pair of $(0.00, 1.00)$ indicates an effective watermark—no false alarms and complete detection.

Table 8: Results of Monte Carlo simulation on 100 unwatermarked synthetic tables of 1K rows. Each Entry show an empirical estimation (first value) and a theoretical value (second value) under standard assumption in Lemma 3.

| Dataset | $\hat{\mu}_{\text{nwm}}/\mu_{\text{nwm}}$ | $\hat{\sigma}_{\text{nwm}}/\sigma_{\text{nwm}}$ | $\hat{q}_{0.01}/q_{0.01}$ | $\hat{q}_{0.005}/q_{0.005}$ | $\hat{q}_{0.001}/q_{0.001}$ |
|---|---|---|---|---|---|
| Adult | 0.86/1.00 | 0.62/0.71 | 2.34/2.32 | 2.59/2.57 | 3.11/3.09 |
| Magic | 2.01/2.00 | 0.87/1.00 | 2.28/2.32 | 2.52/2.57 | 3.03/3.09 |
| Shoppers | 2.03/2.00 | 0.97/1.00 | 2.51/2.32 | 2.78/2.57 | 3.34/3.09 |
| Default | 1.42/1.50 | 0.78/0.87 | 2.39/2.32 | 2.64/2.57 | 3.16/3.09 |
| Drybean | 1.77/2.00 | 0.89/1.00 | 2.52/2.32 | 2.78/2.57 | 3.35/3.09 |

For TabularMark, we replace our Z-score with the one defined in (Zheng et al., 2024):

$$Z = \frac{2\,(n_g - 0.5\,n_w)}{\sqrt{n_w}},$$

where $n_w$ is the total number of key cells and $n_g$ is the count within the "green" domain.

### F.3 Implementation Details for Data Generator

**TabSyn.** TabSyn (Zhang et al., 2024a) is an architecture designed for high-quality tabular data synthesis. It addresses the challenges of mixed-type features by mapping raw tabular inputs—including numerical, categorical, and other modalities—into a continuous latent space using a customized variational autoencoder (VAE (Kingma & Welling, 2014)). The VAE employs Transformer-based encoders and decoders to effectively model inter-column dependencies and generate token-level embeddings. In the embedding space, TabSyn leverages a score-based diffusion model with a simplified linear noise schedule, which enables efficient sampling and preserving fidelity to the original data distribution. The combination of autoencoding and latent-space diffusion allows TabSyn to generate diverse and realistic synthetic tabular data with high efficiency and quality.

**Implementation in our work.** While our experiments are conducted using the TabSyn framework as the generative backbone, we adhere to the implementation in TabWak (Zhu et al., 2025), which is our primary baseline for comparison, to ensure fair and consistent comparison. Specifically, we replace the original score-based diffusion process in TabSyn with DDPM for training and DDIM for sampling. Therefore, our reproduced baseline results ("W/O") reported in Table 2 are closely aligned with those in Table 1 of TabWak.

The discrepancies between our "W/O" results and those reported in the original TabSyn paper (Zhang et al., 2024a) stem from the two modifications mentioned above. As reported in the TabSyn paper itself, the original score-based TabSyn model outperformed the TabSyn-DDPM variant in generation quality owing to its tailored diffusion process in continuous latent space. Additionally, the deterministic nature of the DDIM sampler may reduce data diversity compared to the original score-based SDE sampler. Nevertheless, the DDIM sampler is essential for TabWak, as its watermark detection relies on the inversion process.

In Table 9, we also present the evaluation results of our methods on synthetic tabular data generated by original TabSyn implementation. See Appendix G.1 for more empirical results and analysis.

### F.4 Implementation Details for Watermarking

**Our method.** We sample half of the column indices using a secret key to form the index set $\mathcal{I}$ in Algorithm 3. The selection of hyperparameter $\lambda$ in YJT is implemented by the Python module `scipy.stats.yeojohnson` (SciPy Developers, 2025). For the Adult, Magic and Drybean datasets, we apply Algorithm 1 to all numerical columns. For the Shoppers dataset, we apply watermarking to first 9 numerical columns. For the Default dataset, we select columns $\{0, 4, 17, 18, 19, 20, 21, 22\}$ for watermarking. We provide explanations on this implementation details in Remark 2.

**MUSE.** Following the experimental setting in Fang et al. (2025), we adopt Bernoulli as the scoring function and an adaptive column selection mechanism with three columns. We also adhere to the original configuration of $m = 2$, meaning that between two candidate rows, the one with the higher score is selected as the watermarked sample. Since TabSyn generates tabular data as a whole rather than row by row, we generate twice the target number of rows and then perform selection, consistent with the workflow illustrated in Algorithm 1 of Fang et al. (2025).

**TabWak$^*$.** We rigorously reproduce TabWak$^*$ with valid bit mechanism by following all instructions provided in the official repository `https://github.com/chaoyitud/TabWak`. However, we observed a significant discrepancy between our reproduction results and those reported in the original TabWak paper (Zhu et al., 2025). Below, we clarify the source of this discrepancy.

In TabWak's detection pipeline, the suspect tabular data $\mathbf{X}$ is first mapped into a continuous latent space via the inversion of the VAE decoder to obtain an initial latent representation $\mathbf{z}_0$. This is then

passed through DDIM inversion to recover the watermarked representation $\mathbf{z}_T$. In practice, TabWak codebase obtains $\mathbf{z}_0$ from $\mathbf{X}$ by solving a gradient-based optimization problem to approximate the inversion of the VAE decoder, which is formulated as:

$$\mathbf{z} = \arg\min_{\mathbf{z}} \|\mathbf{x} - f_\theta(\mathbf{z})\|_2,$$

where $f_\theta$ denotes the trained VAE decoder.

However, we found that the optimization procedure often fails to converge to the true latent code $\mathbf{z}_0$, which leads to significantly reduced detectability and robustness of the watermark signal (as reported in our paper). We also note that in the official TabWak codebase, the ground-truth $\mathbf{z}_0$ is saved during generation. Using this saved $\mathbf{z}_0$ bypasses the inversion step and yields strong detectability, which is comparable to that reported in the original TabWak paper. But this approach is impractical in real-world detection where the ground-truth $\mathbf{z}_0$ is unavailable.

**GLW.** We set the number of "green list" intervals to $m = 5$. Since GLW was originally designed for tabular data with continuous density functions, we introduce a minor modification to extend it to mixed-type tabular data. Specifically, for entries with non-zero decimal components, we directly apply the standard method proposed in (He et al., 2024). For integer entries with non-zero units digits, we shift the decimal point one place to the left, apply the method to the transformed values, and then shift them back. To prevent significant distortion, we exclude entries with absolute values less than 1 from watermarking. GLW is applied to all numerical columns across all datasets.

**TabularMark.** Following the original experimental setup, we select the first numerical column as the watermark attribute, from which 10% of the cells are pseudorandomly chosen as key cells. The perturbation range $p$ is set to 25, and the number of unit domains $k$ to 500. To implement the matching algorithm in (Zheng et al., 2024), we extract the first five binary bits from each of five randomly selected attributes and concatenate them to form a 25-bit binary string, which serves as the primary key for each tuple.

### F.5 Implementation Details for Tabular Attacks

We implement the ten post-processing attacks for Table 3 as below:

1. **Row Del.** removes 10% of rows in a table.
2. **Col Del.** replaces 2 columns with unwatermarked values sampled from the same model.
3. **Cell Del.** replaces 10% cells with unwatermarked values sampled from the same model.
4. **G(aussian)-Noise.** adds Gaussian noise with zero mean and a standard deviation equal to 10% of each cell's value for numerical attributes.
5. **C(ategorical)-Noise.** perturbs categorical entries by randomly replacing 10% of cells with values sampled from other rows in the same column.
6. **A(daptive)-Noise.** adds Gaussian noise with zero mean and 0.1 standard deviation to standardized attributes. Specifically, we conduct the process below for each column $j \in \{1, \ldots, p\}$:

   $$z_{ij} = \frac{x_{ij} - \mu_j}{\sigma_j}, \quad z'_{ij} = z_{ij} + \epsilon \cdot \mathcal{N}(0, 1), \quad x'_{ij} = z'_{ij} \cdot \sigma_j + \mu_j,$$

   where $\epsilon = 0.1$ is the attack strength, and $\mu_j$ and $\sigma_j$ are the empirical mean and standard deviation of column $j$ in the original data. Conducting round and clip if $x'_{ij}$ is not in the valid range of column $j$ in the original data.
7. **Truncation.** truncates numerical values at the first significant digit.
8. **Quantization.** discretizes numerical columns using quantile transformation with the 10 quantile bins and maps those discrete quantile levels back to the original data domain with the inverse transform.
9. **Resample.** redistributes samples to achieve equal representation across target classes by super-sampling underrepresented classes and sub-sampling overrepresented ones.
10. **Shuffle.** randomly permutes all rows of the table.

Table 9: Data fidelity and watermark detectability evaluated on tables generated by original TabSyn implementation. No watermarking is denoted as "W/O". Our proposed TAB-DRW uses $(\gamma, \delta) = (0.5, 0.5)$. Fidelity metrics are averaged over 10 trials while Z-score is averaged over 100 trials.

| Datasets | Method | Fidelity Metric | | | | Z-score | |
|---|---|---|---|---|---|---|---|
| | | **Density ↑** | **Corr ↑** | **C2ST ↑** | **MLE ↑** | **1k rows ↑** | **5k rows ↑** |
| Adult | W/O | 0.993±0.001 | 0.982±0.003 | 0.994±0.001 | 0.912±0.002 | – | – |
| | **TAB-DRW** | 0.981±0.004 | 0.967±0.003 | 0.988±0.006 | 0.910±0.003 | 12.57±1.16 | 28.07±0.99 |
| Magic | W/O | 0.990±0.001 | 0.991±0.003 | 0.993±0.002 | 0.936±0.002 | – | – |
| | **TAB-DRW** | 0.983±0.003 | 0.978±0.003 | 0.991±0.004 | 0.933±0.002 | 27.11±0.77 | 61.02±0.82 |
| Shoppers | W/O | 0.985±0.003 | 0.973±0.002 | 0.964±0.003 | 0.920±0.005 | – | – |
| | **TAB-DRW** | 0.976±0.005 | 0.955±0.003 | 0.953±0.007 | 0.919±0.006 | 17.30±1.02 | 39.31±1.06 |
| Default | W/O | 0.987±0.001 | 0.952±0.001 | 0.975±0.002 | 0.764±0.004 | – | – |
| | **TAB-DRW** | 0.982±0.002 | 0.948±0.004 | 0.971±0.009 | 0.764±0.005 | 16.87±0.91 | 37.73±0.88 |
| Drybean | W/O | 0.987±0.002 | 0.992±0.003 | 0.978±0.003 | 0.911±0.006 | – | – |
| | **TAB-DRW** | 0.984±0.004 | 0.977±0.007 | 0.972±0.006 | 0.908±0.009 | 41.23±0.98 | 90.33±1.06 |

While additive Gaussian noise (G-noise) attack adopted in our paper may destroy semantic meanings of columns with large absolute values and very small variance, we adhere to it for two main reasons. First, this ensures a fair comparison of robustness with TabWak$^*$ by replicating the evaluation setup used in its original paper (Zhu et al., 2025). Second, if our watermark signal remains highly detectable under such strong attacks that may significantly distort data utility, it is reasonable to expect stronger performance under milder perturbations. In fact, adaptive Gaussian noise (A-Noise) attack is implemented as a milder variant of Gaussian noise attack.

Additionally, although all the watermark methods including ours show great robustness to the shuffling attack, we believe that the shuffle attack should not be omitted. In practice, the existence of this cost-free row-level attack has important implications for our design. Without shuffle attack, we could use the record indices as hash seeds to deterministically sample pseudorandom bit sequences for each row, enabling accurate recovery during detection and avoiding key collision. However, the bit sequence recovery becomes vulnerable when row-ordering is no longer preserved under shuffle attack.

We do not include the column-level shuffle attack, since table owners or model providers can easily recover the original column order using headers, statistical properties, or semantic features of the columns before watermark detection.

In Appendix G.3, we present extended robustness evaluations against above attacks with higher strength.

# G ADDITIONAL RESULTS AND ANALYSIS

## G.1 ABLATION STUDY

**Model-agnostic property.** Table 9 presents the evaluation results of TAB-DRW on TabSyn implemented using the official codebase. The empirical results show the effectiveness of our method on high-fidelity synthetic data, further justifying our claim that TAB-DRW is practical and the fidelity-detectability trade-off only relies on the hyperparameters.

While our experiments are conducted using TabSyn framework as the generative backbone, we expect TAB-DRW to exhibit similar performance when applied to other synthetic tabular data generators, since TAB-DRW is a post-editing watermarking method that operates independently of the generative model's architecture or sampling procedure.

To justify this claim, we perform evaluations using two additional popular tabular data generators: TabDDPM (Kotelnikov et al., 2023) and STaSy (Kim et al., 2023). The results are presented in Table 10. Overall, our method achieves great fidelity-detectability trade-off across all three models (including TabSyn), demonstrating its effectiveness and model-agnostic property.

Table 10: Data fidelity and watermark detectability evaluated on tables generated by TabDDPM and STaSy. For fiedelity metrics, the first value in each entry denotes the result without watermark, while the second value denotes the result of TAB-DRW with $(\gamma, \delta) = (0.5, 0.5)$.

| Datasets | Model | Fidelity Metric | | | | Z-score | |
|---|---|---|---|---|---|---|---|
| | | Density ↑ | Corr ↑ | C2ST ↑ | MLE ↑ | 1k rows ↑ | 5k rows ↑ |
| Adult | TabDDPM | 0.982/0.967 | 0.969/0.958 | 0.975/0.973 | 0.903/0.894 | 12.44±0.96 | 29.07±1.05 |
| | STaSy | 0.887/0.883 | 0.864/0.858 | 0.408/0.423 | 0.901/0.893 | 13.07±1.22 | 29.95±1.19 |
| Magic | TabDDPM | 0.989/0.971 | 0.983/0.975 | 0.999/0.996 | 0.935/0.923 | 28.74±0.98 | 62.53±1.14 |
| | STaSy | 0.937/0.927 | 0.933/0.919 | 0.694/0.688 | 0.933/0.926 | 27.95±0.89 | 61.48±0.97 |
| Shoppers | TabDDPM | 0.972/0.959 | 0.933/0.921 | 0.834/0.832 | 0.918/0.911 | 17.94±1.24 | 39.27±1.22 |
| | STaSy | 0.906/0.898 | 0.915/0.907 | 0.548/0.553 | 0.914/0.908 | 17.49±1.06 | 38.52±1.17 |
| Default | TabDDPM | 0.985/0.982 | 0.951/0.948 | 0.971/0.967 | 0.756/0.755 | 15.27±0.94 | 35.29±0.92 |
| | STaSy | 0.942/0.940 | 0.940/0.939 | 0.681/0.677 | 0.752/0.749 | 16.65±1.00 | 37.02±1.14 |
| Drybean | TabDDPM | 0.987/0.984 | 0.971/0.960 | 0.967/0.954 | 0.892/0.894 | 40.22±1.16 | 88.59±0.94 |
| | STaSy | 0.949/0.947 | 0.919/0.912 | 0.582/0.596 | 0.890/0.891 | 39.47±1.05 | 86.98±0.91 |

Table 11: Ablation study on YJT. Both methods are applied to TAB-DRW with $(\gamma, \delta) = (0.5, 0.5)$. Fidelity metrics are averaged over 10 trials while Z-score is averaged over 100 trials.

| Datasets | Method | Fidelity Metric | | | | Z-score | |
|---|---|---|---|---|---|---|---|
| | | Density ↑ | Corr ↑ | C2ST ↑ | MLE ↑ | 1k rows ↑ | 5k rows ↑ |
| Adult | W/O YJT | 0.906±0.004 | 0.862±0.003 | 0.601±0.006 | 0.812±0.007 | **13.74±0.87** | **31.62±0.95** |
| | W/ YJT | **0.915±0.005** | **0.864±0.004** | **0.604±0.008** | **0.816±0.009** | 12.81±1.17 | 29.55±1.12 |
| Magic | W/O YJT | 0.907±0.004 | **0.936±0.003** | 0.666±0.002 | 0.817±0.011 | 27.17±0.95 | 60.91±0.93 |
| | W/ YJT | **0.910±0.005** | 0.935±0.003 | **0.676±0.009** | **0.818±0.014** | **27.34±0.93** | **61.42±1.02** |
| Shoppers | W/O YJT | 0.896±0.005 | 0.900±0.002 | 0.704±0.009 | 0.888±0.012 | 12.79±0.96 | 28.50±0.98 |
| | W/ YJT | **0.909±0.006** | **0.902±0.003** | **0.712±0.013** | **0.891±0.014** | **18.18±1.28** | **40.74±1.26** |
| Default | W/O YJT | 0.921±0.007 | 0.906±0.008 | 0.689±0.014 | 0.789±0.011 | 10.05±0.98 | 22.73±1.08 |
| | W/ YJT | **0.929±0.010** | **0.907±0.011** | **0.713±0.018** | **0.791±0.013** | **15.98±0.92** | **35.84±0.91** |
| Drybean | W/O YJT | 0.923±0.009 | 0.911±0.004 | 0.527±0.016 | 0.877±0.014 | 28.12±0.87 | 62.80±0.79 |
| | W/ YJT | **0.931±0.013** | **0.928±0.007** | **0.655±0.029** | **0.880±0.019** | **38.03±1.03** | **85.05±0.67** |

**YJT selection.** In TAB-DRW, the Yeo-Johnson transformation (YJT) serves as a pre-conditioner for constructing the frequency-domain representation. By mapping each feature toward a more Gaussian-like distribution, YJT helps to standardize heterogeneous feature scales and reduce skewness, which is essential for enabling a stable, low-distortion watermarking process in the frequency domain.

We further emphasize that YJT also improves the robustness of rank-based statistics used in our pseudorandom bit generation procedure. Specifically, transforming the feature distributions makes ranks more evenly spread and less sensitive to local density variations, which in turn improves bit consistency under post-processing perturbations.

To support this claim empirically, we include an ablation study comparing TAB-DRW with and without YJT. As shown in the Table 11, the YJT consistently yields a better trade-off between fidelity and watermark detectability, confirming its importance in our design.

**Gray code selection.** We provide additional experimental results using a 1-Gray code and compare it to our adopted variant of 2-Gray code. Specifically, we modify lines 7 and 9 in Algorithm 3 as follows: We change line 7 to "**for** $j \leftarrow 1$ **to** $m$ **do**" and line 9 to "Append 1 to **S** if $k\%4 = 0$ or 3; otherwise append 0". We set the attack strengths for robustness evaluation the same as the strengthened version adopted in Appendix G.3.

From the results, we observe that using a 1-Gray code for bit generation yields a slight improvement in data fidelity, as it more closely mimics an ideal bit sampled from a Bernoulli distribution. However, its detectability and robustness decrease on several datasets, especially those dominated by continuous

Table 12: Data fidelity and watermark detectability evaluated on TAB-DRW using different Gray codes for bit generation. Fidelity metrics are averaged over 10 trials while Z-scores are averaged over 100 trials.

| Datasets | Bit Gen. | Fidelity Metric | | | | Z-score | |
|---|---|---|---|---|---|---|---|
| | | Density ↑ | Corr ↑ | C2ST ↑ | MLE ↑ | 1k rows ↑ | 5k rows ↑ |
| Adult | 1-Gray code | **0.916±0.005** | 0.863±0.005 | 0.600±0.009 | **0.816±0.008** | 11.06±0.99 | 24.95±0.86 |
| | 2-Gray code | 0.915±0.005 | **0.864±0.004** | **0.604±0.008** | 0.816±0.009 | **12.81±1.17** | **29.55±1.12** |
| Magic | 1-Gray code | 0.917±0.006 | 0.936±0.003 | **0.676±0.008** | 0.818±0.014 | 21.74±0.96 | 48.78±0.97 |
| | 2-Gray code | **0.917±0.005** | **0.937±0.003** | 0.676±0.009 | **0.818±0.014** | **27.34±0.93** | **61.42±1.02** |
| Shoppers | 1-Gray code | 0.909±0.006 | **0.909±0.004** | **0.715±0.011** | **0.892±0.014** | 17.30±1.14 | 38.48±1.14 |
| | 2-Gray code | **0.909±0.006** | 0.902±0.003 | 0.712±0.013 | 0.891±0.014 | **18.18±1.28** | **40.74±1.26** |
| Default | 1-Gray code | 0.927±0.010 | **0.918±0.009** | 0.715±0.013 | **0.793±0.013** | 15.07±1.02 | 33.72±0.97 |
| | 2-Gray code | **0.929±0.010** | 0.907±0.011 | **0.717±0.018** | 0.791±0.013 | **15.98±0.92** | **35.84±0.91** |
| Drybean | 1-Gray code | **0.933±0.010** | **0.932±0.007** | **0.659±0.022** | **0.880±0.019** | 25.40±0.92 | 57.74±1.03 |
| | 2-Gray code | 0.931±0.013 | 0.928±0.007 | 0.655±0.029 | 0.880±0.019 | **38.03±1.03** | **85.05±0.67** |

Table 13: Robustness evaluation of TAB-DRW using different Gray codes for bit generation under the strengthened attack setting. Z-scores are averaged over 100 trials on tables with 5k rows.

| Datasets | Bit Gen. | Attacks | | | | | | | | | |
|---|---|---|---|---|---|---|---|---|---|---|---|
| | | Row Del. | Col Del. | Cell Del. | G-Noise | C-Noise | A-Noise | Truncation | Quantization | Resample | Shuffle |
| Adult | 1-Gray code | 22.30 | 10.10 | 11.19 | 12.32 | 18.11 | 21.24 | 24.95 | 16.04 | 21.49 | 24.95 |
| | 2-Gray code | **26.34** | **13.12** | **14.37** | **14.29** | **20.10** | **21.85** | **29.55** | **16.41** | **28.15** | **29.55** |
| Magic | 1-Gray code | 43.47 | 8.46 | 16.27 | 22.81 | 44.14 | 21.42 | 38.83 | 19.32 | 24.58 | 48.78 |
| | 2-Gray code | **54.85** | **11.75** | **21.60** | **33.93** | **48.38** | **29.06** | **52.62** | **26.43** | **37.61** | **61.42** |
| Shoppers | 1-Gray code | 34.36 | 13.40 | **14.00** | 35.54 | 25.05 | **13.12** | **31.00** | 23.36 | 14.34 | 38.48 |
| | 2-Gray code | **36.21** | **13.75** | 13.27 | **37.71** | **32.60** | 13.10 | 30.28 | **25.72** | **29.28** | **40.74** |
| Default | 1-Gray code | 29.84 | 19.45 | 15.05 | 21.66 | 26.14 | 12.52 | 33.72 | 11.06 | 30.08 | 33.72 |
| | 2-Gray code | **31.92** | **20.70** | **15.44** | **23.75** | **27.20** | **16.99** | **35.84** | **14.10** | **32.36** | **35.84** |
| Drybean | 1-Gray code | 51.38 | 15.48 | 16.87 | 12.73 | 48.93 | 15.50 | 22.94 | 15.31 | 39.44 | 57.74 |
| | 2-Gray code | **75.91** | **32.92** | **35.27** | **23.57** | **77.70** | **42.48** | **42.14** | **45.95** | **68.69** | **85.05** |

variables such as **Magic** and **Drybean**. We attribute this to the finer-grained rank-bin partition induced by the 1-Gray code and to the greater instability of the sum-based score for continuous features under perturbations, which makes cross-bin rank shifts more likely to happen.

**Column selection for watermarking.** In the main paper, our empirical evaluation focuses on numerical columns (including mixed continuous and discrete types) to enable a fair comparison with the other post-editing watermarking methods, GLW and TabularMark, which both suffer substantial fidelity degradation when applied to all columns. Here we also report results obtained by applying TAB-DRW to all columns, showing how different selection strategies influence the tradeoff between fidelity and detectability. In general, using more columns for watermarking improves robustness (Theorem 2 shows that the lower bound of the Z-score scales with the number of selected columns), while incurring slightly higher distortion.

The results in Table 14 show that watermarking more columns improves detectability while reducing fidelity, consistent with our theoretical analysis.

**Impact of rounding and clipping on watermark detectability.** Since the outputs of the inverse DFT and YJT are real-valued, rounding and clipping are necessary for discrete features to preserve semantic validity. However, these operations may also perturb the sign-bit alignment in the frequency domain, potentially weakening the watermark signal. Fortunately, the sign-bit alignment of TAB-DRW is highly insensitive to such mild nonlinear perturbations. In addition, because our method preserves fidelity well under appropriate choices of $(\gamma, \delta)$, clipping occurs only rarely and rounding magnitudes remain minimal.

Table 15 shows the results of an ablation study comparing Z-scores with and without rounding and clipping across five datasets, together with the frequency and magnitude of these operations. For **Magic** dataset there are no rounding or clipping happening since all the columns are continuous.

Table 14: Data fidelity and watermark detectability evaluated on TAB-DRW with different columns selected for watermarking. Fidelity metrics are averaged over 10 trials, and Z-scores are averaged over 100 trials.

| Datasets | Col. Selection | Fidelity Metric | | | | Z-score | |
|---|---|---|---|---|---|---|---|
| | | Density ↑ | Corr ↑ | C2ST ↑ | MLE ↑ | 1k rows ↑ | 5k rows ↑ |
| Adult | All Col. | 0.909 ± 0.005 | 0.859 ± 0.005 | 0.597 ± 0.009 | 0.808 ± 0.008 | **14.56 ± 0.99** | **32.89 ± 1.06** |
| | Original | **0.915 ± 0.005** | **0.864 ± 0.004** | **0.604 ± 0.008** | **0.816 ± 0.009** | 12.81 ± 1.17 | 29.55 ± 1.12 |
| Magic | All Col. | 0.914 ± 0.006 | 0.936 ± 0.003 | 0.674 ± 0.008 | 0.818 ± 0.014 | 24.66 ± 1.08 | 55.47 ± 1.09 |
| | Original | **0.917 ± 0.005** | **0.937 ± 0.003** | **0.676 ± 0.009** | **0.818 ± 0.014** | **27.34 ± 0.93** | **61.42 ± 1.02** |
| Shoppers | All Col. | 0.901 ± 0.005 | 0.897 ± 0.003 | 0.704 ± 0.009 | 0.887 ± 0.012 | **19.59 ± 1.08** | **43.84 ± 1.14** |
| | Original | **0.909 ± 0.006** | **0.902 ± 0.003** | **0.712 ± 0.013** | **0.891 ± 0.014** | 18.18 ± 1.28 | 40.74 ± 1.26 |
| Default | All Col. | 0.919 ± 0.009 | 0.902 ± 0.013 | 0.705 ± 0.019 | 0.787 ± 0.011 | **22.21 ± 1.03** | **49.96 ± 0.99** |
| | Original | **0.929 ± 0.010** | **0.907 ± 0.011** | **0.717 ± 0.018** | **0.791 ± 0.013** | 15.98 ± 0.92 | 35.84 ± 0.91 |
| Drybean | All Col. | 0.929 ± 0.010 | 0.924 ± 0.007 | 0.649 ± 0.022 | 0.878 ± 0.019 | **38.35 ± 0.89** | **85.47 ± 0.73** |
| | Original | **0.931 ± 0.013** | **0.928 ± 0.007** | **0.655 ± 0.029** | **0.880 ± 0.019** | 38.03 ± 1.03 | 85.05 ± 0.67 |

Table 15: Detection performance of TAB-DRW with or without the rounding and clipping operations. Z-scores are averaged over 100 trials on tables with 1k rows. "Rounding magnitude" denotes the average rounding magnitude of discrete entries, and "Clipping ratio" denotes the fraction of discrete entries that are clipped.

| Dataset | W/O round and clip | W/ round and clip | Rounding magnitude | Clipping ratio |
|---|---|---|---|---|
| Adult | $15.21 \pm 1.00$ | $12.81 \pm 1.17$ | $0.0911 \pm 0.0015$ | $0.0008 \pm 0.0004$ |
| Magic | $27.34 \pm 0.93$ | $27.34 \pm 0.93$ | $0.0000 \pm 0.0000$ | $0.0000 \pm 0.0000$ |
| Shoppers | $21.00 \pm 1.15$ | $18.18 \pm 1.28$ | $0.0969 \pm 0.0042$ | $0.0244 \pm 0.0036$ |
| Default | $17.94 \pm 0.95$ | $15.98 \pm 0.92$ | $0.0542 \pm 0.0018$ | $0.0151 \pm 0.0013$ |
| Drybean | $37.79 \pm 1.02$ | $38.03 \pm 1.03$ | $0.1285 \pm 0.0027$ | $0.0145 \pm 0.0022$ |

For other datasets, the impact of these two post-processing operations on watermark detectability is negligible.

### G.2 RUNTIME EVALUATION

Tables 16 and 17 report the average runtimes for watermark embedding and detection on five benchmark datasets. Each result is averaged over 100 independent trials on synthetic tabular data with 1K rows. We run the experiments on a M1 Pro CPU and a 40GB NVIDIA A100 GPU.

Among all compared methods, MUSE incurs the highest embedding cost, as it selects the highest-scoring row from multiple candidates, thereby requiring the generation of multiple unwatermarked samples. By contrast, TabWak* imposes the highest detection cost, since it embeds watermarks in the latent space of a large diffusion model and relies on the DDIM inversion process for detection, which demands GPU resources for tensor acceleration. In contrast, post-editing approaches such as GLW, TabularMark, and our proposed TAB-DRW embed and detect watermarks after table generation, without accessing the generative pipeline. These methods are significantly more efficient—achieving detection speeds several orders of magnitude faster than TabWak*—and can be executed entirely on CPU without loss of performance.

As shown in Tables 16, 17, and 18, once unwatermarked samples are generated, the grid search tuning described in Section 2 can be applied to the synthetic data with negligible overhead—on the order of seconds for datasets comparable in scale to our benchmarks—and can be executed entirely on CPU.

Table 16: Comparison of watermark embedding runtimes (in seconds) across methods. For each entry, the first value indicates the total GPU time to generate a 1K-row watermarked table with TabSyn, and the second value denotes the watermark embedding CPU time for an existing 1K-row table.

| **Dataset** | GLW | MUSE | TabWak* | TabularMark | TAB-DRW |
|---|---|---|---|---|---|
| Adult | 1.896(0.031) | 3.878(0.593) | 1.904(–) | 1.896(0.008) | 1.896(0.112) |
| Magic | 1.897(0.026) | 3.963(0.555) | 1.893(–) | 1.897(0.007) | 1.897(0.106) |
| Shoppers | 1.912(0.026) | 3.938(0.629) | 1.932(–) | 1.912(0.008) | 1.912(0.142) |
| Default | 1.925(0.052) | 4.114(0.688) | 1.945(–) | 1.925(0.009) | 1.925(0.205) |
| Drybean | 1.920(0.025) | 3.967(0.601) | 1.936(–) | 1.920(0.009) | 1.920(0.152) |

Table 17: Comparison of watermark detection runtimes (in seconds) on 1K-row watermarked table across watermarking methods. Values for TabWak* denote GPU runtimes, whereas all other methods are measured in CPU time.

| **Dataset** | GLW | MUSE | TabWak* | TabularMark | TAB-DRW |
|---|---|---|---|---|---|
| Adult | 0.003 | 0.177 | 27.96 | 0.717 | 0.100 |
| Magic | 0.001 | 0.161 | 21.78 | 0.729 | 0.076 |
| Shoppers | 0.005 | 0.188 | 30.27 | 0.682 | 0.106 |
| Default | 0.004 | 0.234 | 35.49 | 0.713 | 0.152 |
| Drybean | 0.003 | 0.193 | 26.03 | 0.582 | 0.120 |

Table 18: Comparison of watermark detection runtimes (in seconds) on 100K-row watermarked table across watermarking methods. Values for TabWak* denote GPU runtimes, whereas all other methods are measured in CPU time.

| **Dataset** | GLW | MUSE | TabWak* | TabularMark | TAB-DRW |
|---|---|---|---|---|---|
| Adult | 0.20 | 14.61 | 1808.60 | 80.69 | 2.05 |
| Magic | 0.10 | 13.06 | 1472.28 | 71.89 | 2.02 |
| Shoppers | 0.31 | 15.76 | 1659.47 | 89.42 | 3.12 |
| Default | 0.38 | 17.91 | 1994.51 | 87.85 | 3.88 |
| Drybean | 0.15 | 16.63 | 1569.88 | 82.37 | 2.87 |

### G.3 ADDITIONAL ROBUSTNESS EVALUATION

**Post-processing attaks with high strength.** In this section, we benchmark the robustness of TAB-DRW and other watermarking methods using attacks with higher strength. Specifically, we use the setting below:

1. **Row Del.** removes 20% of rows in a table.
2. **Col Del.** replaces 3 columns with unwatermarked values sampled from the same model.
3. **Cell Del.** replaces 20% cells with unwatermarked values sampled from the same model.
4. **G(aussian)-Noise.** adds Gaussian noise with zero mean and a standard deviation equal to 20% of each cell's value for numerical attributes.
5. **C(ategorical)-Noise.** perturbs categorical entries by randomly replacing 20% of cells with values sampled from other rows in the same column.
6. **A(daptive)-Noise.** adds Gaussian noise with zero mean and 0.2 standard deviation to standardized attributes.

Table 19: Watermark robustness against attacks with higher strength. Average Z-score on 5k rows under seven variable-strength attacks. Each value is obtained by repeating the attacks 100 times (10 times for "TabWak*") and averaging the results. Our proposed TAB-DRW is evaluated with the hyperparameter $(\gamma, \delta) = (0.5, 0.5)$. Best performances are shown in **bold**, and second-best are underlined.

| Datasets | Method | Attacks | | | | | | |
|---|---|---|---|---|---|---|---|---|
| | | Row Del. | Col Del. | Cell Del. | G-Noise | C-Noise | A-Noise | Quantization |
| | | 20% | 3 col | 20% | 20% | 20% | 20% | 20% |
| Adult | GLW | 14.76 | 13.10 | 13.19 | 0.00 | 16.54 | 2.77 | 3.03 |
| | MUSE | 13.31 | 4.96 | 6.17 | 11.84 | 8.05 | 3.99 | 10.99 |
| | TabWak* | 14.44 | 8.05 | 7.87 | 0.02 | 15.67 | 10.23 | 5.56 |
| | TabularMark | 20.29 | **13.92** | 12.99 | 3.31 | 5.54 | 0.62 | 0.00 |
| | **TAB-DRW** | **26.34** | 13.12 | **14.37** | **14.29** | **20.10** | **21.85** | **16.41** |
| Magic | GLW | **153.98** | **123.60** | **137.64** | 0.10 | **172.20** | 0.31 | 14.08 |
| | MUSE | 31.56 | 3.70 | 9.30 | 8.39 | 33.34 | 4.06 | 0.39 |
| | TabWak* | 17.27 | 7.47 | 13.45 | 16.44 | 19.76 | 13.39 | 12.86 |
| | TabularMark | 16.09 | 10.68 | 11.74 | 0.00 | 19.39 | 0.68 | 0.00 |
| | **TAB-DRW** | 54.85 | 11.75 | 21.60 | **33.93** | 48.38 | **29.06** | **26.43** |
| Shoppers | GLW | **36.82** | **34.46** | **32.33** | 0.00 | **39.08** | 1.13 | 0.00 |
| | MUSE | 25.85 | 8.64 | 9.05 | 21.58 | 16.20 | **16.26** | 13.61 |
| | TabWak* | 8.93 | 2.26 | 0.97 | 0.00 | 10.47 | 1.22 | 0.69 |
| | TabularMark | 13.68 | 8.74 | 10.13 | 0.98 | 13.29 | 0.00 | 1.42 |
| | **TAB-DRW** | 36.21 | 13.75 | 13.27 | **37.71** | 32.60 | 13.10 | **25.72** |
| Default | GLW | 25.67 | 19.89 | **19.88** | 0.00 | 27.08 | 6.49 | 9.60 |
| | MUSE | 30.80 | 8.52 | 7.30 | 14.01 | 17.67 | 3.79 | 4.97 |
| | TabWak* | 21.96 | 12.84 | 13.49 | **23.77** | 23.70 | **18.52** | **20.25** |
| | TabularMark | 19.72 | 16.21 | 11.17 | 0.00 | 17.10 | 0.80 | 2.58 |
| | **TAB-DRW** | **31.92** | **20.70** | 15.44 | 23.75 | **27.20** | 16.99 | 14.10 |
| Drybean | GLW | **116.96** | **104.46** | **98.54** | 0.18 | **123.28** | 5.05 | 27.90 |
| | MUSE | 28.12 | 4.58 | 6.34 | 6.13 | 27.71 | 2.85 | 0.00 |
| | TabWak* | 16.01 | 0.00 | 0.00 | 11.21 | 17.53 | 10.42 | 3.43 |
| | TabularMark | 12.06 | 5.27 | 3.22 | 0.00 | 13.54 | 2.43 | 0.00 |
| | **TAB-DRW** | 75.91 | 32.92 | 35.27 | **23.57** | 77.70 | **42.48** | **45.95** |

7. **Quantization.** discretizes numerical columns using quantile transformation with the 10 quantile bins and maps those discrete quantile levels back to the original data domain with the inverse transform.

Since the **Truncation**, **Resample**, and **Shuffle** attacks are applied with fixed strength, we omit them here.

Table 19 reports the average one-sided $Z$-score over 5k rows, evaluated under the enhanced attacks. Our watermarking method still demonstrates superior robustness across all attack types and datasets, ranking either first or second. Figure 7 shows TPR@0.1%FPR versus the number of rows under three representative and strong attacks with higher strength setting. Among eight out of fifteen cases, our method reaches 1.0 TPR@0.1%FPR using only 400 rows, with the remaining seven requiring fewer than 1K rows. In contrast, baseline methods often suffer reduced true positive rates or completely lose detectability under these conditions.

To provide a more comprehensive view of robustness, we include additional empirical results under row deletion and column deletion attacks with varying deletion strengths in Figure 8. The results show that our method ranks first or second across most attack levels, demonstrating strong resilience even under high-strength attacks.

**Robustness to adaptive attacks.** Table 20 shows the fidelity degradation under rewatermarking attacks of varying rounds, serving as an extension to Table 4.

We also clarify the robustness of TAB-DRW under spoofing attacks mentioned in Ngo et al. (2024). In our setting, spoofing attack refers to making unwatermarked data showcase strong watermark signal

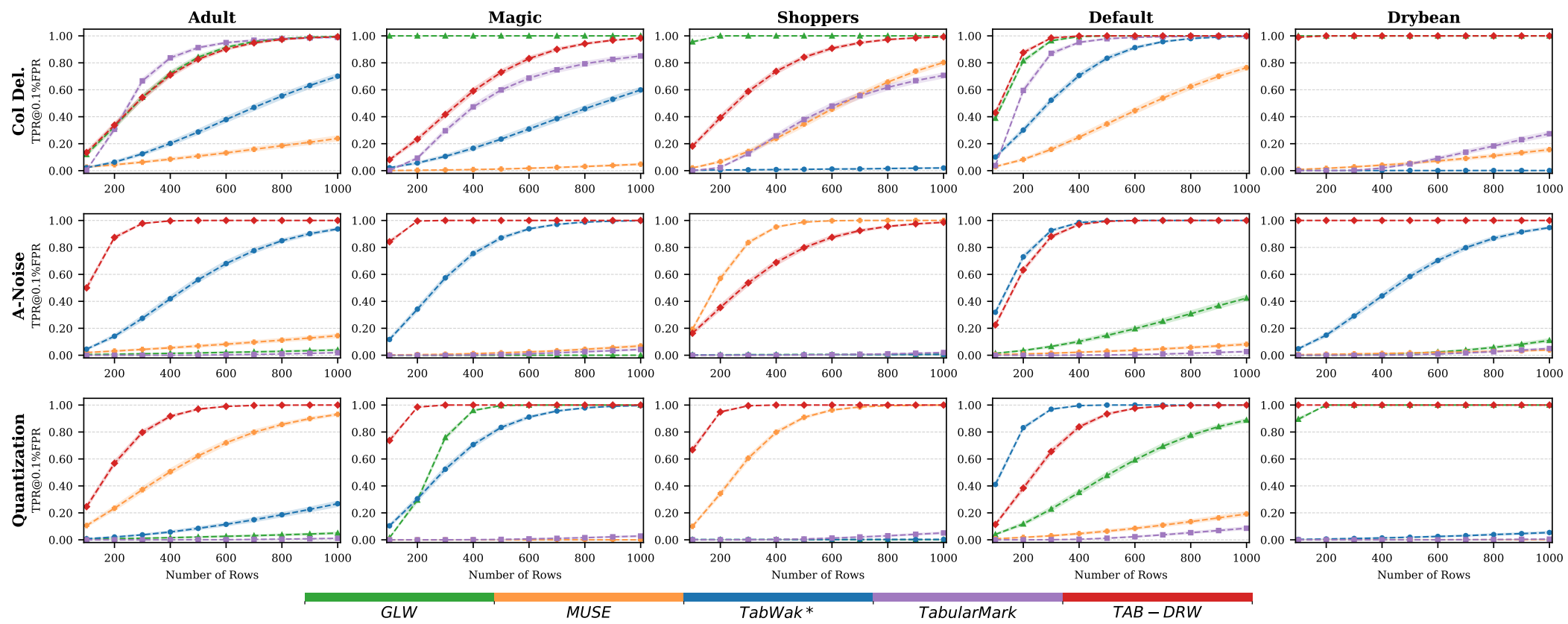

Figure 7: TPR@0.1%FPR versus row count under three representative attacks with higher strength. Dashed lines show the bootstrap mean estimate (500 resamples), and shaded regions indicate the 90% confidence interval.

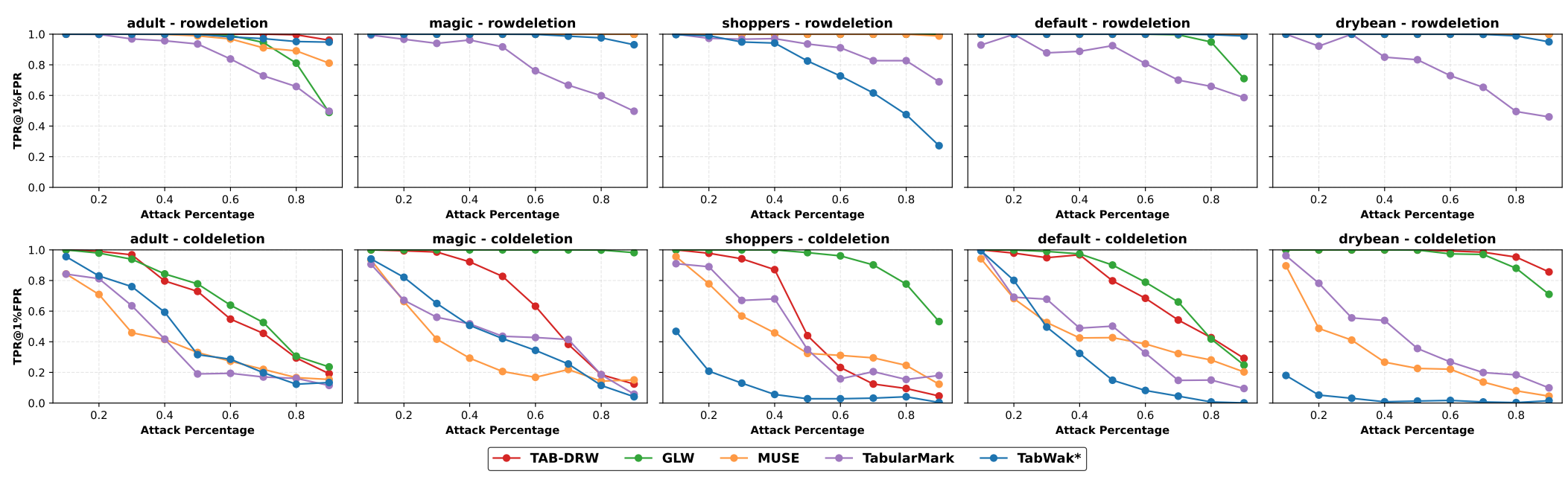

Figure 8: TPR@1%FPR versus attack strength under row and column deletion attacks. All experiments are conducted on tables with 1K rows. Each value denotes the result of 100 independent trials.

Table 20: Robustness of TAB-DRW against rewatermarking attacks of varying strength. Fidelity is averaged over four metrics across 10 independent trials, and the Z-scores are computed on tables with 5k rows and averaged over 100 independent trials. "Rewatermarking@$n$" denotes rewatermarking the table using $n$ randomly sampled keys.

| Datasets | No-attack | | Rewatermarking@1 | | Rewatermarking@3 | | Rewatermarking@10 | |
|---|---|---|---|---|---|---|---|---|
| | Fidelity | Z-score | Fidelity | Z-score | Fidelity | Z-score | Fidelity | Z-score |
| Adult | $0.799 \pm 0.006$ | $29.55 \pm 1.12$ | $0.787 \pm 0.008$ | $23.66 \pm 1.17$ | $0.772 \pm 0.006$ | $16.26 \pm 1.09$ | $0.766 \pm 0.009$ | $17.26 \pm 1.34$ |
| Magic | $0.837 \pm 0.008$ | $61.42 \pm 1.02$ | $0.822 \pm 0.008$ | $53.23 \pm 0.91$ | $0.813 \pm 0.007$ | $34.32 \pm 0.93$ | $0.799 \pm 0.008$ | $29.17 \pm 1.00$ |
| Shoppers | $0.854 \pm 0.009$ | $40.74 \pm 1.26$ | $0.847 \pm 0.008$ | $31.97 \pm 1.15$ | $0.829 \pm 0.009$ | $20.14 \pm 1.09$ | $0.813 \pm 0.008$ | $16.67 \pm 1.09$ |
| Default | $0.836 \pm 0.013$ | $35.84 \pm 0.91$ | $0.827 \pm 0.011$ | $32.85 \pm 1.00$ | $0.811 \pm 0.013$ | $19.40 \pm 1.07$ | $0.809 \pm 0.010$ | $26.28 \pm 1.18$ |
| Drybean | $0.849 \pm 0.017$ | $85.05 \pm 0.67$ | $0.832 \pm 0.017$ | $44.79 \pm 0.81$ | $0.801 \pm 0.017$ | $29.47 \pm 0.83$ | $0.806 \pm 0.014$ | $33.77 \pm 0.95$ |

Table 21: Robustness of TAB-DRW against distilling-based spoofing attack. Entry entry denotes the result over 100 independent trials.

| Datasets | 1k rows | | 5k rows | |
|---|---|---|---|---|
| | Z-score | TPR@0.1%FPR | Z-score | TPR@0.1%FPR |
| Adult | 0.88±1.03 | 0.01 | 1.62±1.04 | 0.05 |
| Magic | 1.87±1.12 | 0.15 | 3.75±1.08 | 0.74 |
| Shoppers | 1.21±0.86 | 0.01 | 2.19±0.97 | 0.13 |
| Default | 0.62±0.94 | 0.00 | 1.28±1.06 | 0.03 |
| Drybean | 2.02±0.98 | 0.12 | 4.13±1.25 | 0.79 |

under the detection using a specific key. TAB-DRW is explicitly designed to make this extremely difficult, as detailed in Appendix B. In conclusion, the privacy-enhanced variant applies a key-dependent column permutation before the YJT and DFT, creating a large key space and yielding empirically negligible cross-key collisions (i.e., a watermark embedded with one key cannot be misdetected under another). This arises because the imaginary sign-bit alignments induced by different keys (i.e., different column orders) are approximately unrelated. Since the detection key is private, an adversary without access to it cannot efficiently tune modifications to increase the Z-score; naive or heuristic modifications either fail to spoof the watermark or noticeably degrade data fidelity. Even if attackers imitate the imaginary sign pattern of the frequency-domain representation from a watermarked table, they still cannot produce a detectable watermark signal as long as their keys differ from the specific detection key. We refer the readers to Appendix B and Appendix G.4 for additional analysis and empirical justification.

However, even without access to the watermark key, an adversary could still attempt a model-level spoofing attack by distilling a new generator from watermarked outputs, similar to the spirit of distilling from watermarked LLMs (Gu et al., 2024). To evaluate this threat, we simulate such an attacker as follows: for each of the five datasets, we first sample a large corpus of synthetic tables from a TabSyn model equipped with TAB-DRW, and then train a fresh TabSyn model only on these watermarked samples, using the same architecture and training protocol as the original generator. We then generate tables from the distilled model and test them with our standard detector.

The results in Table 21 indicate that this distilled generator generally fails to successfully spoof the TAB-DRW watermark. For **Adult**, **Shoppers**, and **Default**, tables with 1k rows produced by the distilled model are statistically indistinguishable from unwatermarked data. In contrast, for **Magic** and **Drybean**, we observe a moderate watermark signal. Empirically, datasets dominated by continuous attributes appear more vulnerable to this kind of spoofing by distillation, which we hypothesize is due to the stronger and more smoothly distributed watermark signal embedded in their continuous feature distributions.

### G.4 PRIVACY-ENHANCED TAB-DRW EVALUATION

Table 22: Data fidelity and watermark detectability of privacy-enhanced TAB-DRW under varying watermark keys. All experiments use $(\gamma, \delta) = (0.5, 0.5)$. Fidelity metrics are averaged over 10 trials, and the $Z$-score is averaged over 100 trials.

| Datasets | Key | Fidelity Metric | | | | Z-score | |
|---|---|---|---|---|---|---|---|
| | | Density ↑ | Corr ↑ | C2ST ↑ | MLE ↑ | 1k rows ↑ | 5k rows ↑ |
| Adult | W/O | 0.922±0.001 | 0.872±0.001 | 0.611±0.004 | 0.824±0.005 | – | – |
| | Key 48 | 0.912±0.003 | 0.862±0.003 | 0.598±0.008 | 0.814±0.009 | 11.98±0.97 | 26.48±1.07 |
| | Key 496 | **0.916±0.003** | **0.869±0.004** | 0.601±0.006 | **0.819±0.009** | **16.69±1.24** | **37.39±1.37** |
| | Key 928 | 0.915±0.005 | 0.864±0.004 | **0.604±0.008** | 0.816±0.009 | 12.81±1.17 | 29.55±1.12 |
| Magic | W/O | 0.917±0.001 | 0.945±0.003 | 0.672±0.004 | 0.823±0.007 | – | – |
| | Key 48 | **0.915±0.004** | 0.934±0.005 | **0.676±0.009** | **0.821±0.009** | 24.06±0.75 | 53.19±0.86 |
| | Key 496 | 0.915±0.004 | **0.939±0.005** | 0.666±0.007 | 0.819±0.012 | 27.33±1.04 | 61.17±1.08 |
| | Key 928 | 0.910±0.005 | 0.935±0.003 | 0.676±0.009 | 0.818±0.014 | **27.34±0.93** | **61.42±1.02** |
| Shoppers | W/O | 0.919±0.002 | 0.910±0.001 | 0.704±0.005 | 0.902±0.012 | – | – |
| | Key 48 | **0.912±0.004** | **0.907±0.002** | 0.706±0.008 | **0.893±0.015** | 16.15±1.11 | 36.11±1.16 |
| | Key 496 | 0.904±0.005 | 0.902±0.003 | 0.698±0.011 | 0.889±0.015 | 14.84±1.10 | 34.28±1.16 |
| | Key 928 | 0.909±0.006 | 0.902±0.003 | **0.712±0.013** | 0.891±0.014 | **18.18±1.28** | **40.74±1.26** |
| Default | W/O | 0.930±0.001 | 0.907±0.001 | 0.717±0.003 | 0.797±0.009 | – | – |
| | Key 48 | **0.929±0.010** | 0.906±0.007 | **0.715±0.010** | **0.791±0.013** | 13.56±1.02 | 30.32±0.98 |
| | Key 496 | 0.929±0.010 | **0.907±0.010** | 0.714±0.012 | 0.791±0.013 | 13.82±0.96 | 30.90±0.99 |
| | Key 928 | 0.929±0.010 | 0.907±0.011 | 0.713±0.018 | 0.791±0.013 | **15.98±0.92** | **35.84±0.91** |
| Drybean | W/O | 0.932±0.001 | 0.935±0.001 | 0.640±0.003 | 0.878±0.009 | – | – |
| | Key 48 | 0.930±0.007 | 0.926±0.005 | 0.649±0.014 | **0.881±0.017** | 37.21±0.84 | 83.14±0.91 |
| | Key 496 | 0.930±0.007 | **0.929±0.006** | 0.631±0.025 | 0.875±0.011 | 30.98±0.91 | 70.27±0.83 |
| | Key 928 | **0.931±0.013** | 0.928±0.007 | **0.655±0.029** | 0.880±0.019 | **38.03±1.03** | **85.05±0.67** |

Table 23: Multi-key evaluation on five benchmarks. The randomly selected keys along the horizontal axis are used for sampling, while those along the vertical axis are used for detection: FPR/TPR(diagonal) of 1K independent trials under threshold $q_\alpha = 6$ on 1K rows.

| **Dataset** | **Detection key** | **Sampling keys** | | | | |
|---|---|---|---|---|---|---|
| | | Key 48 | Key 275 | Key 496 | Key 643 | Key 928 |
| Adult | Key 48 | 1.000 | 0.000 | 0.000 | 0.000 | 0.000 |
| | Key 275 | 0.000 | 0.998 | 0.000 | 0.000 | 0.000 |
| | Key 496 | 0.000 | 0.000 | 1.000 | 0.001 | 0.000 |
| | Key 643 | 0.000 | 0.007 | 0.000 | 0.996 | 0.000 |
| | Key 928 | 0.000 | 0.000 | 0.000 | 0.000 | 1.000 |
| Magic | Key 48 | 1.000 | 0.000 | 0.000 | 0.000 | 0.000 |
| | Key 275 | 0.000 | 1.000 | 0.000 | 0.000 | 0.000 |
| | Key 496 | 0.000 | 0.000 | 1.000 | 0.001 | 0.000 |
| | Key 643 | 0.000 | 0.000 | 0.000 | 1.000 | 0.000 |
| | Key 928 | 0.000 | 0.000 | 0.000 | 0.000 | 1.000 |
| Shoppers | Key 48 | 1.000 | 0.000 | 0.000 | 0.000 | 0.000 |
| | Key 275 | 0.000 | 1.000 | 0.000 | 0.000 | 0.000 |
| | Key 496 | 0.000 | 0.000 | 1.000 | 0.000 | 0.000 |
| | Key 643 | 0.000 | 0.000 | 0.000 | 1.000 | 0.000 |
| | Key 928 | 0.000 | 0.000 | 0.000 | 0.000 | 1.000 |
| Default | Key 48 | 1.000 | 0.000 | 0.000 | 0.000 | 0.000 |
| | Key 275 | 0.002 | 1.000 | 0.000 | 0.000 | 0.000 |
| | Key 496 | 0.000 | 0.000 | 1.000 | 0.000 | 0.000 |
| | Key 643 | 0.000 | 0.000 | 0.000 | 1.000 | 0.000 |
| | Key 928 | 0.000 | 0.000 | 0.000 | 0.000 | 1.000 |
| Drybean | Key 48 | 1.000 | 0.000 | 0.000 | 0.000 | 0.000 |
| | Key 275 | 0.000 | 1.000 | 0.000 | 0.000 | 0.000 |
| | Key 496 | 0.000 | 0.000 | 1.000 | 0.000 | 0.000 |
| | Key 643 | 0.000 | 0.000 | 0.000 | 1.000 | 0.004 |
| | Key 928 | 0.000 | 0.000 | 0.000 | 0.000 | 1.000 |

**Data fidelity vs. watermark detectability.** Table 22 shows that privacy-enhanced TAB-DRW achieves consistently high data fidelity and detectability across three randomly sampled keys. Although minor variations exist, they remain within an acceptable range, indicating that users need not devote much effort to tuning the key. Additionally, the empirical results further strengthen our claim that the key-dependent variability in the frequency-domain representation does not substantially affect watermark distortion or detectability.

**Multi-key scenarios.** Under a deployment scenario with multiple watermark key holders, we evaluate potential key collision, i.e., how many different keys $\kappa$ in Algorithm 2 & 3 can be used for a dataset without leading to false positives during detection. In practice, there exists an upper bound on the number of watermark keys that can be supported without introducing elevated false positives. And this capacity is influenced by the number of dataset columns. The cross-key confusion matrices in Table 23 present empirical results on the ability to detect and distinguish between different watermark keys, demonstrating the superiority of our method in avoiding potential key collisions in multi-user scenarios.

## H A CASE STUDY ON LOW-CARDINALITY CATEGORICAL VARIABLE

TAB-DRW handles low-cardinality categorical variables such as "gender" in a conservative and adaptive manner. The row-wise DFT maps each standardized sample into a frequency-domain representation whose components are linear combinations of all entries in the row. As a result, watermark embedding via imaginary sign-bit alignment operates on this joint representation rather

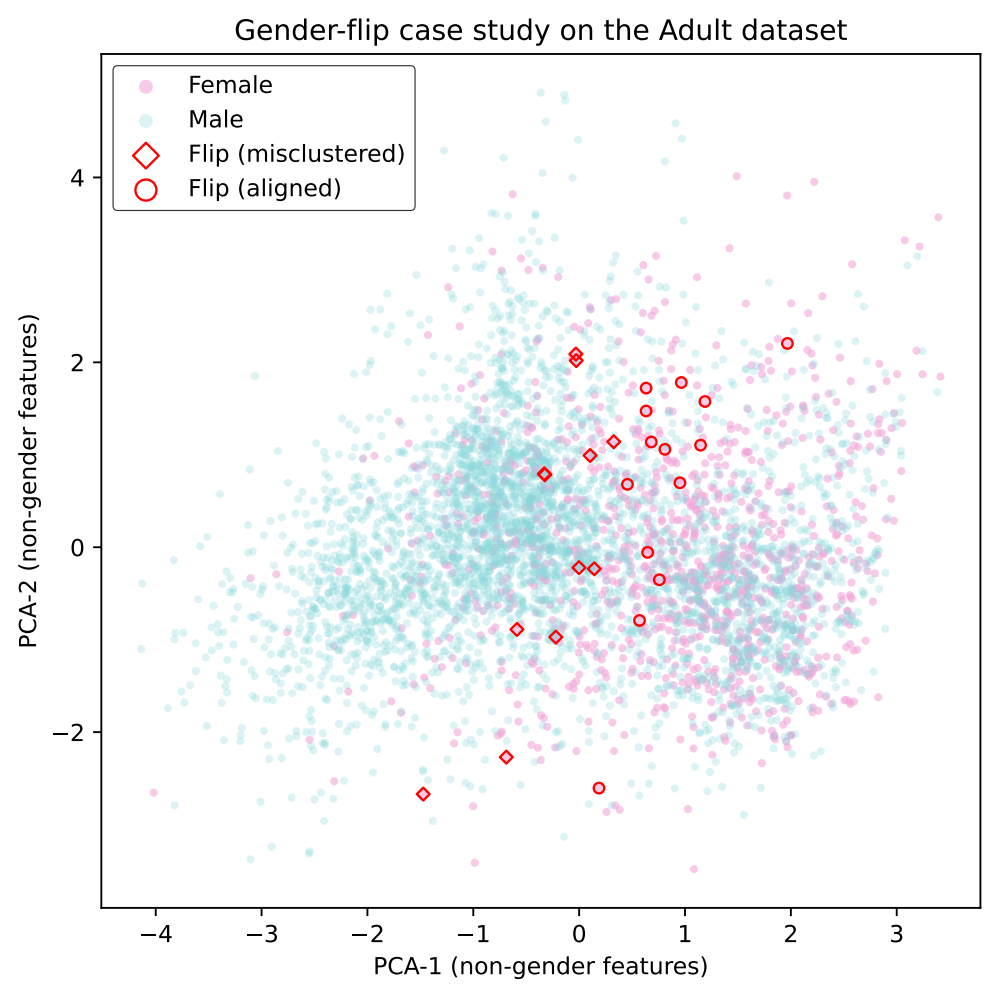

(a)

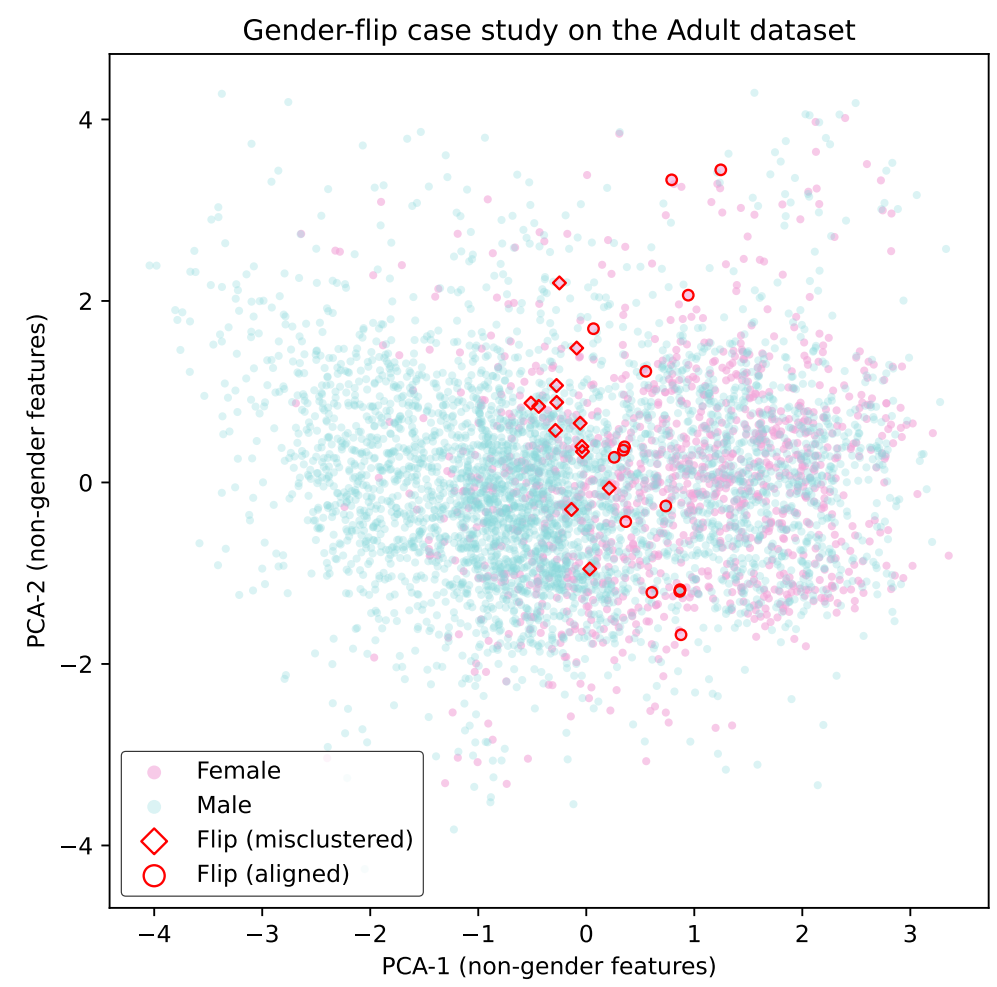

(b)

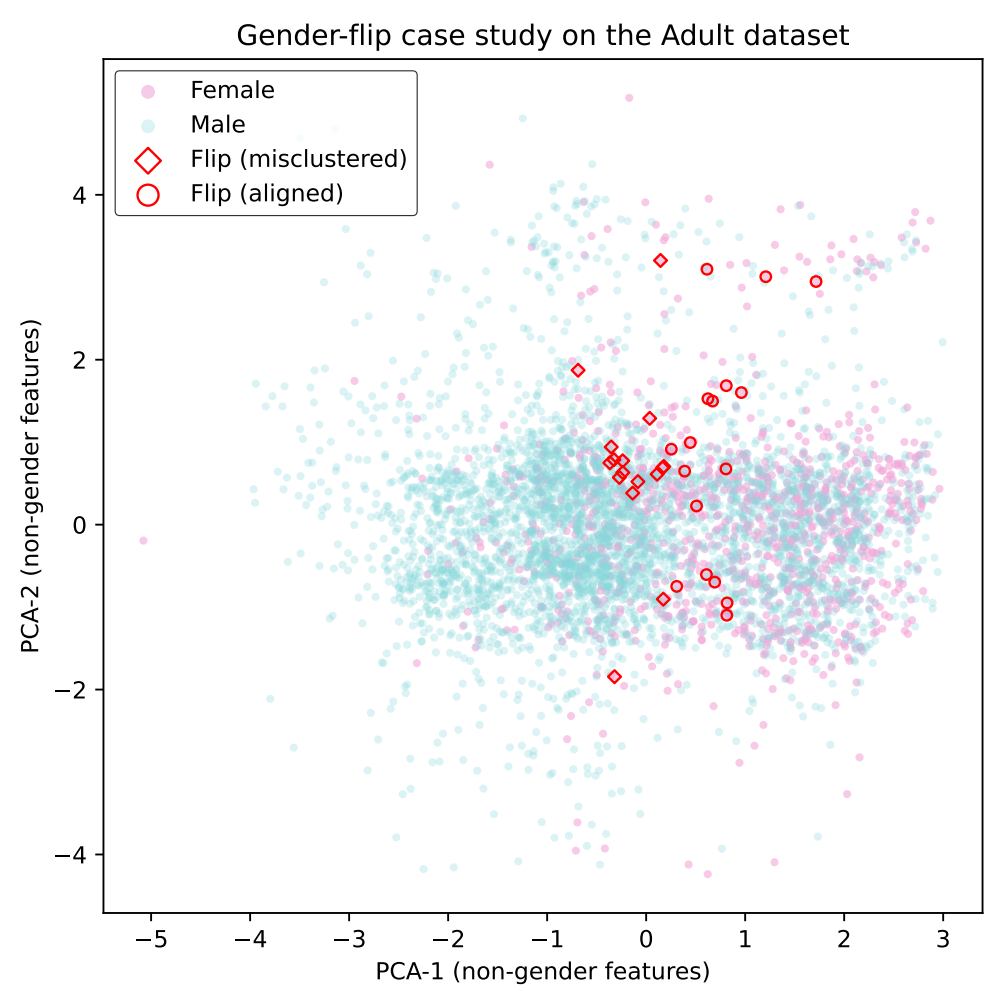

(c)

Figure 9: Visualization of the gender-flipping case study on the Adult dataset. Each subfigure corresponds to a synthetic 5K-row table pair (watermarked vs. unwatermarked). "Flip (misclustered)" denotes samples whose original "gender" label conflicts with their cluster label and subsequently flips after watermarking. "Flip (aligned)" denotes samples whose original "gender" label matches the cluster label but still flips after watermarking. In subfigure (a), 26 out of 5K samples exhibit a gender flip (46.2% misclustered); their mean distance to the cluster boundary is 0.31, compared with 0.78 for the remaining samples. In subfigure (b), 27 out of 5K samples exhibit a gender flip (48.1% misclustered); their mean distance to the cluster boundary is 0.26, compared with 0.75 for the remaining samples. In subfigure (c), 33 out of 5K samples exhibit a gender flip (48.5% misclustered); their mean distance to the cluster boundary is 0.29, compared with 0.77 for the remaining samples.

than on any single attribute. Whether a sample's "gender" value remains unchanged or flips therefore depends on how the sample is positioned relative to the distribution of its other features. When we cluster unwatermarked samples using all non-"gender" variables, the samples whose "gender" will flip after watermarking always lie near cluster boundaries or appear misclustered. In this sense, the flipped "gender" value remains semantically compatible with the rest features of the sample.

To justify this claim, we conduct a case study on the **Adult** dataset and focus on the flipping of "gender" variable. Specifically, we randomly select three pairs of 5K-row tables (watermarked vs. unwatermarked). After applying the YJT to standardize feature scales, we performed PCA on the

non-"gender" variables and retained the first two principal components for each sample. Based on these two principal components, we applied $k$-means clustering with $k = 2$, yielding two clusters that correspond to female-like" and male-like" profiles. We then examined the samples whose "gender" flips after watermarking, dividing them into two groups: those whose original "gender" label align with the cluster label and those that are misclustered. The visualization results in Figure 9 reveal several findings:

1. Flips in low-cardinality categorical variables are rare under TAB-DRW. In a 5K-row table, we typically observe only around 30 such cases.
2. Nearly half of the flipped samples are misclustered before watermarking, suggesting that the watermark embedding improves rather than disrupts their semantic coherence.
3. Among the remaining aligned samples that flip, we find that they tend to lie close to the cluster boundary. Across the three table examples, their average distance to the boundary is $0.29$, compared with $0.77$ for the other samples. This indicates that even when a flip occurs, the resulting "gender" value remains semantically meaningful.

# I DISCUSSION AND FUTURE WORK

Several directions remain open for future work. First, it is worth exploring whether there exists a provably optimal strategy for modifying the DFT components to maximize detectability while minimizing distortion—and if so, in what sense. Second, integrating TAB-DRW with differential privacy or membership inference protections could provide unified mechanisms for data traceability and privacy preservation. Third, adapting the watermark strength based on feature importance or downstream task performance could further improve the fidelity–detectability trade-off. We hope these directions inspire further research toward robust and responsible use of synthetic tabular data.

# J THE USE OF LARGE LANGUAGE MODELS (LLMS)

We acknowledge the use of a large language model (LLM) solely for polishing writing. The LLM was not employed for developing mathematical theorems, proofs, or any part of the experimental results or analysis. All text edited with the assistance of the LLM has been carefully reviewed to ensure that it does not introduce plagiarism or scientific misconduct. We take full responsibility for all content presented in this work.