# OpenReview forum: "TAB-DRW: A DFT-based Robust Watermark for Generative Tabular Data"
_ICLR.cc/2026/Conference — Submitted to ICLR 2026_

### Official Review · Reviewer_D23T · 2025-10-29

**Soundness:** 3
**Presentation:** 3
**Contribution:** 2
**Rating:** 4
**Confidence:** 4

**Summary:**

This paper proposes TAB-DRW, a post-editing watermarking method for synthetic tabular data that embeds signals in the frequency domain. The method first applies a Yeo-Johnson transform and standardization, then a row-wise discrete Fourier transform, and then modifies selected imaginary parts of DFT coefficients to match pseudorandom bits. A rank-based procedure generates pseudorandom bits without storing per-table keys, thereby improving robustness and memory efficiency. The authors provide theoretical analysis on distortion and robustness, and evaluate TAB-DRW on five benchmark datasets with a range of attacks and baselines.

**Strengths:**

-The paper is well written and easy to follow.
- The method is lightweight and model-agnostic.
- The paper provides formal bounds on distortion and a lower bound on expected detection statistic.
- Extensive experiments with five datasets, multiple fidelity metrics, many attack scenarios, and comparisons to multiple baselines.

**Weaknesses:**

-The evaluated attacks are broad and realistic, but the paper does not explore adversaries that aim specifically to target the rank-based bit retrieval or to invert the DFT modification. A discussion or small experiment on adaptive attackers who know the method class but not the key would be useful.
- The theoretical robustness analysis assumes transformed data is multivariate Gaussian. While the paper defends this choice after Yeo-Johnson transform, the assumption could fail on extreme non-Gaussian features. More discussion or a small robustness check under strongly non-Gaussian settings would strengthen the claims.

**Questions:**

- What is the runtime cost for detection on very large tables (e.g. millions of rows)?
- When converting the data back after watermarking, how often do rounding or clipping steps change the values enough to affect downstream tasks? Are there any cases where this process harms data quality or model performance?
- Have the authors tried stronger adversaries that know the general algorithm and can design perturbations to move ranks across bins? If not, how would the method handle this kind of targeted attack?

---

> ### Author Response · Authors · 2025-11-19
> **Response to the Reviewer D23T (1)**
>
> ## **Response to Reviewer D23T**
>
> We thank the reviewer for the recognition of our **clear writing**, **extensive evaluation**, and **theoretical analysis**. Below, we address your concerns point by point.
>
> ---
>
> ### **W1&Q3: Robustness under targeted attacks**
>
> **R1:** We thank the reviewer for the insightful and constructive suggestions on additional attacks. Here we implemented two adaptive attacks. In both cases, we consider adversaries who fully understand our pipeline, including the privacy-enhanced version (cf. Appendix B) for real-world deployment, but do not know the secret key.
>
> The first attack, **Adaptive Row Deletion**, aims to **corrupt the row ranking** and thus impair rank-based bit retrieval. The attacker generates a random key, computes the normalized rank of each row (following lines 3–6 of Algorithm 3), and then deletes a block of rows whose ranks form a contiguous interval. For example, under strength 0.1, the adversary removes rows whose normalized ranks lie within a randomly selected interval of length 0.1 in $[0,1]$. This manipulation disrupts the ranking structure significantly more than random row deletion.
>
> **Table A. Robustness of TAB-DRW against adaptive row deletion attacks of varying strength. Z-scores are computed on tables with 5K rows and averaged over 100 independent trials.**
>
> |**Datasets**||**No-attack**|**Adv. Row Del.@0.1**|**Adv. Row Del.@0.2**|**Adv. Row Del.@0.5**|
> |-:|:-:|:-:|:-:|:-:|:-:|
> |**Adult**||29.55±1.12|28.55±1.44|26.35±2.61|18.79±4.47|
> |**Magic**||61.42±1.02|56.71±2.98|49.36±5.77|28.41±7.28|
> |**Shoppers**||40.74±1.26|36.47±2.62|30.40±3.71|17.12±4.18|
> |**Default**||35.84±0.91|31.91±1.90|27.13±3.23|15.36±4.54|
> |**Drybean**||85.05±0.67|79.27±2.67|71.67±5.04|52.39±9.99|
>
> The results show that TAB-DRW **remains highly detectable** even under substantial adaptive row-deletion attacks. Although detectability decreases slightly compared with random row deletion, the use of a secret key and the stable tree-based bit-storage enables TAB-DRW to be resilient to these attacks specifically crafted to disrupt the row-ranking process.
>
> The second attack, **Rewatermarking**, targets to **erase sign-bit alignment** in the frequency domain. It exploits two properties of our privacy-enhanced TAB-DRW: first, its strong fidelity-preserving performance, and second, the fact that a watermark embedded with one key cannot be detected using another. An informed adversary can therefore repeatedly rewatermark the table with multiple different keys, aiming to perturb the original alignment and render the watermark undetectable to the original detection key.
>
> **Table B. Robustness of TAB-DRW against rewatermarking attacks of varying strength. Fidelity is averaged over four metrics across 10 independent trials, and the Z-scores are computed on tables with 5K rows and averaged over 100 independent trials. "Rewatermarking@$n$" denotes rewatermarking the table using $n$ randomly sampled keys.**
>
> |**Datasets**||**No-attack**||**Rewatermarking@1**||**Rewatermarking@3**||**Rewatermarking@10**||
> |-:|:-:|:-:|:-:|:-:|:-:|:-:|:-:|:-:|:-:|
> |||**Fidelity**|**Z-score**|**Fidelity**|**Z-score**|**Fidelity**|**Z-score**|**Fidelity**|**Z-score**|
> |**Adult**||0.799±0.006|29.55±1.12|0.787±0.008|23.66±1.17|0.772±0.006|16.26±1.09|0.766±0.009|17.26±1.34|
> |**Magic**||0.837±0.008|61.42±1.02|0.822±0.008|53.23±0.91|0.813±0.007|34.32±0.93|0.799±0.008|29.17±1.00|
> |**Shoppers**||0.854±0.009|40.74±1.26|0.847±0.008|31.97±1.15|0.829±0.009|20.14±1.09|0.813±0.008|16.67±1.09|
> |**Default**||0.836±0.013|35.84±0.91|0.827±0.011|32.85±1.00|0.811±0.013|19.40±1.07|0.809±0.010|26.28±1.18|
> |**Drybean**||0.849±0.017|85.05±0.67|0.832±0.017|44.79±0.81|0.801±0.017|29.47±0.83|0.806±0.014|33.77±0.95|
>
> From the results, we observe that TAB-DRW **remains highly detectable** even after ten rounds of rewatermarking, at which point the fidelity of the tabular data has already been noticeably degraded. These findings demonstrate that, without knowledge of the key, an attacker—despite understanding the TAB-DRW pipeline—cannot substantially disrupt the sign-bit alignment while preserving data fidelity.
>
> We will incorporate these attacks and the corresponding empirical results in the revised version of our paper.

---

> > ### Author Response · Authors · 2025-11-20
> > **Response to the Reviewer D23T (2)**
> >
> > ### **W2: Gaussian setting of robustness analysis**
> >
> > **R2:** We thank the reviewer for pointing this out. As acknowledged in Appendix D, although real-world tabular data with heterogeneity rarely follow a strict Gaussian distribution even after applying YJT, we use the multivariate Gaussian model in the main paper as a simplified yet widely adopted analytical tool to derive a closed-form lower bound of the Z-score, thereby providing insight into how sign-bit alignment in the frequency domain contributes to robustness. As the saying goes, "All models are wrong, but some are useful."
> >
> > To extend our robustness analysis to a broader class of distributions, we relax the Gaussian assumption in Theorem 2 to a **sub-Gaussian setting** and derive a corresponding lower bound on the Z-score under noise corruption. This setting accommodates features with **light-tailed distributions**, including **bounded or discrete categorical features**. For your convenience, we present the formal theorem and proof for sub-Gaussian setting as below:
> >
> > **Theorem (Robustness under sub-Gaussian samples).**
> > Assume that unwatermarked tabular data $\mathbf{X}\in\mathbb{R}^{N\times p}$ has i.i.d. rows
> > $\boldsymbol{x}\_i\in\mathbb{R}^p$ with zero mean and covariance $\Sigma\in\mathbb{R}^{p\times p}$,
> > and are *$\Sigma$–sub-Gaussian*, meaning there exists $\kappa \ge 1$ such that
> > for every $u\in\mathbb{R}^p$, $\\|\langle u,\boldsymbol{x}\_i\rangle\\|\_{\psi\_2}\ \le\ \kappa \sqrt{u^\top \Sigma u}$ [3].
> > Denote the smallest eigenvalue of $\Sigma$ by $\lambda_{\min}$. Define $Z(\gamma, \delta, \sigma)$ as the standard Z-score (as in Eq.(3)) computed on the Gaussian noise-corrupted table $\mathbf{X}\_{\mathrm{wm}} + \boldsymbol{\varepsilon}$, where $\mathbf{X}\_{\mathrm{wm}} \in \mathbb{R}^{N \times p}$ denote the table watermarked under soft hyperparameters $(\gamma, \delta)$ and $\varepsilon\_{i,j} \overset{\mathrm{i.i.d.}}{\sim} \mathcal{N}(0,\sigma^2)$. Fix any $\theta\in(0,1)$ and let $C_4>0$ denote a constant such that for every real sub-Gaussian $U$ with variance $v$ we have $\mathbb{E}U^4\le C_4\kappa^4v^2$. Define $\rho(\kappa,\theta)\ :=\ \frac{(1-\theta)^2}{2C_4\kappa^4}$.
> > Then, for any $\gamma\in[0,1]$ and $\delta,\sigma>0$,
> >
> > $$
> > \mathbb{E} \big[ Z(\gamma,\delta,\sigma)\big] \ge \sqrt{mN} \gamma \sup\_{\theta \in (0,1)} \left\\{ \rho(\kappa,\theta) \left[2 - \exp \left(-\frac{\theta \lambda\_{\min}}{2 \sigma^2}\right)- \exp \left(-\frac{\theta \lambda\_{\min} \delta^2}{2 \sigma^2}\right)\right] \right\\}.
> > $$
> >
> >
> >
> > ---
> >
> > ### **To be continued ...**

---

> > > ### Author Response · Authors · 2025-11-20
> > > **Response to the Reviewer D23T (3)**
> > >
> > > **R2 (continued):**
> > > **Proof.**
> > > Let $\boldsymbol{x}=(x_0,\ldots,x_{p-1})$ be a standardized row and $\boldsymbol{y}=\mathrm{DFT}(\boldsymbol{x})$.
> > > For the $t$-th effective frequency, we have
> > >
> > > $$
> > > \Im(y_t)\ =\ -\frac{1}{\sqrt{p}}\sum\_{n=0}^{p-1} x_n\sin\Big(\frac{2\pi tn}{p}\Big)
> > > \ =\ -\frac{1}{\sqrt{p}}\boldsymbol{s}_t^\top \boldsymbol{x},
> > > \quad
> > > \\|\boldsymbol{s}_t\\|_2^2=\sum\_{n=0}^{p-1}\sin^2\Big(\frac{2\pi tn}{p}\Big)=\frac{p}{2}.
> > > $$
> > >
> > > By the $\Sigma$–sub-Gaussian assumption and linearity, $\Im(y_t)$ is sub-Gaussian with
> > >
> > > $$
> > > v_t := \mathrm{Var}[\Im(y_t)]=\frac{1}{p}\boldsymbol{s}\_t^\top \Sigma \boldsymbol{s}\_t\ \in\ \Big[\frac{\lambda\_{\min}}{2},\frac{\lambda\_{\max}}{2}\Big],
> > > \quad \\|\Im(y_t)\\|\_{\psi_2} = \left\\| -\frac{1}{\sqrt{p}}\boldsymbol{s}\_t^\top \boldsymbol{x} \right\\|_{\psi_2} \le\ \kappa\sqrt{v_t}.
> > > $$
> > >
> > > Let $\alpha_t:=|\Im(y_t^{\mathrm{wm}})|$ and
> > > $z_t:=\Im(\widehat{\varepsilon}_t)\sim \mathcal{N}(0,\sigma^2/2)$ be the imaginary-part noise
> > > (Lemma 2). Follwing the analysis in Theorem 2, for each effective
> > > entry the alignment probability with its bit satisfies
> > >
> > > $$
> > > p\_{i,j}\ =\frac{1+\gamma}{2}-\frac{\gamma}{2}\Big(\mathbb{P}\_{\mathrm{flip}}(\sigma)+\mathbb{P}\_{\mathrm{flip}}^{(\delta)}(\sigma)\Big), \tag{**}
> > > $$
> > >
> > > where $\mathbb{P}\_{\mathrm{flip}}(\sigma)=\mathbb{E}\big[\mathbb{I}\\{z_t>\alpha_t\\}\big]
> > > =\mathbb{E}\big[Q(\sqrt{2}\alpha\_t/\sigma)\big]$, and $\mathbb{P}\_{\mathrm{flip}}^{(\delta)}$ is the same
> > > quantity with the amplitude scaled by $|\delta|$.
> > > Here $Q(u)=1-\Phi(u)$.
> > >
> > > By Lemma 1, $Q(u)\le \tfrac{1}{2}e^{-u^2/2}$, hence
> > > $\mathbb{P}_{\mathrm{flip}}(\sigma)\le \tfrac12\mathbb{E}\exp(-\alpha_t^2/\sigma^2)$.
> > > Write $U:=\Im(y_t)$ and $Y:=U^2$. For any $\theta\in(0,1)$,
> > > Paley–Zygmund inequality gives
> > >
> > > $$
> > > \mathbb{P}\big(Y\ge \theta\mathbb{E}Y\big)
> > > \ \ge\ \frac{(1-\theta)^2(\mathbb{E}Y)^2}{\mathbb{E}Y^2}
> > > \ =\ \frac{(1-\theta)^2v_t^2}{\mathbb{E}U^4}
> > > \ \ge\ \frac{(1-\theta)^2}{C_4\kappa^4}
> > > \ =:\ \eta,
> > > $$
> > >
> > > where we used the sub-Gaussian fourth-moment bound $\mathbb{E}U^4\le C_4\kappa^4 v_t^2$.
> > > Thus, by conditioning on the event $\\{Y\ge \theta v_t\\}$,
> > >
> > > $$
> > > \mathbb{E}\exp(-\alpha\_t^2/\sigma^2)
> > > \ \le\ (1-\eta)\cdot 1\ +\ \eta\cdot \exp\Big(-\frac{\theta v\_t}{\sigma^2}\Big)
> > > \ \le\ 1-\eta\Big(1-e^{-\theta\lambda\_{\min}/(2\sigma^2)}\Big).
> > > $$
> > >
> > > Consequently,
> > >
> > > $$
> > > \mathbb{P}\_{\mathrm{flip}}(\sigma)\ \le\ \frac12\Big[1-\eta\Big(1-e^{-\theta\lambda\_{\min}/(2\sigma^2)}\Big)\Big],
> > > \quad
> > > \mathbb{P}\_{\mathrm{flip}}^{(\delta)}(\sigma)\ \le\ \frac12\Big[1-\eta\Big(1-e^{-\theta\lambda\_{\min}\delta^2/(2\sigma^2)}\Big)\Big].
> > > $$
> > >
> > > Plugging these bounds into (**), then we obtain
> > >
> > > $$
> > > p_{i,j}\ \ge\ \frac12\ +\ \frac{\gamma\eta}{4}
> > > \bigg[2-e^{-\theta\lambda_{\min}/(2\sigma^2)}-e^{-\theta\lambda_{\min}\delta^2/(2\sigma^2)}\bigg].
> > > $$
> > >
> > > Using $\mathbb{E}Z=2\sqrt{mN}(p_{i,j}-\tfrac12)$, we arrive at
> > >
> > > $$
> > > \mathbb{E}[Z(\gamma,\delta,\sigma)]
> > > \ \ge\ \sqrt{mN}\gamma\frac{\eta}{2}
> > > \bigg[2-e^{-\theta\lambda_{\min}/(2\sigma^2)}-e^{-\theta\lambda_{\min}\delta^2/(2\sigma^2)}\bigg].
> > > $$
> > >
> > > Recalling that $\eta = (1-\theta)^2/(C_4 \kappa^4)$ gives the result for $\rho(\kappa,\theta) = (1-\theta)^2/(2C_4 \kappa^4)$. Since $\theta \in (0,1)$ is a fixed hyperparameter, it can be tuned to obtain the tightest possible lower bound
> > >
> > > $$
> > > \mathbb{E}\big[Z(\gamma,\delta,\sigma)\big]
> > > \ \ge\ \sqrt{mN}\gamma\sup\_{\theta \in (0,1)} \left\\{ \rho(\kappa,\theta)
> > > \bigg[
> > > 2-\exp\Big(-\frac{\theta\lambda\_{\min}}{2\sigma^2}\Big)
> > > -\exp\Big(-\frac{\theta\lambda\_{\min}\delta^2}{2\sigma^2}\Big)
> > > \bigg]\right\\}.
> > > $$
> > >
> > > **Remark (What the sub-Gaussian assumption covers).**
> > > The $\Sigma$-sub-Gaussian assumption strictly generalizes the Gaussian assumption used in
> > > Theorem 2 and many non-Gaussian settings that are common in tabular data: 1) bounded/quantized/discrete features (e.g., "gender" or education features); and 2) finite mixtures of light-tailed distributions with uniformly bounded covariances (a finite mixture of sub-Gaussians is sub-Gaussian with the worst-component parameter). In conclusion, **non-Gaussian distribution with light tails are covered**.

---

> > > > ### Author Response · Authors · 2025-11-20
> > > > **Response to the Reviewer D23T (4)**
> > > >
> > > > ### **Q1: Runtime cost on very large tables**
> > > >
> > > > **R3:** Since prior work on tabular data watermarking primarily evaluates tables with no more than 10K rows [1][2], our main experiments also focus on this scale. Here, we report the average runtimes for watermark detection on tables with **100K rows** (due to GPU memory constraints, we cannot use TabSyn to generate 1M-row tables), averaged over 10 trials. Experiments across five datasets are conducted on a 40GB NVIDIA A100 GPU and M1 Pro CPU.
> > > >
> > > >
> > > > **Table C. Comparison of watermark detection runtimes (in seconds) across watermarking methods. Values for TabWak\* denote GPU runtimes, whereas all other methods are measured in CPU time.**
> > > >
> > > > | **Dataset** | **GLW** | **MUSE** | **TabWak*** | **TabularMark** | **TAB-DRW** |
> > > > |:-----------:|:-------------------------:|:------------------------:|:-------------------------:|:------------------------:|:------------------------:|
> > > > | **Adult**       |  0.20| 14.61 | 1808.60 | 80.69 | 2.05 |
> > > > | **Magic**       |  0.10| 13.06 | 1472.28 | 71.89 | 2.02 |
> > > > | **Shoppers**    |  0.31| 15.76 | 1659.47 | 89.42 | 3.12 |
> > > > | **Default**     |  0.38| 17.91 | 1994.51 | 87.85 | 3.88 |
> > > > | **Drybean**     |  0.15| 16.63 | 1569.88 | 82.37 | 2.87 |
> > > >
> > > > ---
> > > >
> > > > ### **Q2: Impact of rounding and clipping on downstream tasks**
> > > >
> > > > **R4:** We would like to emphasize that the **rounding and clipping operations do not harm downstream task performance**. Since the outputs of the inverse DFT and YJT are real-valued, these operations are required for discrete features to preserve semantic validity. In fact, omitting rounding and clipping can harm downstream performance; for instance, values such as a gender of 0.9 or an age of 32.1 are not semantically meaningful and cannot be used reliably by downstream models.
> > > >
> > > > Below are the results of an ablation study comparing Z-scores with and without rounding and clipping across five datasets, together with the frequency and magnitude of these operations. For **Magic** dataset there are no rounding or clipping happening since all the columns are continuous. For other datasets, the impact of these two post-processing operations on watermark detectability is negligible.
> > > >
> > > > **Table D. Detection performance of TAB-DRW with or without the rounding and clipping operations. Z-scores are averaged over 100 trials on tables with 1K rows. "Rounding magnitude" denotes the average rounding magnitude of discrete entries, and "Clipping ratio" denotes the fraction of discrete entries that are clipped.**
> > > >
> > > > | **Dataset** | **W/O round and clip** | **W/ round and clip** | **Rounding magnitude** | **Clipping ratio** |
> > > > |:-----------:|:-------------------------:|:------------------------:|:-------------------------:|:------------------------:|
> > > > | **Adult**       | 15.21 ± 1.00 | 12.81 ± 1.17 | 0.0911 ± 0.0015| 0.0008 ± 0.0004|
> > > > | **Magic**       | 27.34 ± 0.93 | 27.34 ± 0.93 | 0.0000 ± 0.0000| 0.0000 ± 0.0000|
> > > > | **Shoppers**    | 21.00 ± 1.15 | 18.18 ± 1.28 | 0.0969 ± 0.0042| 0.0244 ± 0.0036|
> > > > | **Default**     | 17.94 ± 0.95 | 15.98 ± 0.92 | 0.0542 ± 0.0018| 0.0151 ± 0.0013|
> > > > | **Drybean**     | 37.79 ± 1.02 | 38.03 ± 1.03 | 0.1285 ± 0.0027| 0.0145 ± 0.0022|
> > > >
> > > > ---
> > > >
> > > > ### **References**
> > > >
> > > > [1] Zhu, Chaoyi, et al. TabWak: A watermark for tabular diffusion models. *In The Thirteenth International Conference on Learning Representations*, 2025.
> > > >
> > > > [2] Fang, Liancheng, et al. MUSE: Model-Agnostic Tabular Watermarking via Multi-Sample Selection. *arXiv preprint arXiv:2505.24276*, 2025.
> > > >
> > > > [3] Vershynin, Roman. High-dimensional probability: An introduction with applications in data science. *Cambridge university press*, 2018.

---

> ### Comment · Reviewer_D23T · 2025-11-23
> **Addition**
>
> One last question from my side: How would the robustness experiment under different intensities for Tabwak look like? The authors have diligently addressed my other questions/concerns.

---

> > ### Author Response · Authors · 2025-11-23
> > **Response to the Reviewer D23T (5)**
> >
> > We sincerely thank the reviewer for the encouraging feedback and are glad that our previous response has addressed your concerns. Below, we provide our reply regarding the robustness evaluation of **TabWak\*** under the two adaptive attacks introduced earlier.
> >
> > We reproduce both the adaptive row deletion attack and the rewatermarking attack for TabWak*. However, we would like to clarify that these two attacks were originally designed to target the row ranking and sign-bit alignment mechanisms in **TAB-DRW**. Consequently, when applied to TabWak*, the first attack essentially works like random row deletion, while the second acts as a form of frequency-domain noise perturbation. The corresponding results are shown below.
> >
> >
> > **Table E. Robustness of TabWak\* against adaptive row deletion attacks of varying strength. Z-scores are computed on tables with 5K rows and averaged over 10 independent trials.**
> >
> > |**Datasets**||**No-attack**|**Adv. Row Del.@0.1**|**Adv. Row Del.@0.2**|**Adv. Row Del.@0.5**|
> > |-:|:-:|:-:|:-:|:-:|:-:|
> > |**Adult**||15.67±0.97|14.79±0.93|13.96±1.03|11.28±1.09|
> > |**Magic**||19.83±1.01|18.53±0.92|16.79±0.95|13.19±1.14|
> > |**Shoppers**||10.38±1.02|9.59±0.96|8.97±0.99|7.24±1.16|
> > |**Default**||23.60±1.00|22.74±1.01|21.93±0.95|16.48±0.99|
> > |**Drybean**||17.80±0.97|16.93±0.99|15.97±1.03|12.16±1.15|
> >
> > **Table F. Robustness of TabWak\* against rewatermarking attacks of varying strength. Z-scores are computed on tables with 5K rows and averaged over 10 independent trials. "Rewatermarking@$n$" denotes rewatermarking the table using $n$ randomly sampled keys.**
> >
> > |**Datasets**||**No-attack**|**Rewatermarking@1**|**Rewatermarking@3**|**Rewatermarking@10**|
> > |-:|:-:|:-:|:-:|:-:|:-:|
> > |**Adult**||15.67±0.97|6.89±2.47|3.14±1.59|2.28±1.30|
> > |**Magic**||19.83±1.01|14.77±3.01|15.10±3.23|10.25±2.28|
> > |**Shoppers**||10.38±1.02|4.91±1.64|4.15±1.49|1.71±1.02|
> > |**Default**||23.60±1.00|19.77±3.35|17.27±3.17|12.36±2.96|
> > |**Drybean**||17.80±0.97|14.87±2.46|14.09±2.59|9.31±2.15|

---

### Official Review · Reviewer_vXSA · 2025-10-30

**Soundness:** 3
**Presentation:** 3
**Contribution:** 2
**Rating:** 6
**Confidence:** 3

**Summary:**

This paper addresses the problem of tracing synthetically generated tabular data to prevent its misuse. Existing watermarking solutions are often computationally costly, lack robustness, or cannot handle mixed data types. The authors propose TAB-DRW, a post-editing watermarking scheme that operates in the frequency domain. . Experiments on five benchmark datasets demonstrate that TAB-DRW achieves strong watermark detectability and high data fidelity. The method shows superior robustness against a wide range of post-processing attacks compared to existing techniques. The authors claim TAB-DRW is a computationally efficient and broadly applicable solution for generative tabular data.

**Strengths:**

1.  The rank-based pseudorandom bit generation is a novel, storage-free mechanism. It enhances robustness by deterministically recomputing bits from stable row statistics, as detailed in Algorithm 3.
2.  The method effectively handles mixed-type data by combining Yeo-Johnson transformation with DFT. This creates a standardized frequency domain for uniform watermark embedding, as shown in Section 2.1.
3.  Experimental evaluation is rigorous, testing against ten distinct attacks and four recent baselines. Table 3 and Figure 4 demonstrate superior robustness across various attack types and datasets.

**Weaknesses:**

1.  The paper fails to clearly articulate the strategy for selecting specific numerical columns for watermark embedding; these details are only mentioned ad-hoc in the appendix as implementation notes. It is recommended that the main methodology section includes a discussion on the principles of column selection and an analysis of how different selection strategies impact the fidelity-robustness trade-off.
2.  The logic for pseudo-random bit generation in Algorithm 3, particularly the rule for determining bit-pairs based on `k%4`, lacks sufficient theoretical motivation. The paper should provide a more detailed explanation for why this specific mapping is adopted and how it ensures robustness for bit sequences from adjacent bins, for instance, by explicitly connecting it to concepts like Gray codes.
3.  A subtle discrepancy exists between the theoretical analysis (Section 3), which assumes fixed YJT and standardization parameters, and the practical detection process (Appendix D), which re-fits these parameters on suspect data. Although experiments in Appendix D suggest this gap has a negligible impact, the theoretical guarantees in the main text should be more cautiously qualified to clarify that the analysis is conducted under an idealized model.
4.  The paper lacks an impact analysis of the final rounding and clipping step for discrete and bounded features. This non-linear operation could potentially weaken or even erase the watermark signal introduced by small modifications in the frequency domain; a discussion or experimental analysis of its potential effect on watermark detectability is advised.

**Questions:**

1.  Regarding column selection: Could you elaborate on the strategy for choosing which columns to watermark? Is this selection based on certain feature properties (e.g., variance, data type, correlation with other features)? Have you analyzed the sensitivity of the method's performance to this choice?
2.  Regarding the bit generation algorithm: In Algorithm 3, the bit-pair assignment rule (`k%4 = 0 or 3`) appears designed to make bit sequences of adjacent bins more similar. Could you provide a more formal justification for this choice? Is it an approximation of a Gray code, and have you considered alternative encoding schemes and their effectiveness?

---

> ### Author Response · Authors · 2025-11-19
> **Response to the Reviewer vXSA (1)**
>
> ## **Response to Reviewer vXSA**
>
> We thank the reviewer for the detailed and insightful assessment, and for recognizing our **novel and effective methodology** as well as our **rigorous experimental evaluation**. We address each concern point by point below.
>
> ---
>
> ### **W1&Q1: Column selection for watermarking**
>
> **R1:** Thank you for raising this important point. Due to space constraints, we did not explicitly clarify in the main paper that we **do not employ any specialized column-selection strategy** and exclude only **columns with extreme distributions**. We will add this explanation in the revised version.
>
> We clarify that our empirical evaluation focuses on numerical columns (including mixed continuous and discrete types) to enable a **fair comparison** with the other post-editing watermarking methods, GLW and TabularMark, which both suffer substantial fidelity degradation when applied to all columns. We exclude only columns with **extremely skewed or degenerate distributions**, such as columns consisting mostly of zeros with a few very large outliers, because they contribute little to the watermark signal and can cause scaling issues in a small number of rows. Apart from this minimal exclusion, we **do not use any additional column-selection strategy** in our experiments. Developing adaptive column-selection methods that achieve optimal fidelity-detectability trade-off remains an appealing direction for future work, as noted in Appendix I.
>
> Since TAB-DRW is a post-editing watermark with high computational efficiency, tabular model providers or dataset owners can flexibly **choose any subset of columns to watermark based on their needs**. For example, in synthetic medical datasets, the watermark may be applied only to columns containing sensitive or high-value information that attackers are less likely to modify. In general, using more columns for watermarking **improves robustness** (Theorem 2 shows that the lower bound of the Z-score scales with the number of selected columns), while incurring **slightly higher distortion**.
>
> Based on the reviewer's suggestions, we also report results obtained by applying TAB-DRW to all columns, showing how different selection strategies influence the tradeoff between fidelity and detectability.
>
> **Table A. Data fidelity and watermark detectability evaluated on TAB-DRW with different columns selected for watermarking. Fidelity metrics are averaged over 10 trials, and Z-scores are averaged over 100 trials.**
>
> | **Datasets** | **Col. Selection** | **Density ↑** | **Corr ↑** | **C2ST ↑** | **MLE ↑** | **1k rows ↑** | **5k rows ↑** |
> |-------------:|:----------:|:------------:|:---------:|:---------:|:---------:|:-------------:|:-------------:|
> | **Adult**        | **All Col.**    | 0.909 ± 0.005 | 0.859 ± 0.005 | 0.597 ± 0.009 | 0.808 ± 0.008 | **14.56 ± 0.99** | **32.89 ± 1.06** |
> |        | **Original**     | **0.915 ± 0.005** | **0.864 ± 0.004** | **0.604 ± 0.008** | **0.816 ± 0.009** | 12.81 ± 1.17  | 29.55 ± 1.12  |
> | **Magic**        | **All Col.**    | 0.914 ± 0.006 | 0.936 ± 0.003 | 0.674 ± 0.008 | 0.818 ± 0.014 | 24.66 ± 1.08  | 55.47 ± 1.09  |
> |         | **Original**     | **0.917 ± 0.005** | **0.937 ± 0.003** | **0.676 ± 0.009** | **0.818 ± 0.014** | **27.34 ± 0.93** | **61.42 ± 1.02** |
> | **Shoppers**     | **All Col.**    | 0.901 ± 0.005 | 0.897 ± 0.003 | 0.704 ± 0.009 | 0.887 ± 0.012 | **19.59 ± 1.08**  | **43.84 ± 1.14**  |
> |      | **Original**     | **0.909 ± 0.006** | **0.902 ± 0.003** | **0.712 ± 0.013** | **0.891 ± 0.014** | 18.18 ± 1.28 | 40.74 ± 1.26 |
> | **Default**      | **All Col.**    | 0.919 ± 0.009 | 0.902 ± 0.013 | 0.705 ± 0.019 | 0.787 ± 0.011 | **22.21 ± 1.03**  | **49.96 ± 0.99**  |
> |       | **Original**     | **0.929 ± 0.010** | **0.907 ± 0.011** | **0.717 ± 0.018** | **0.791 ± 0.013** | 15.98 ± 0.92 | 35.84 ± 0.91 |
> | **Drybean**      | **All Col.**    | 0.929 ± 0.010 | 0.924 ± 0.007 | 0.649 ± 0.022 | 0.878 ± 0.019 | **38.35 ± 0.89**  | **85.47 ± 0.73**  |
> |       | **Original**     | **0.931 ± 0.013** | **0.928 ± 0.007** | **0.655 ± 0.029** | **0.880 ± 0.019** | 38.03 ± 1.03 | 85.05 ± 0.67 |
>
> The results above show that watermarking more columns improves detectability while reducing fidelity, consistent with our theoretical analysis.

---

> ### Author Response · Authors · 2025-11-19
> **Response to the Reviewer vXSA (2)**
>
> ### **W2&Q2: Motivation for choosing $k \\% 4 = 0\ \text{or}\ 3$ for bit-pair assignment**
>
> **R2:** We appreciate the reviewer for connecting our bit generation scheme to **Gray codes**. Thank you for bringing this to our attention. We would like to clarify that $k \\% 4 = 0\ \text{or}\ 3$ characterizes the distribution pattern of bit pairs across each level of the storage tree (repeating in the cycle $[1,0][0,1][0,1][1,0]$) and their corresponding mapping to rank bins. From the perspective of Gray codes, our construction can be viewed as a **special case of a 2-Gray code**. We present the formal construction process below:
>
> Let $n \in \mathbb{N}$. Denote by $\\{0,1\\}^n$ the set of all binary strings of length $n$, and by $d_H(\cdot,\cdot)$ the Hamming distance on $\\{0,1\\}^n$. Let $G_n = (g_0^n,g_1^n,\ldots,g_{2^n-1}^n)$ with $g_i^n \in \\{0,1\\}^n$ be a standard $n$-bit 1-Gray code, specified up to cyclic permutation. By definition, $d_H(g_i^n,g_{i+1}^n) = 1\ \text{for all\ } i \in \\{0,\ldots,2^n-1\\}$.
>
> We define the **pair-encoding map** $\varphi : \\{0,1\\} \to \\{0,1\\}^2, \varphi(0) = 10, \varphi(1) = 01$ and extend $\varphi$ to a map $\phi: \\{0,1\\}^n \to \\{0,1\\}^{2n}$
> by applying it coordinatewise: for $g^n = b_1 b_2 \cdots b_n \in \\{0,1\\}^n$, $b_j \in \\{0,1\\}$, let $\phi(g^n) = \varphi(b_1)\varphi(b_2)\cdots\varphi(b_n)$.
>
> For $n \in \mathbb{N}$, we then can define the set $H_{2n} = (h_0^{2n},h_1^{2n},\ldots,h_{2^n-1}^{2n})$ and $H_{2n-1} = (\pi(h_0^{2n}),\pi(h_1^{2n}),\ldots,\pi(h_{2^n-1}^{2n}))$, where $h_i^{2n} := \phi(g_i^{n}) \in \\{0,1\\}^{2n}$ and $\pi : \\{0,1\\}^{2n} \to \\{0,1\\}^{2n-1}$ be the projection that deletes the last coordinate: $\pi(x_1,\ldots,x_{2n}) = (x_1,\ldots,x_{2n-1})$.
>
> By construction, the family $\\{H_n\\}$ forms a collection of 2-Gray codes, and any two consecutive codewords $h_i^n$ and $h_{i+1}^n$ satisfy $d_H(h_i^n, h_{i+1}^n) \leq 2$. In our paper, a tree of depth $N$ stores the sequence $\\{H_n\\}_{n=1}^{2N}$, and each rank bin corresponds to a distinct $h_i^{2N}$ or $\pi(h_i^{2N})$. Compared with using a standard 1-Gray code, this construction has the advantage of slowing the growth rate of rank bins as the number of table columns increases, since $|H_n| = 2^{\lceil n/2 \rceil}$ whereas $|G_n| = 2^{n}$. This reduction leads to greater robustness of bit retrieval against rank shifts. Furthermore, the condition $k \\% 4 = 0\ \text{or}\ 3$ is uniquely determined by the **chosen cyclic permutation of the underlying 1-Gray code**.
>
> We provide additional experimental results using a **1-Gray code** for bit sequence generation to further justify the robustness claim above. Specifically, we modify lines 7 and 9 in Algorithm 3 as follows:
>
> - Line 7: **for** $j \leftarrow 1$ **to** $m$ **do**
> - Line 9: Append $1$ to $\mathbf{S}$ if $k \\% 4 = 0$ or $3$; otherwise append $0$.
>
> **Table B. Data fidelity and watermark detectability evaluated on TAB-DRW using different Gray codes for bit generation. Fidelity metrics are averaged over 10 trials, and Z-scores are averaged over 100 trials. 2-Gray code are schemes applied in our paper.**
>
> | **Datasets** | **Bit Gen.** | **Density ↑** | **Corr ↑** | **C2ST ↑** | **MLE ↑** | **1k rows ↑** | **5k rows ↑** |
> |-------------:|:----------:|:------------:|:---------:|:---------:|:---------:|:-------------:|:-------------:|
> | **Adult**        | **1-Gray code** | **0.916 ± 0.005** | 0.863 ± 0.005 | 0.600 ± 0.009 | **0.816 ± 0.008** | 11.06 ± 0.99 | 24.95 ± 0.86 |
> |                  | **2-Gray code** | 0.915 ± 0.005 | **0.864 ± 0.004** | **0.604 ± 0.008** | 0.816 ± 0.009 | **12.81 ± 1.17**  | **29.55 ± 1.12**  |
> | **Magic**        | **1-Gray code** | 0.917 ± 0.006 | 0.936 ± 0.003 | **0.676 ± 0.008** | 0.818 ± 0.014 | 21.74 ± 0.96  | 48.78 ± 0.97  |
> |                  | **2-Gray code** | **0.917 ± 0.005** | **0.937 ± 0.003** | 0.676 ± 0.009 | **0.818 ± 0.014** | **27.34 ± 0.93** | **61.42 ± 1.02** |
> | **Shoppers**     | **1-Gray code** | 0.909 ± 0.006 | **0.909 ± 0.004** | **0.715 ± 0.011** | **0.892 ± 0.014** | 17.30 ± 1.14  | 38.48 ± 1.14  |
> |                  | **2-Gray code** | **0.909 ± 0.006** | 0.902 ± 0.003 | 0.712 ± 0.013 | 0.891 ± 0.014 | **18.18 ± 1.28** | **40.74 ± 1.26** |
> | **Default**      | **1-Gray code** | 0.927 ± 0.010 | **0.918 ± 0.009** | 0.715 ± 0.013 | **0.793 ± 0.013** | 15.07 ± 1.02  | 33.72 ± 0.97  |
> |                  | **2-Gray code** | **0.929 ± 0.010** | 0.907 ± 0.011 | **0.717 ± 0.018** | 0.791 ± 0.013 | **15.98 ± 0.92** | **35.84 ± 0.91** |
> | **Drybean**      | **1-Gray code** | **0.933 ± 0.010** | **0.932 ± 0.007** | **0.659 ± 0.022** | **0.880 ± 0.019** | 25.40 ± 0.92  | 57.74 ± 1.03  |
> |                  | **2-Gray code** | 0.931 ± 0.013 | 0.928 ± 0.007 | 0.655 ± 0.029 | 0.880 ± 0.019 | **38.03 ± 1.03** | **85.05 ± 0.67** |
>
> ---
> ### **To be continued ...**

---

> > ### Author Response · Authors · 2025-11-19
> > **Response to the Reviewer vXSA (3)**
> >
> > **R2 (continued):**
> > **Table C. Robustness evaluation of TAB-DRW using different Gray codes for bit generation under the strengthened attack setting: Z-scores are averaged over 100 trials on tables with 5K rows. 2-Gray codes is the scheme applied in our paper.**
> >
> > | **Datasets** | **Bit Gen.** | **Row Del.@20\%** | **Col Del.@3** | **Cell Del.@20\%** | **G-Noise@\20%** | **C-Noise@20\%** | **A-Noise@20\%** | **Truncation** | **Quantization@20\%** | **Resample** | **Shuffle** |
> > |-------------:|:----------:|:------------:|:---------:|:---------:|:---------:|:-------------:|:-------------:|:-------------:|:-------------:|:-------------:|:-------------:|
> > | **Adult**        | **1-Gray code** | 22.30 | 10.10 | 11.19 | 12.32 | 18.11| 21.24| 24.95 | 16.04 | 21.49 | 24.95 |
> > |                  | **2-Gray code** | **26.34** | **13.12** | **14.37** | **14.29** | **20.10**| **21.85**| **29.55** | **16.41** | **28.15** | **29.55** |
> > | **Magic**        | **1-Gray code** | 43.47 | 8.46 | 16.27 | 22.81 | 44.14 | 21.42 | 38.83 | 19.32 | 24.58 | 48.78 |
> > |                  | **2-Gray code** | **54.85** | **11.75**| **21.60** | **33.93** | **48.38** | **29.06** | **52.62** | **26.43** | **37.61** | **61.42** |
> > | **Shoppers**     | **1-Gray code** | 34.36 | 13.40 | **14.00** | 35.54 | 25.05 | **13.12** | **31.00** | 23.36 | 14.34 | 38.48 |
> > |                  | **2-Gray code** | **36.21** | **13.75** | 13.27 | **37.71** | **32.60** | 13.10 | 30.28 | **25.72** | **29.28** | **40.74** |
> > | **Default**      | **1-Gray code** | 29.84 | 19.45 | 15.05 | 21.66 | 26.14 | 12.52 | 33.72 | 11.06 | 30.08 | 33.72 |
> > |                  | **2-Gray code** | **31.92** | **20.70** | **15.44** | **23.75** | **27.20** | **16.99** | **35.84** | **14.10** | **32.36** | **35.84** |
> > | **Drybean**      | **1-Gray code** | 51.38 | 15.48 | 16.87 | 12.73 | 48.93 | 15.50 | 22.94 | 15.31 | 39.44 | 57.74 |
> > |                  | **2-Gray code** | **75.91** | **32.92** | **35.27** | **23.57** | **77.70** | **42.48** | **42.14** | **45.95** | **68.69** | **85.05** |
> >
> >
> > From the results, we observe that using a 1-Gray code for bit generation yields a slight improvement in data fidelity, as it more closely mimics an ideal bit sampled from a Bernoulli distribution. However, its detectability and robustness decrease on several datasets, especially those dominated by continuous variables such as **Magic** and **Drybean**. We attribute this to the finer-grained rank-bin partition induced by the 1-Gray code and to the greater instability of the sum-based score for continuous features under perturbations, which makes cross-bin rank shifts more likely to happen.

---

> > > ### Author Response · Authors · 2025-11-19
> > > **Response to the Reviewer vXSA (4)**
> > >
> > > ### **W3: Discrepancy between the theoretical analysis and practical implementation**
> > >
> > > **R3:** We appreciate the reviewer’s observation. We acknowledge that our robustness analysis is carried out under **an idealized model that does not refit the YJT parameters**. However, under the Gaussian assumption used in the analysis, the strong fidelity preservation of TAB-DRW keeps the watermarked data very close to the original Gaussian distribution. Consequently, the effect of YJT refitting on both the data distribution and the watermark signal induced by sign-bit alignment is **minimal**. This makes the idealized model **appropriate for the theoretical robustness study**, and it does not alter the conclusions we derive.
> > >
> > > Introducing an adaptive, nonlinear YJT transformation, where the parameter $\lambda$ is estimated via maximum likelihood, would prevent us from obtaining a closed-form lower bound on the Z-score and would not offer additional insight into the core mechanism driving robustness. The theoretical analysis is not intended to guarantee performance under all practical refinements of the pipeline, but rather to gain an intuition on why our method is fundamentally robust. We will clarify these points more explicitly in the revised version.
> > >
> > > To validate our point in the first parapgraph, we provide a case study to illustrate the **minimal difference between the original YJT parameters and those refitted after watermark embedding**. Specifically, we generate multivariate Gaussian tables with row counts $N \in \\{100, 1000, 10000\\}$ and column counts $p \in \\{10, 50, 100\\}$. Two covariance structures are considered: the identity matrix and an AR(1) matrix $\Sigma_{ij} = \rho^{|i-j|}$ with $\rho = 0.4$. Each table is watermarked using TAB-DRW with $(\gamma, \delta) = (0.5, 0.5)$. The YJT parameters $\lambda$, mean $\mu$, and standard deviation $\sigma$ are recorded before and after watermarking and averaged across all columns. The results show that TAB-DRW introduces **negligible perturbations to these parameters** across different dimensions and covariance structures.
> > >
> > >
> > > **Table D. YJT Parameters Summary (before vs. after watermarking) across varying dimensions and covariance structures.**
> > >
> > > | **$\Sigma$** | **N**   | **p**  | **$\lambda$ Before** |**$\lambda$ After** | **$\mu$ Before** | **$\mu$ After** | **$\sigma$ Before** | **$\sigma$ After** |
> > > |------------|------|-----|-----------|----------|--------------|-------------|-------------|------------|
> > > | Identity   | 100  | 10  | 1.0164    | 1.0117   | -0.0230      | -0.0229     | 0.9792      | 0.9592     |
> > > | Identity   | 100  | 50  | 1.0065    | 1.0030   | -0.0017      | -0.0027     | 0.9957      | 0.9835     |
> > > | Identity   | 100  | 100 | 0.9743    | 0.9807   | -0.0190      | -0.0166     | 0.9917      | 0.9817     |
> > > | Identity   | 1000 | 10  | 1.0113    | 1.0113   | 0.0418       | 0.0420      | 1.0002      | 0.9883     |
> > > | Identity   | 1000 | 50  | 0.9971    | 0.9940   | -0.0070      | -0.0081     | 1.0020      | 0.9939     |
> > > | Identity   | 1000 | 100 | 0.9885    | 0.9861   | -0.0056      | -0.0064     | 1.0014      | 0.9943     |
> > > | Identity   | 10000| 10  | 0.9970    | 0.9999   | 0.0017       | 0.0026      | 0.9988      | 0.9889     |
> > > | Identity   | 10000| 50  | 1.0029    | 1.0027   | 0.0000       | -0.0001     | 1.0000      | 0.9925     |
> > > | Identity   | 10000| 100 | 1.0003    | 1.0011   | 0.0017       | 0.0020      | 0.9997      | 0.9930     |
> > > | AR(1)      | 100  | 10  | 0.9849    | 0.9735   | 0.0075       | 0.0028      | 1.0367      | 1.0231     |
> > > | AR(1)      | 100  | 50  | 1.0158    | 1.0128   | 0.0072       | 0.0064      | 0.9766      | 0.9680     |
> > > | AR(1)      | 100  | 100 | 0.9917    | 0.9921   | -0.0222      | -0.0223     | 0.9811      | 0.9724     |
> > > | AR(1)      | 1000 | 10  | 0.9866    | 0.9892   | -0.0024      | -0.0015     | 0.9963      | 0.9873     |
> > > | AR(1)      | 1000 | 50  | 0.9926    | 0.9911   | -0.0046      | -0.0051     | 0.9988      | 0.9926     |
> > > | AR(1)      | 1000 | 100 | 0.9972    | 0.9967   | 0.0010       | 0.0009      | 0.9954      | 0.9898     |
> > > | AR(1)      | 10000| 10  | 1.0037    | 1.0051   | 0.0008       | 0.0013      | 0.9958      | 0.9873     |
> > > | AR(1)      | 10000| 50  | 1.0023    | 1.0032   | -0.0051      | -0.0048     | 0.9998      | 0.9938     |
> > > | AR(1)      | 10000| 100 | 0.9998    | 0.9998   | -0.0001      | -0.0000     | 0.9999      | 0.9946     |

---

> > > > ### Author Response · Authors · 2025-11-19
> > > > **Response to the Reviewer vXSA (5)**
> > > >
> > > > ### **W4: Impact of rounding and clipping on watermark detectability**
> > > >
> > > > **R4:** We appreciate the reviewer for noticing this detail. Since the outputs of the inverse DFT and YJT are real-valued, rounding and clipping are necessary for discrete features to preserve semantic validity. However, we emphasize that the sign-bit alignment of TAB-DRW in the frequency domain is highly **insensitive to such mild nonlinear perturbations**. Moreover, since our method achieves strong fidelity preservation under appropriate choices of $(\gamma,\delta)$, clipping **occurs only rarely** and rounding magnitudes are **minimal**.
> > > >
> > > > Below are the results of an ablation study comparing Z-scores with and without rounding and clipping across five datasets, together with the frequency and magnitude of these operations. For **Magic** dataset there are no rounding or clipping happening since all the columns are continuous. For other datasets, the impact of these two post-processing operations on watermark detectability is negligible.
> > > >
> > > > **Table E. Detection performance of TAB-DRW with or without the rounding and clipping operations. Z-scores are averaged over 100 trials on tables with 1K rows. "Rounding magnitude" denotes the average rounding magnitude of discrete entries, and "Clipping ratio" denotes the fraction of discrete entries that are clipped.**
> > > >
> > > > | **Dataset** | **W/O round and clip** | **W/ round and clip** | **Rounding magnitude** | **Clipping ratio** |
> > > > |:-----------:|:-------------------------:|:------------------------:|:-------------------------:|:------------------------:|
> > > > | **Adult**       | 15.21 ± 1.00              | 12.81 ± 1.17             | 0.0911 ± 0.0015| 0.0008 ± 0.0004|
> > > > | **Magic**       | 27.34 ± 0.93              | 27.34 ± 0.93             | 0.0000 ± 0.0000| 0.0000 ± 0.0000|
> > > > | **Shoppers**    | 21.00 ± 1.15              | 18.18 ± 1.28             | 0.0969 ± 0.0042| 0.0244 ± 0.0036|
> > > > | **Default**     | 17.94 ± 0.95              | 15.98 ± 0.92             | 0.0542 ± 0.0018| 0.0151 ± 0.0013|
> > > > | **Drybean**     | 37.79 ± 1.02              | 38.03 ± 1.03             | 0.1285 ± 0.0027| 0.0145 ± 0.0022|

---

> > > > > ### Comment · Reviewer_vXSA · 2025-11-26
> > > > >
> > > > > Thanks for your rebuttal. I will keep my score but will increase the confidence score.

---

> > > > > > ### Author Response · Authors · 2025-11-26
> > > > > >
> > > > > > Dear Reviewer vXSA,
> > > > > >
> > > > > > Thank you very much for your insightful and constructive feedback, which has greatly helped us improve the paper. If you have any further concerns or questions, we would be more than happy to continue the discussion and provide any additional clarification.
> > > > > >
> > > > > > Best,\
> > > > > > Authors

---

### Official Review · Reviewer_neaq · 2025-10-31

**Soundness:** 3
**Presentation:** 3
**Contribution:** 2
**Rating:** 4
**Confidence:** 4

**Summary:**

The paper proposes a novel tabular watermarking approach for synthetically generated datasets utilizing invertible transformations such a Yeo-Johnson and discrete fourier transform (DFT). Experiments are shown on a wide variety of datasets as well as attacks to show the fidelity and robustness of the proposed approach.

**Strengths:**

(A) The paper tackles the difficult issue of handling both categorical and numerical data which can occur in different scales. The use of the invertible YJT handles the scale issue.

(B) Theoretical results are provided for the robustness of the approach under Gaussian noise as well as distortion of the watermarked dataset such as mean of the columns, correlations and the Wasserstein distance with the unwatermarked dataset.

(C) Experiments are shown on 5 datasets with a variety of competing approaches including MUSE, TabWak, TabularMark and performs well in most settings. These include fidelity and a variety of attacks.

**Weaknesses:**

(i) The presentation does not show the importance of categorical versus numerical columns for the watermarking results.

(ii) The paper does not show the performance to typical real-world permutation or spoofing type attacks.

**Questions:**

(1) How robust is the approach to a permutation attack which is typical? Does the approach assume an ordering of the columns which must be known to align with the secret bits?
(2) How easy is it for an adversary to scrub the watermark(*)?
(3) How easy is it for an adversary to spoof the watermark(**)?
(4) How does the approach work with different percentages of numerical versus categorical columns? When gender variables are flipped, do the corresponding row features make sense?


* Watermarks in the Sand: Impossibility of Strong Watermarking for Generative Models. https://arxiv.org/abs/2311.04378
** Adaptive and Robust Watermark for Generative Tabular Data. https://arxiv.org/abs/2409.14700

---

> ### Author Response · Authors · 2025-11-19
> **Response to the Reviewer neaq (1)**
>
> ## **Response to Reviewer neaq**
>
> We thank the reviewer for recognizing our efforts in **addressing the challenging problem of watermarking mixed-type tabular data**. We are also glad that the **theoretical and empirical contributions** of the paper are appreciated. Below, we address the reviewer’s concerns in detail.
>
> ---
>
> ### **W1&Q4: Impact of continuous–discrete column ratios and gender variable on TAB-DRW Performance**
>
> **R1:** **In Appendix F.1** we demonstrate the ratio of discrete versus continuous columns across five datasets used in our experiments. Specifically, the **Adult** and **Default** datasets contain only discrete features, while **Magic** and **Drybean** are dominated by continuous features. **Shoppers** dataset includes mixed ratio of continuous and discrete columns. Therefore, we believe the selected datasets provide a sufficiently **balanced evaluation across both discrete and continuous data types**. Empirical results show that TAB‑DRW maintains high fidelity and strong detectability and remains robust across five datasets, justifying the claim that our approach is **widely applicable to datasets with different continuous–discrete column ratios**.
>
> TAB-DRW also handles **low-cardinality categorical variables such as gender** in a conservative and adaptive manner. After standardization, each row is transformed via a row-wise DFT into a frequency-domain representation whose coefficients are linear combinations of all entries in that row. Watermark embedding (which is implemented through imaginary sign-bit alignment) operates on this joint representation rather than on any single feature. As a result, whether a sample’s gender value remains unchanged or flips depends on how the sample is positioned relative to the distribution of its other features. In particular, when we cluster unwatermarked samples using all non-gender features, we observe that the few samples whose gender flips after watermarking consistently lie near **cluster boundaries or appear misclustered even before watermarking**. This indicates that the watermarked gender value **remains semantically compatible** with the sample’s overall feature profile, regardless of whether a flip occurs.
>
> To support the claim above empirically, we conducted a case study on the gender variable using the Adult dataset. We refer the reviewer to **Appendix H (cf. Figure 9)** in the revised version for additional details, empirical observations, and findings.
>
> ---
>
> ### **W2&Q1: Lack of column permutation evaluation**
>
> **R2:** As clarified in Appendix F.5, we do not include column-level shuffling in our attack suite because the original column order can typically be **accurately recovered** by using **headers**, basic **statistical properties** (e.g., mean and standard deviation), or **semantic features** of each column. These characteristics are usually distinctive enough to make column permutation easily reversible. This assumption is also **standard** in prior works [1][2].
>
> ---
>
> ### **References**
>
> [1] Zhu, Chaoyi, et al. TabWak: A watermark for tabular diffusion models. *In The Thirteenth International Conference on Learning Representations*, 2025.
>
> [2] Fang, Liancheng, et al. MUSE: Model-Agnostic Tabular Watermarking via Multi-Sample Selection. *arXiv preprint arXiv:2505.24276*, 2025.

---

> > ### Author Response · Authors · 2025-11-19
> > **Response to the Reviewer neaq (2)**
> >
> > ### **Q2: Robustness to scrubbing attack**
> >
> > **R3:** We thank the reviewer for raising this interesting question. We acknowledge that TAB-DRW does not circumvent the theoretical impossibility results established in [3]. However, these impossibility results apply only under a threat model in which the adversary has access to both:
> >
> > - a **quality oracle** that reliably distinguishes high-quality tabular samples from degraded ones, and
> > - a **perturbation oracle** that proposes small, quality-preserving modifications that progressively reduce watermark detectability.
> >
> > To the best of our knowledge, **no existing work** provides either a quality oracle or a perturbation oracle for large-scale tabular data. Unlike text watermarking, where paraphrasing via a language model offers a feasible scrubbing mechanism, "paraphrasing" tabular data requires matching high-dimensional joint distributions and preserving intricate statistical properties. Whether LLMs or other generative models can reliably achieve this remains an open question. Consequently, implementing an idealized scrubbing attack for tabular watermarking is **still open and highly challenging** in practice.
> >
> > To further address your concern, we implemented two adaptive attacks that follow the spirit of scrubbing attacks. Motivated by reviewer D23T’s suggestion, these attacks are designed as oracle-free variants of scrubbing attacks. In both cases, the adversary is assumed to fully understand our pipeline—including the privacy-enhanced version used for real-world deployment (cf. Appendix B)—but does not know the secret key.
> >
> > The first attack, **Adaptive Row Deletion**, aims to corrupt the row ranking and thus impair rank-based bit retrieval. The attacker generates a random key, computes the normalized rank of each row (following lines 3–6 of Algorithm 3), and then deletes a block of rows whose ranks form a contiguous interval. For example, under strength 0.1, the adversary removes rows whose normalized ranks lie within a randomly selected interval of length 0.1 in $[0,1]$. This manipulation disrupts the ranking structure significantly more than random row deletion.
> >
> > **Table A. Robustness of TAB-DRW against adaptive row deletion attacks of varying strength. Z-scores are computed on tables with 5K rows and averaged over 100 independent trials.**
> >
> > |**Datasets**||**No-attack**|**Adv. Row Del.@0.1**|**Adv. Row Del.@0.2**|**Adv. Row Del.@0.5**|
> > |-:|:-:|:-:|:-:|:-:|:-:|
> > |**Adult**||29.55±1.12|28.55±1.44|26.35±2.61|18.79±4.47|
> > |**Magic**||61.42±1.02|56.71±2.98|49.36±5.77|28.41±7.28|
> > |**Shoppers**||40.74±1.26|36.47±2.62|30.40±3.71|17.12±4.18|
> > |**Default**||35.84±0.91|31.91±1.90|27.13±3.23|15.36±4.54|
> > |**Drybean**||85.05±0.67|79.27±2.67|71.67±5.04|52.39±9.99|
> >
> > The results show that TAB-DRW **remains highly detectable** even under substantial adaptive row-deletion attacks. Although detectability decreases slightly compared with random row deletion, the use of a secret key and the stable tree-based bit-storage enables TAB-DRW to be resilient to these attacks specifically crafted to disrupt the row-ranking process.
> >
> > ### **To be continued...**
> > ---
> >
> > ### **References**
> >
> > [3] Zhang, Hanlin, et al. Watermarks in the sand: Impossibility of strong watermarking for generative models. *arXiv preprint arXiv:2311.04378*, 2023.

---

> > > ### Author Response · Authors · 2025-11-19
> > > **Response to the Reviewer neaq (3)**
> > >
> > > **R3 (continued):**
> > > The second attack, **Rewatermarking**, targets the sign-bit alignment in the frequency domain. It exploits two properties of our privacy-enhanced TAB-DRW: first, its strong fidelity-preserving performance, and second, the fact that a watermark embedded with one key cannot be detected using another. An informed adversary can therefore repeatedly rewatermark the table with multiple different keys, aiming to perturb the original alignment and render the watermark undetectable to the original detection key.
> > >
> > > **Table B. Robustness of TAB-DRW against rewatermarking attacks of varying strength. Fidelity is averaged over four metrics across 10 independent trials, and the Z-scores are computed on tables with 5K rows and averaged over 100 independent trials. "Rewatermarking@$n$" denotes rewatermarking the table using $n$ randomly sampled keys.**
> > >
> > > |**Datasets**||**No-attack**||**Rewatermarking@1**||**Rewatermarking@3**||**Rewatermarking@10**||
> > > |-:|:-:|:-:|:-:|:-:|:-:|:-:|:-:|:-:|:-:|
> > > |||**Fidelity**|**Z-score**|**Fidelity**|**Z-score**|**Fidelity**|**Z-score**|**Fidelity**|**Z-score**|
> > > |**Adult**||0.799±0.006|29.55±1.12|0.787±0.008|23.66±1.17|0.772±0.006|16.26±1.09|0.766±0.009|17.26±1.34|
> > > |**Magic**||0.837±0.008|61.42±1.02|0.822±0.008|53.23±0.91|0.813±0.007|34.32±0.93|0.799±0.008|29.17±1.00|
> > > |**Shoppers**||0.854±0.009|40.74±1.26|0.847±0.008|31.97±1.15|0.829±0.009|20.14±1.09|0.813±0.008|16.67±1.09|
> > > |**Default**||0.836±0.013|35.84±0.91|0.827±0.011|32.85±1.00|0.811±0.013|19.40±1.07|0.809±0.010|26.28±1.18|
> > > |**Drybean**||0.849±0.017|85.05±0.67|0.832±0.017|44.79±0.81|0.801±0.017|29.47±0.83|0.806±0.014|33.77±0.95|
> > >
> > > From the results, we observe that TAB-DRW **remains highly detectable** even after ten rounds of rewatermarking, at which point the fidelity of the tabular data has already been noticeably degraded. These findings demonstrate that, without knowledge of the key, an attacker—despite understanding the TAB-DRW pipeline—cannot substantially disrupt the sign-bit alignment while preserving data fidelity.
> > >
> > > We will incorporate these attacks and the corresponding empirical results in the revised version of our paper.
> > >
> > > ---
> > >
> > > ### **Q3:Robustness to spoofing attack**
> > >
> > > **R4:**
> > > We thank the reviewer for mentioning spoofing attacks. In our setting, spoofing refers to making unwatermarked data showcase strong watermark signal under the detection using a specific key. TAB-DRW is explicitly designed to **make this extremely difficult**, as detailed in Appendix B. In conclusion, the privacy-enhanced variant applies a key-dependent column permutation before the YJT and DFT, creating a large key space and yielding empirically negligible cross-key collisions (i.e., a watermark embedded with one key cannot be misdetected under another). This arises because the imaginary sign-bit alignments induced by different keys (i.e., different column orders) are approximately unrelated.
> > >
> > > Since the **detection key is private**, an adversary without access to it cannot efficiently tune modifications to increase the Z-score; naive or heuristic modifications either fail to spoof the watermark or noticeably degrade data fidelity. Even if attackers imitate the imaginary sign pattern of the frequency-domain representation from a watermarked table, they still **cannot produce a detectable watermark signal** as long as their keys differ from the specific detection key. We refer the reviewer to Appendix B and Appendix G.4 for additional analysis and empirical justification.

---

### Official Review · Reviewer_Qt5B · 2025-10-31

**Soundness:** 2
**Presentation:** 3
**Contribution:** 2
**Rating:** 6
**Confidence:** 4

**Summary:**

This paper proposes a post-editing watermarking method for synthetic tabular data, called TAB-DRW. It embeds watermark signals in the frequency domain by modifying the imaginary components of the discrete Fourier transform (DFT) of the data. The authors provide theoretical guarantees on bounded distortion and robustness to Gaussian noise. Empirically, TAB-DRW outperforms or matches prior methods across multiple benchmarks in both fidelity and watermark detectability.

**Strengths:**

1. The proposed approach of watermarking tabular data through modifications in its discrete Fourier representation is both novel and well-motivated.

2. The paper establishes theoretical guarantees on distortion bounds and robustness under Gaussian noise.

3. The author conducts extensive experiments to empirically validate the performance of the proposed method. It considers both post-processing and generative watermarking methods as baselines and shows that TAB-DRW consistently matches or outperforms baselines on detectability, fidelity, and robustness.

**Weaknesses:**

Although the proposed method is both intuitively sound and theoretically well-grounded, I have the following concerns:

1. In line 252, the statement “compute a sum-based score over the selected entries” lacks sufficient detail. The exact formulation of this score is not clearly defined and should be explicitly described.

2. Following 1, if the pseudorandom bit is generated based on a subset of selected entries, the method may be vulnerable to attacks that perturb or alter the values of those specific entries.

3. The post-processing attacks introduced in Appendix F.5 appear relatively weak. For instance, the row-deletion attack removes only 10% (or 20% in Appendix G.3) of the rows, and the column-deletion attack deletes only two columns (three in Appendix G.3). Given that tabular datasets often contain dozens of columns, such fixed and small-scale deletions may not adequately stress-test robustness. A more comprehensive evaluation with varying degrees of row/column deletions would better demonstrate the method’s robustness to post-processing attacks.

**Questions:**

Could the author elaborate on how/why shrinking the imaginary part by a factor $\delta∈[−1,1]$ helps to limit the distortion?

---

> ### Author Response · Authors · 2025-11-19
> **Response to the Reviewer Qt5B**
>
> ## **Response to Reviewer Qt5B**
>
> We thank the reviewer for the recognition of the **novelty** in our method. We are also pleased that the paper’s **theoretical guarantees** and **empirical results** are appreciated. Below, we respond to the reviewer’s concerns point by point.
>
> ---
>
> ### **W1: Lack of algorithm details**
>
> **R1:** Thank you for pointing this out. Due to space constraints, we provide the formal description of the pseudorandom bit generation scheme in **Appendix C**. As shown in **Algorithm 3**, TAB-DRW computes the **sum of key-selected entries** as its sum-based score. We have clarified this explicitly in the main paper of the revised version.
>
> ---
>
> ### **W2: Robustness if those specific entries are perturbed**
>
> **R2:** Thanks for raising this concern. We agree that if an attacker knows exactly which entries are selected and introduces large, targeted modifications, the pseudorandom-bit generation could indeed be vulnerable. However, our design incorporates two mechanisms that substantially enhance robustness:
>
> - **Key-selected entries prevent targeted attacks.:** The subset of columns $\mathcal{I}$ used for score computation is determined by the **secret key $\kappa$**. An adversary without $\kappa$ cannot selectively and targetedly modify the specific entries that contribute to pseudorandom bit generation.
>
> - **Robustness of the bit-storage scheme mitigates small-level modifications.:**
> The sum-based rank statistic is **highly stable**, so small perturbations to a subset of columns (even those within $\mathcal{I}$) often do not change the bin to which the row belongs. As a result, the recovered pseudorandom bits typically remain correct. Even when the statistic shifts noticeably, it usually moves only to an adjacent bin. Our node–bit binding policy ensures that adjacent bins differ by only a single bit pair, which limits the effect of such shifts on the recovered bit sequence.
>
> Since the detector aggregates alignment statistics across all rows and all effective frequencies, sparse bit mismatches have only a limited impact on the final Z-score. Our empirical robustness evaluation in Section 4.3 reflects a realistic setting in which the adversary **does not know $\kappa$**. Figure 4 reports the 90% confidence interval of TPR@0.1%FPR over 100 randomized attacks. These results show that, with high probability (including cases where the attack corrupts a nontrivial fraction of key-selected entries) TAB-DRW **remains robust** owing to the mechanisms described above.
>
> ---
>
> ### **W3: Weak attack strength**
>
> **R3:** Thanks for your valuable suggestions. We would like to clarify serveral points about the robust evaluation setup.
>
> - For row-deletion attacks of varying strength, the impact on watermark detectability is predictable because the Z-score **scales approximately with $\sqrt{N}$**. For example, a table with 5K rows attains a Z-score that is approximately $\sqrt{5}$ times that of a table with 1K rows. Given this behavior, the two representative row-deletion strengths we reported already capture the trend of degradation under this attack.
> - The column-deletion attack is **intrinsically very strong**. Without access to high-quality unwatermarked data, an adversary who removes too many columns inevitably causes **semantic loss** and **substantial utility degradation**, making the resulting data unusable for downstream tasks such as classification or regression. In practice, more realistic adaptive strategies operate on values rather than structure, such as adding noise or applying quantization.
>
> To further address your concern and provide a more comprehensive robustness evaluation, we include additional empirical results under varying deletion strengths in the **Appendix G.3 (cf. Figure 8)** in the revised version. From the results, we observe that our method **ranks first or second** across most attack levels, demonstrating strong resilience even under high-strength attacks.
>
> ---
>
> ### **Q1: Why $\delta \in [−1,1]$ helps to limit the distortion**
>
> **R4:** By the Proposition 1 and Theorem 1, both the entry-wise differences and the upper bound of Wasserstein-2 distance are scaled by the same quantity $\alpha$, which depends linearly on $1+\delta$.
>
> - When $\delta = -1$, we have $\alpha=0$, resulting in zero distortion and no watermark signal.
> - As $\delta$ increases to 1, both the entry-wise distortion $\Delta x_{i,j}$ and the upper bound of Wasserstein-2 distance $\mathcal{W}_2(\rho_j, \rho_j^{\mathrm{wm}})$ increase monotonically.
>
> Therefore, reducing $\delta$ helps limit both entry-level and distribution-level distortion.

---

### Author Response · Authors · 2025-12-04
**Summary of Rebuttal (I)**

Dear Area Chair,

We fully understand the additional burden caused by the system information leakage and sincerely appreciate your time and effort in reassessing our submission and the review process. For your convenience, we provide below a summary of the discussion and how we addressed the concerns raised by the reviewers.

---
### **Summary of discussion phase**

During the discussion, Reviewer **D23T** (score 4) stated that *“the authors have diligently addressed my other questions and concerns,”* with one last question regarding the performance of the baseline TabWak under our newly proposed adaptive attacks, which we addressed in our follow-up response. Reviewer **vXSA** (score 6) expressed appreciation for our rebuttal and subsequently *increased the confidence score to 5*. Reviewers **Qt5B** (score 6) and **neaq** (score 4) did not participate in the discussion.

---
### **Summary of initial review and How we have addressed them**

1. **Reviewer Qt5B** indicated that our proposed method is **novel**, **well-motivated**, and **theoretically grounded**. Below, we summarize the reviewer’s concerns and how we addressed them in our rebuttal and the revised paper.
   - **Lack of algorithmic details:** We added a clear description of how to compute the sum-based score in **Lines 262–264**.
   - **Robustness under specific perturbation:** We clarified that (1) the secret-key mechanism and (2) the customized bit-storage scheme ensure robustness.
   - **Need for robustness experiments under varying degrees of row/column deletion:** We conducted an empirical evaluation of the robustness of various tabular watermarking methods under row and column deletion attacks with strengths ranging from 0.1 to 0.9 in increments of 0.1. We included the results in **Appendix G.3 (Figure 8)**.
   - **Effect of $\delta$ on limiting distortion:** We explained this using the theoretical results in Proposition 1 and Theorem 1.

2. **Reviewer neaq** acknowledged that our paper addresses the challenging problem of watermarking tabular data with mixed data types. Below, we summarize the reviewer’s concerns and how we addressed them in our rebuttal and the revised paper.
   - **Impact of continuous–discrete column ratios and the gender variable on TAB-DRW performance:** We reported the ratio information of the benchmark datasets in **Appendix F.1** to show that our evaluation spans varying ratios and that TAB-DRW performs consistently well. We also added a case study in **Appendix H (Figure 9)** demonstrating that TAB-DRW handles low-cardinality categorical variables, such as "gender", in a conservative, semantic-preserving way.
   - **Robustness under column permutation:** We clarified in **Appendix F.5** that the original column order can be accurately recovered using headers, basic statistical properties, or semantic features of each column. This assumption is also standard in prior work [1][2].
   - **Need for robustness experiments under scrubbing and spoofing attacks:** We introduced two scrubbing attacks targeting the rank-based bit retrieval and sign-bit alignment mechanisms of TAB-DRW, as well as a distillation-based spoofing attack. The results are included in **Section 4.4 (Table 4)** and **Appendix G.3 (Table 21)**, respectively.

3. **Reviewer vXSA** stated that our scheme is **novel**, **effective**, and supported by **rigorous** empirical evaluation. Below, we summarize the reviewer’s concerns and how we addressed them in our rebuttal and the revised paper.
   - **Column selection for watermarking:** We clarified in **Section 2.1 (Remark 2)** that we did not employ any specialized column-selection strategy and excluded only columns with extreme distributions. We also added an ablation study in **Appendix G.1 (Table 14)** showing that TAB-DRW is insensitive to column selection choices.
   - **Motivation for choosing $k \bmod 4 = 0 \text{ or } 3$ for bit-pair assignment:** We derived our node-bit binding policy as a variant of 2-Gray code and included a comparison with 1-Gray code in **Appendix G.1 (Tables 12–13)**.
   - **Discrepancy between theoretical analysis and practical implementation:** We justified the soundness of our idealized robustness model with a case study in **Appendix D (Table 5)** and showed that YJT refitting has negligible impact on data distribution and watermark signals.
   - **Impact of rounding and clipping on watermark detectability:** Experiments in **Appendix G.1 (Table 15)** show that TAB-DRW is highly insensitive to these operations, with clipping occurring rarely and rounding magnitudes remaining minimal.

---

> ### Author Response · Authors · 2025-12-04
> **Summary of Rebuttal (II)**
>
> 4. **Reviewer D23T** indicated that our paper is **well-written**, the scheme is **lightweight**, and it is supported by **extensive** empirical evaluation. Below, we summarize the reviewer’s concerns and how we addressed them in our rebuttal and the revised paper.
>    - **Need for robustness experiments under targeted attacks:** We introduced two adaptive attacks targeting the rank-based bit retrieval and sign-bit alignment mechanisms of TAB-DRW, with results added in **Section 4.4 (Table 4)**.
>    - **Gaussian setting in robustness analysis:** We clarified that the Gaussian setting serves as a representative case for robustness analysis and derived new theoretical results under a sub-Gaussian setting in **Appendix E.4 (Theorem 4 & Remark 5)**, covering a broader class of light-tailed distributions.
>    - **Runtime cost on very large tables:** We added runtime evaluation experiments on tables with 100K rows in **Appendix G.2 (Table 18)**.
>    - **Impact of rounding and clipping on downstream tasks:** We clarified that rounding and clipping are necessary operations to preserve semantic information and do not harm downstream performance.
>
> ---
>
> ### **References**
>
> [1] Zhu, Chaoyi, et al. TabWak: A watermark for tabular diffusion models. *In The Thirteenth International Conference on Learning Representations*, 2025.
>
> [2] Fang, Liancheng, et al. MUSE: Model-Agnostic Tabular Watermarking via Multi-Sample Selection. *arXiv preprint arXiv:2505.24276*, 2025.
>
>
> Best,
> Authors

---

### Meta-Review · Area_Chair_azjq · 2026-01-07

**Summary:**

This paper presents a method of watermarking tabular data based on a special normalization and the discrete Fourier transform. It modifies the imaginary parts of the DFT outcome, which can be easily detected when an attacker modifies values. For this, it designs a row-wise pseudorandom bit generation algorithm. To detect, it depends on a simple couting-based statistical testing. The authors conducted experiments with recent baselines at five different datasets.

**Reviewer Concerns:**

There are several concerns. In particular, the concerns of Reviewer neaq make sense to me as well: i) does this method assume a special ordering of columns?, ii) how do you select columns to watermark?, iii) how robust the method is when the percentages of numerical vs. categoriam columns changes, etc.

However, I am more concerned about one point that no reviewers had raised. The discrete Fourier transform assumes time series with a periodic boundary condition. When we say it as a graph, it corresponds a ring graph, each node of which represents a column in the case of this paper. However, tabular data cannot assume periodic boundary condition and a ring graph. I think the fundamental design of this paper lacks theoretical soundeness. I think they have chosed other types of transformation that do not rely on any ordering of columns with the periodic boundary condition. Becase of this reason, this paper's method may be obviously vulnerable to a specific ordering of columns. A different ordering produces a different DFT outcome and in the worse case, it can be very vulnerable to attacks.

**Reviewer Scores:**

I think reviewers still remain at their original scores. When I checked the answers, they cannot successfully address those fundamental weaknesses.

---

### Decision · Program_Chairs · 2026-01-26

Reject